# Feature Alignment with Equivariant Convolutions for Burst Image Super-Resolution

## Abstract

Burst image processing (BIP), which captures and integrates multiple frames into a single high-quality image, is widely used in consumer cameras. As a typical BIP task, Burst Image Super-Resolution (BISR) has achieved notable progress through deep learning in recent years. Existing BISR methods typically involve three key stages: alignment, upsampling, and fusion, often in varying orders and implementations. Among these stages, alignment is particularly critical for ensuring accurate feature matching and further reconstruction. However, existing methods often rely on techniques such as deformable convolutions and optical flow to realize alignment, which either focus only on local transformations or lack theoretical grounding, thereby limiting performance. To alleviate these issues, we propose a novel framework for BISR, featuring an equivariant convolution-based alignment, ensuring consistent transformations between the image and feature domains. This enables the alignment transformation to be learned via explicit supervision in the image domain and easily applied in the feature domain in a theoretically sound way, effectively improving alignment accuracy. Additionally, we design an effective reconstruction module with advanced architectures for upsampling and fusion to obtain the final BISR result. Extensive experiments on BISR benchmarks show our superior performance in both quantitative metrics and visual quality.

## 1 Introduction

Image super-resolution is an important task in image processing. Conventionally, it's mainly dealt with in the context of Single Image Super Resolution (SISR) [1, 2] and significant progress has been made in the last decades. By the advances in image acquisition technologies, a new kind of super-resolution technique, Burst Image Super-Resolution (BISR) [3, 4] has emerged as an increasingly valuable alternative. Unlike SISR, BISR reconstructs a high-resolution (HR) image by leveraging a sequence of low-resolution (LR) images captured in rapid succession, making it inherently more robust to noise and artifacts. Despite its advantages, BISR faces significant challenges, including accurate alignment for handling motion variations and effective multi-frame fusion.

The general pipeline for BISR typically involves three key stages: alignment, upsampling, and fusion, with their order and implementation varying across methods. Among them, alignment plays a crucial role in addressing spatial misalignments between successive frames, enabling accurate feature matching and high-quality reconstruction. Early methods [5, 6] mainly relied on Deformable Convolution Networks (DCNs) [7] for alignment, owing to their strong ability in modeling spatial transformations. Recently, optical flow [8] was adopted in the BurstM [9] method, showing better feature alignment performance than DCN, leveraging its explicit supervision in the image domain and a stronger ability to capture global transformations. However, since the transformation is estimated in the image domain, it may not be strictly applicable to the feature space without further constraints on

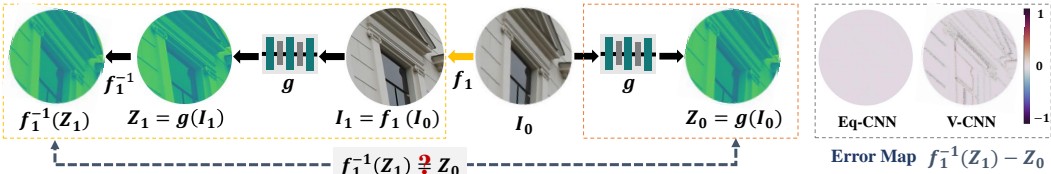

Figure 1: Illustration of transformation consistency in vanilla (V-CNN) and equivariant (Eq-CNN) convolutional networks. $f_1$ denotes a transformation (rotation in this example) and $g$ is a CNN that extracts features from images. Suppose $I_1$ is the image obtained by applying $f_1$ to $I_0$, i.e., $I_1 = f_1(I_0)$, and $Z_0$ and $Z_1$ are features extracted from $I_0$ and $I_1$, respectively. We expect that $Z_1$ can be close to $f_1(Z_0)$, the affine transformation of $Z_0$, such that one can align $Z_1$ to $Z_0$ in the feature domain by applying the inverse transformation $f_1^{-1}$, which can be learned by explicit supervision in the image domain. The right box compares the error between $f_1^{-1}(Z_1)$ and $Z_0$, and it can be observed that Eq-CNN can more effectively achieve this goal than V-CNN.

the feature extractors, as intuitively illustrated in Figure 1. As a result, the feature alignment could be less accurate, as shown in Section 3.1, which negatively affects the final performance.

To alleviate the limitations of alignment in existing methods, we propose to leverage equivariant convolutional networks (Eq-CNNs) [10, 11, 12] with learnable transformation matrices as a solution. Compared with vanilla convolutional neural networks (V-CNNs) [13, 14], Eq-CNNs can extract features that are theoretically equivariant to input images under certain spatial transformations, e.g., rotation and translation. Then, if each source frame within burst images can be approximately modeled by an affine transformation (or more specifically, rotation plus translation) of the reference frame due to the acquisition mechanism [3], such a property of Eq-CNNS enables us to learn the transformation (or its inverse) with the image domain supervision and then apply the inverse transformation in the feature domain to achieve an easy while theoretically sound alignment from the source frame to the reference one, as illustrated in Figure 1.

With the aligned features as aforementioned, we further designed a reconstruction module for upsampling and fusion to generate the final sRGB image using advanced techniques. Specifically, considering its ability in capturing intricate inter-frame correlations, we adopt the Multi-Dconv Head Transposed Attention (MDTA) block [15] for feature interaction among frames; and due to its flexibility in multi-scale upsampling, we use the implicit neural representation (INR) technique [16] to upsample the features for final fusion, following [9].

To summarize, our contributions are as follows:

- We propose a new alignment framework for BISR based on Eq-CNN, which enables us to learn the alignment transformation with image domain supervision and apply it in the feature domain in a theoretically sound way. The corresponding theoretical analysis also advances the theory of Eq-CNN to a certain extent.

- We incorporate the proposed alignment framework with advanced techniques for upsampling and fusion, including Restomer and INR, and build a new deep model for BISR.

- We apply the proposed model to BISR benchmarks, demonstrate its superiority against current state-of-the-art methods.

## 2 Related Work

### 2.1 Burst image super-resolution

BISR and its related task, Muti Frame Super-Resolution (MFSR), have been extensively studied using both traditional approaches and deep learning techniques. The pioneering work by Tsai et al. [17] tackled the problem in the frequency domain, while subsequent research [18, 19, 20] put more focus on the spatial domain for resolution enhancement. With the rapid advances of deep learning, significant progress has been made. Initial studies [21, 22] employed relatively simple network architectures to address the MFSR task. Then, Bhat et al. [23] proposed a BISR pipeline

Table 1: Comparison of relative alignment error on $\times 4$ SyntheticBurst and BurstSR.

| Dataset | Type | Domain | Flow | Ours |
|---|---|---|---|---|
| SyntheticBurst | synthetic | image | **0.18** | 0.20 |
| | | feature | 1.03 | **0.94** |
| BurstSR | real | image | 0.26 | **0.17** |
| | | feature | 2.06 | **1.94** |

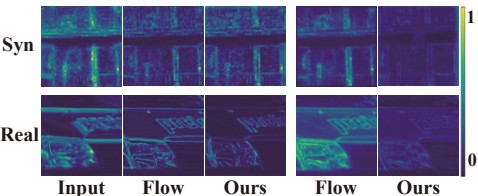

Figure 2: Error maps of aligned images (left) and features (right) .

incorporating alignment, feature fusion, and upsampling modules, together with the first real-world burst SR benchmark, inspiring numerous successive studies [24, 25, 5, 26, 6, 27, 28].

Within the BISR pipeline, alignment plays an important role, as highlighted by Kang et al. [9]. In previous methods [5, 6], DCN [7] is mainly adopted, but is insufficient for global transformation [9]. In contrast, Kang et al. [9] introduced optical flow [8] to achieve alignment, improving the performance. However, the image-domain estimated transformation may not be sufficient to achieve a theoretically sound and accurate feature alignment, motivating our development of a more principled alignment method via Eq-CNN.

In addition to alignment, fusion and upsampling have also benefited from recent architectural advances. Transformers [29] were used in [6, 26] for long-range modeling; QMambaBSR [28] introduced Mamba [30] for efficient sub-pixel integration; BSRD [27] employed diffusion models [31] for refined reconstruction; and BurstM [9] leveraged INRs [32, 16] for multi-scale upsampling.

### 2.2 Equivariant convolutions

One key factor behind the success of CNNs in computer vision is their inherent translation equivariance, which ensures spatial consistency. This principle has motivated the development of rotation-equivariant convolutions. GCNN [33] and HexaConv [34] enforce $\frac{\pi}{2}$ and $\frac{\pi}{3}$ rotational equivariance, while Xie et al. [35] extended this to near-continuous angles via Fourier-based filter parameterization, showing strong practical performance [12]. Leveraging such equivariance, Eq-CNN enables learning alignment transformations from image-domain supervision while preserving theoretical validity in the feature domain, which is an essential property in our alignment framework (see Section 3.1).

## 3 Proposed Method

We first discuss the motivation of our alignment framework for BISR. Then, we discuss the details of the proposed method. We also provide a theoretical justification for the validity of the proposed alignment framework.

### 3.1 Motivation

Alignment is a crucial component in the BISR pipeline. As discussed in the Introduction and Related Work, early deep learning approaches [5, 6] commonly employed DCN [7] for alignment. However, Kang et al. [9] pointed out that DCN struggles to capture global transformations and demonstrated that optical flow provides more effective feature alignment with supervision and global matching. Despite these advantages, the optical flow-based alignment has a theoretical limitation that should be noted. Specifically, the estimated transformations by optical flow are supervised in the image domain while applied in the feature domain, but there is no rigorous guarantee that the transformations of the two domains are consistent without further constraints on the feature extractor, as shown in Figure 1. To further investigate this issue, we compute the relative alignment error of BurstM, which uses the optical flow to align features, and compared with that of our method, as shown in Table 1 and Figure 2. It can be seen that both methods achieve comparable image alignment, but the feature alignment of optical flow is much worse.

To alleviate this issue, we propose a theoretically sound alignment approach based on Eq-CNN. We first briefly introduce the concept of equivariance in deep learning. Suppose $g$ is a deep feature extractor mapping from input to the feature space and $F$ is a transformation group, we say $g$ is equivariant with respect to $F$ if for any $f \in F$, it holds that $f(g(I)) = g(f(I))$, or equivalently,

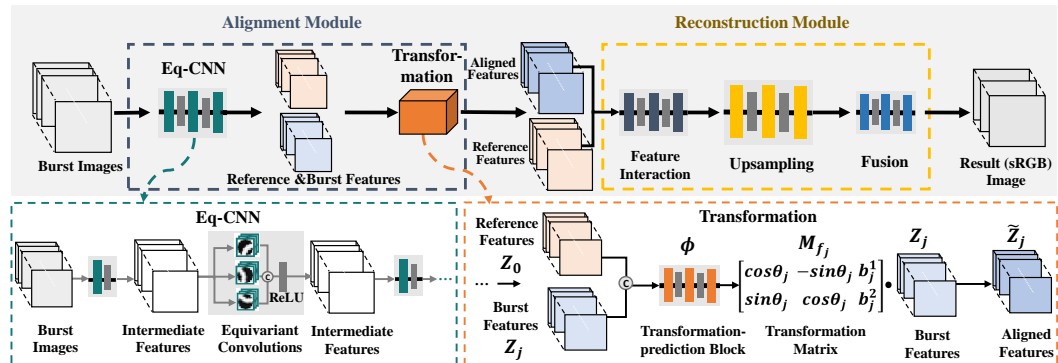

Figure 3: Overview of our proposed method. The top row shows the whole workflow. The bottom left shows the detailed equivariant convolution layers of Eq-CNN. The bottom right shows the process of feature alignment by predicted transformation, detailed in Section 3.2.2.

$g(I) = f^{-1}\big(g(f(I))\big)$ if $f$ is invertible. Note that, we abuse the notation $f$ a little to denote the same transformation applied in different domains. It is well known that V-CNNs are only equivariant under translation, while previous studies on equivariance [10, 35] constructed CNNs that are also equivariant with respect to rotation and reflection, which are often specifically referred to as Eq-CNNs.

This theoretical foundation of Eq-CNN allows us to design an alignment framework where a transformation (e.g., rotation and translation) estimated and supervised in the image domain remains valid in the feature domain. Considering that misalignments in burst images typically arise from slight camera shifts and can often be approximated by simple geometric transformations [3], such a rotation-translation modeling is reasonable. We leverage Eq-CNNs to extract features from input frames, and then apply explicit transformation matrices for alignment directly in the feature domain. The results in Table 1 and Figure 2 support this idea, that our method achieves a better feature alignment, especially on the real dataset.

## 3.2 Our method

### 3.2.1 Problem setting and processing pipeline

Given $B$ low-resolution (LR) RAW burst frames $\{I_j^L\}_{j=0}^{B-1}$ with each $I_j^L \in \mathbb{R}^{h \times w \times 1}$, we first process it a 4-channel format following the RGGB Bayer pattern [9, 6]. Then, one frame is selected as the reference frame, which serves as the reference for high-resolution (HR) reconstruction, and the rest frames are used to assist the reference one in super-resolving. The reconstructed HR image $I^S \in \mathbb{R}^{sh \times sw \times 3}$ is in sRGB format, where $s$ is the scale factor. As shown in Figure 3, our processing pipeline includes two main steps, i.e., alignment and reconstruction. The alignment step aims to extract and align features from the LR burst images using the Eq-CNN, and the reconstruction step tries to upsample and fuse the features to get the final reconstruction.

### 3.2.2 Alignment Module

Let $I_0^L$ denote the RAW LR reference frame and $\{I_j^L\}_{j=1}^{B-1}$ represent the remaining source frames in the burst image. Following the discussions in Section 3.1, we approximately model the relationship between each $I_j^L$ ($j \neq 0$) and $I_0^L$ as

$$I_j^L = f_j(I_0^L), \tag{1}$$

where $f_j$ is a rotation-translation transformation. After feature extraction using an Eq-CNN $g$, we can obtain features $Z_0 = g(I_0)$ and $Z_j = g(I_j)$, respectively. Assuming the equivariance property of $g$ strictly holds, we have that

$$Z_j = g(I_j) = g(f_j(I_0^L)) = f_j(g(I_0^L)) = f_j(Z_0). \tag{2}$$

This indicates if we can accurately estimate $f_j$ or $f_j^{-1}$ in the image domain, we can apply it to $Z_j$:

$$\tilde{Z}_j = f_j^{-1}(Z_j), \tag{3}$$

such that $\tilde{Z}_j$ is well aligned to $Z_0$. Therefore, we learn $f_j^{-1}$ via the image domain supervision with the following loss:

$$\mathcal{L}_{\text{align}} = \frac{1}{B-1} \sum_{j=1}^{B-1} \| f_j^{-1}(I_j^L) - I_0^L \|_2^2. \tag{4}$$

In practice, however, due to the discretization of the rotation angles in Eq-CNNs, we cannot strictly guarantee that $f_j^{-1}(Z_j) = Z_0$. Nevertheless, we can prove the following result:

**Proposition 1.** *For an images $I_0$ and $I_j$ of size $H \times W \times n_0$, a rotation-translation Eq-CNN $g(\cdot)$ with discretized angles, and a rotation-translation transformation $f_j(\cdot)$, let $Z_0 = g(I_0)$ and $Z_j = g(I_j)$ be the feature maps, where $Z_0, Z_j \in \mathbb{R}^{H \times W \times tC}$, and then the following result holds:*

$$\| f_j^{-1}(Z_j) - Z_0 \|_\infty \leq C_3 \left\| f_j^{-1}(I_j) - I_0 \right\|_2 + C_1 h^2 + C_2 p h t^{-1}, \tag{5}$$

*where $t, p, h, C_1, C_2, C_3$ constants.*

Proposition 1 suggests that we can minimize the distance between $f_j^{-1}(Z_j)$ and $Z_0$, the main goal of the Alignment module, through minimizing the distance between $f_j^{-1}(I_j^L)$ and $I_0^L$, which is we are trying to do by the loss defined in Eq. (4). The detailed version of Proposition 1 and its proof is are provided in Appendix A.4.

The next question is then how to parameterize the transformation $f_j^{-1}$. Since we assume $f_j$ is a rotation-translation transformation following [3] as discussed in Section 3.1, its inverse $f_j^{-1}$ is also a rotation-translation transformation, which can be parameterized using a matrix $M_{f_j} \in \mathbb{R}^{2 \times 3}$:

$$M_{f_j} = \begin{bmatrix} \cos\theta_j & -\sin\theta_j & b_j^1 \\ \sin\theta_j & \cos\theta_j & b_j^2 \end{bmatrix}, \tag{6}$$

where $\theta_j$ is the rotation angle, and $\boldsymbol{b}_j = (b_j^1, b_j^2)^T$ is the translation vector. Then, a pixel at location $(x_1, x_2)^T$ will be mapped to a new location $(x_1', x_2')^T$ after applying $f_j^{-1}$:

$$\begin{bmatrix} x_1' \\ x_2' \end{bmatrix} = M_{f_j} \cdot \begin{bmatrix} x_1 \\ x_2 \\ 1 \end{bmatrix} = \begin{bmatrix} x_1 \cos\theta_j - x_2 \sin\theta_j + b_j^1 \\ x_1 \sin\theta_j + x_2 \cos\theta_j + b_j^2 \end{bmatrix}. \tag{7}$$

Then we further parameterize $\{\theta_j, \boldsymbol{b}_j\}$ using a network block $\phi$, referred to as transformation prediction block in Figure 3, with $Z_0, Z_j$ as its input:

$$\{\theta_j, \boldsymbol{b}_j\} = \phi \left( \text{concat}[Z_0, Z_j] \right), \tag{8}$$

such that we can directly predict the alignment transformation $f_j^{-1}$ during inference.

### 3.2.3 Reconstruction Module

After alignment, the aligned features $\{\tilde{Z}_j\}_{j=0}^{B-1}$ of all frames (we let $\tilde{Z}_0 := Z_0$ for convenience) are then further processed for reconstructing the HR sRGB image.

**Feature interaction**. Before upsampling, feature interaction between reference and source frames is necessary for enriching frame information. Instead of concatenation [5, 6, 9] or pixel-wise attention [36], we adopt the MDTA block from Restormer [15], which performs channel-wise self-attention via cross-covariance, capturing global context and enabling more effective integration of reference-source features. The interaction process for the aligned features of all frames is as follows ($j = 0, \ldots, B-1$):

$$\mathbf{Q}_j = W_d^Q W_p^Q(\text{concat}[\tilde{Z}_j, \tilde{Z}_0]), \quad \mathbf{K}_j = W_d^K W_p^K(\text{concat}[\tilde{Z}_j, \tilde{Z}_0]), \tag{9}$$

$$\mathbf{V}_j = W_d^V W_p^V(\text{concat}[\tilde{Z}_j, \tilde{Z}_0]), \quad \hat{Z}_j = \mathbf{V}_j \cdot \text{Softmax}(\mathbf{K}_j \cdot \mathbf{Q}_j / \alpha_j) + \tilde{Z}_j, \tag{10}$$

where $\{\hat{Z}_j\}_{j=0}^{B-1}$ are the features after interaction, $W_p^{(\cdot)}$ and $W_d^{(\cdot)}$ refer to $1 \times 1$ pixel-wise and $3 \times 3$ depth-wise convolutions, respectively, and $\alpha_j$ is a learnable scaling parameter. The term $\text{Softmax}(\mathbf{K}_j \cdot \mathbf{Q}_j / \alpha_j)$ captures feature correlations and dynamically weights value vectors based on similarity, enabling context-aware fusion.

**Upsampling and fusion.** For upsampling, we adopt the LTE framework [16], utilizing INR and frequency domain processing to recover high-frequency details. LTE offers two key advantages for

BISR: (1) It enables multi-scale upsampling [9], allowing a single model to cover diverse scenarios; (2) Its grid sampling mechanism effectively recovers sub-pixel information, crucial for burst images with subtle camera shifts. After upsampling, we use a fusion block with channel attention to integrate the upscaled features, along with a skip connection to preserve reference frame information. To be specific, the upsampling and final fusion process can be formulated as

$$I^S = \text{PS}\left(\tilde{I}_0^L \uparrow + \text{Avg}_W\left(\{\Phi_{\text{up}_j}(\hat{Z}_j)\}_{j=0}^{B-1}\right)\right), \ \tilde{I}_0^L \uparrow = \text{Conv}_{1\times 1}(\text{Up}[I_0^L], . \tag{11}$$

where $I^S \in \mathbb{R}^{sh \times sw \times 3}$ is the final output in sRGB format, $\text{PS}(\cdot)$ denotes the pixel shuffle operation, $\text{Avg}_W(\cdot)$ refers to the weighted average operation with parameters learned via convolutions from $\hat{Z}_j$s, $\Phi_{\text{up}_j}(\cdot)$ denotes the LTE-based upsampling, $\tilde{I}_0^L \uparrow \in \mathbb{R}^{(sh/2) \times (sw/2) \times 12}$, $\text{Up}[\cdot]$ refers to the bilinear upsampling operation, and $\text{Conv}_{1\times 1}(\cdot)$ is a $1 \times 1$ convolution layer.

#### 3.2.4 Training loss

The whole network is trained in an end-to-end way using the following loss:

$$\mathcal{L} = \mathcal{L}_{\text{align}} + \mathcal{L}_{\text{fidelity}} = \mathcal{L}_{\text{align}} + \left\|I^S - I^{\text{GT}}\right\|_1, \tag{12}$$

where $\mathcal{L}_{\text{align}}$ is defined in Eq. (4), and $I^{\text{GT}}$ the ground truth HR sRGB image.

### 3.3 Theoretical results

This section provides further discussions on the theoretical aspects of our method, and the readers who are not interested in the theory of Eq-CNN can just skip it. As mentioned in Section 3.2.2, Proposition 1 theoretically guarantee the reasonability of our feature alignment strategy. However, its proof is not trivial and relies on the following theorem:

**Theorem 1.** *For an image $I_0$ of size $H \times W \times n_0$, a rotation-translation Eq-CNN $g(\cdot)$ with discretized angles, and a rotation-translation transformation $f_j(\cdot)$, under certain conditions, the following result holds:*

$$\|f_j^{-1}\big(g(f_j(I_0))\big) - g(I_0)\|_\infty \leq C_1 h^2 + C_2 p h t^{-1}, \tag{13}$$

*where $t, p, h, C_1, C_2$ are constants.*

Different from existing theories in Eq-CNN showing that input transformations can be predictably reflected in the feature domain, by measuring the error between $g(f_j(I_0))$ and $f_j(g(I_0))$, Theorem 1 further analyzes the residual errors caused by inverse transformation applied to these two objects in discrete settings of Eq-CNN, and suggests that such an error can also be upper-bounded. Such an analysis provides a theoretical understanding of how input-level inverse transformations affect feature relationships, which advances the theory of Eq-CNN to a certain extent. The detailed version of Theorem 1 and its proof are in Appendix A.3.

## 4 Experiments

In this section, we conduct experiments to validate the effectiveness of our proposed method. We first evaluate the proposed method on standard benchmarks for BSIR in comparison with existing methods. Then, we conduct ablation studies to demonstrate the reasonablity of our method, specifically concerning the alignment mechanism.

### 4.1 Experiments on BISR benchmarks

#### 4.1.1 Settings

**Datasets.** We follow previous studies [6, 9] and conduct experiments on two datasets: **(1) Synthet-icBurst Dataset** [3], which consists of 46,839 burst sequences for training and 300 for validation. Each burst sequence contains 14 RAW LR frames generated from an HR sRGB image using the standard pipeline [5, 9]. Specifically, unprocessing techniques [37] are firstly applied to simulate RAW sensor data, and random rotations and translations are implemented to simulate real camera motion. Following [9], we generate multi-scale LR images through random down-sampling (×2, ×3, ×4). Finally, Bayer mosaicking and random noise are added to more closely reproduce real-world imaging conditions. **(2) BurstSR Dataset** [23], which comprises 200 full-size RAW burst sequences, with 5,405 patches of size 80×80 extracted for training and 882 patches for validation. The LR images

Table 2: Quantitative results on SyntheticBurst and BurstSR datasets. The best and second-best results are highlighted in bold and underlined, respectively.

| Method | SyntheticBurst | | | | | | BurstSR | | BurstSR | |
| | x2 | | x3 | | x4 | | x4 | | Params.(M) | FLOPs(G) |
| | PSNR | SSIM | PSNR | SSIM | PSNR | SSIM | PSNR | SSIM | | |
| Bicubic | 38.30 | 0.948 | 33.94 | 0.886 | 33.02 | 0.862 | 42.55 | 0.962 | - | - |
| DBSR [23] | 40.51 | 0.965 | 40.11 | 0.959 | 40.76 | 0.959 | 48.05 | 0.984 | 13.01 | 111.71 |
| MFIR [24] | 41.25 | 0.971 | 41.81 | 0.972 | 41.56 | 0.964 | 48.33 | 0.985 | 12.13 | 121.01 |
| BIPNet [5] | 37.58 | 0.928 | 40.83 | 0.955 | 41.93 | 0.960 | 48.49 | 0.985 | 6.7 | 326.47 |
| Burstormer [6] | 37.06 | 0.925 | 40.26 | 0.953 | 42.83 | 0.973 | 48.06 | 0.986 | 2.5 | 38.33 |
| GMTNet [26] | - | - | - | - | 42.36 | 0.961 | 48.95 | 0.986 | - | 300 |
| BSRT-Small [25] | 40.64 | 0.966 | 42.30 | 0.975 | 42.72 | 0.971 | 48.57 | 0.986 | 4.92 | 178.82 |
| BSRT-Large [25] | 40.33 | 0.965 | 42.87 | 0.979 | **43.62** | **0.975** | 48.57 | 0.986 | 20.71 | 362.63 |
| BurstM [9] | 46.01 | **0.985** | 44.79 | 0.982 | 42.87 | 0.973 | 49.12 | **0.987** | 14.0 | 436.21 |
| Ours | **46.10** | 0.985 | **44.95** | **0.983** | 43.18 | 0.974 | **49.22** | **0.987** | 8.7 | 170.21 |

BIPNet    Burstormer    BSRT-L    BurstM    Ours    GT

Figure 4: Visual comparison of x4 BISR on the SyntheticBurst dataset.

are captured using a smartphone, while the HR ground truth images are obtained from a DSLR under the same scenes. Each LR burst sequence consists of 14 frames, and the scale factor between LR and HR images in this dataset is fixed (×4).

**Competing methods and evaluation metrics.** We evaluate our method against 8 representative ones, including traditional Bicubic interpolation and current state-of-the-art methods for the BISR task: DBSR [23], MFIR [24], BIPNet [5], GMTNet [26], Burstormer [6], BSRT [25], and BurstM [9]. We employ two widely used metrics, PSNR and SSIM, to quantitatively assess the reconstruction quality of each method. Additionally, we report model complexity metrics, including the number of parameters and GFLOPs, to show the computational efficiency of each method as a reference.

**Implementation details.** All experiments are implemented using PyTorch on an NVIDIA 4090 GPU. For the SyntheticBurst dataset, the initial learning rate is set to $1 \times 10^{-4}$ and gradually adjusted to $1 \times 10^{-6}$ over 300 epochs. The batch size is set to 1, and the patch size is $48 \times 48$. For the BurstSR dataset, we fine-tune the model pre-trained on SyntheticBurst following [9], using an initial learning rate of $1 \times 10^{-5}$ and CosineAnnealingLR to adjust it to $1 \times 10^{-6}$ over 30 epochs. The batch size is 1, and the patch size is $80 \times 80$. For other compared deep learning-based methods, we test using the author-released models, except GMTNet for which we directly quote the results reported in the original paper since the model is not released.

### 4.1.2 Results

**Results on SyntheticBurst Dataset** [3]. We present the quantitative and qualitative evaluation results in Table 2 and Fig. 4, respectively, with full-size and additional visual results available in Appendix B because of space limitations.

As shown in Table 2, our method outperforms existing BISR approaches across nearly all evaluation metrics. Specifically, for the widely-used ×4 SR setting, our method achieves the results with a PSNR of 43.18 and an SSIM of 0.974, surpassing competing methods with comparable model complexities. This quantitatively demonstrates its effectiveness in reconstructing the original HR sRGB image.

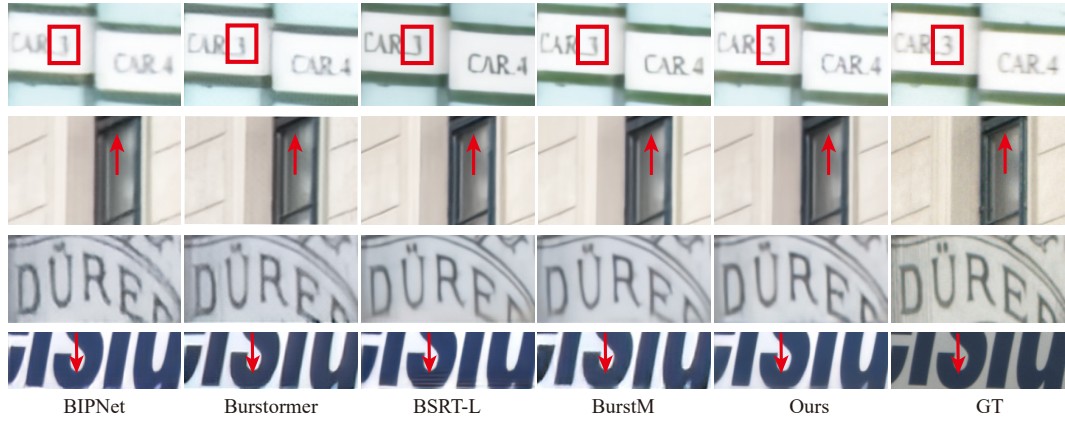

| BIPNet | Burstormer | BSRT-L | BurstM | Ours | GT |

Figure 5: Visual comparison of x4 BISR on the BurstSR dataset.

Notably, our approach outperforms the current state-of-the-art multi-scale BISR method BurstM [9] while requiring fewer parameters and less FLOPs, showing both its efficiency and effectiveness. Furthermore, our method consistently performs well across different SR factor settings, suggesting its promising generalization ability. Visual results in Fig. 4 show that our method achieves competitive performance in several aspects. For example, our approach better preserves fine-grained textual details while maintaining structural fidelity. In addition, the method shows its ability to suppress noise without introducing unexpected artifacts or severe color distortions. These qualitative advantages of our method are consistent with its quantitative performance. It should be mentioned that though our model achieves slightly lower numeric results due to its fewer parameters (8.7M) compared to BSRT-Large (20.71M), it delivers comparable visual quality. Additional visual results of other scales are provided in the supplementary material for a more comprehensive comparison.

**Results on BurstSR Dataset**. The quantitative results on the BurstSR dataset are summarized in Table 2. It can be seen that our method achieves the best performance in terms of both PSNR and SSIM among all competing ones. Note that, in this real dataset, although the degradation process of the LR burst images is unknown, and the relationship between the source and reference frames might not be more complex than assumed, our method still performs promisingly. This indicates that, though relatively simple, the rotation-translation assumption for the align transformation made in our model is rational and effective in real scenarios.

The visual results in Fig. 5 further validate the effectiveness of our approach. Overall, our method keeps more fine-grained details and produces fewer unexpected artifacts compared with existing methods. For example, as shown in the first row, our result better keeps the morphology of characters and digital numbers, and in the last row, our method can better suppress artifacts while producing relatively sharper edges. More visual results on this dataset are provided in the Appendices.

## 4.2 Ablation Study

In this subsection, we conduct experiments on the SyntheticBurst dataset at ×4 scale to validate the rationality and effectiveness of the proposed alignment framework in our model. The overall quantitative and visual results are summarized in Table 3 and shown in Fig. 6 - Fig. 7, respectively.

**Effectiveness of the overall alignment module (a) & (b).** We first replace the whole alignment module in our method with implicit alignment strategies using Restormer (a) and deformable convolutions (b). As shown in Table 3, both methods exhibit significant performance degradation, which can be more intuitively observed in the visual results illustrated in Fig. 6 (a) and (b), that the fine-grained textures are not well kept. This can be attributed to the misalignment of features, as can be observed in Fig. 7 (a) and (b). These results clearly substantiate the effectiveness of our alignment module.

**Effectiveness of equivariant feature extraction (c).** We then conduct an ablation study by replacing the ENet, which is an Eq-CNN, with a V-CNN without the rotation equivariance for feature extraction. As shown in Table 3 (c) and Fig. 6 (c), this variant exhibits a noticeable performance drop compared to the proposed model in quantitative metrics and also produces blurry textures. The reason can be attributed to the lack of consistency of the alignment transformations between the image and feature domains, leading to mismatching among aligned features. Another interesting observation

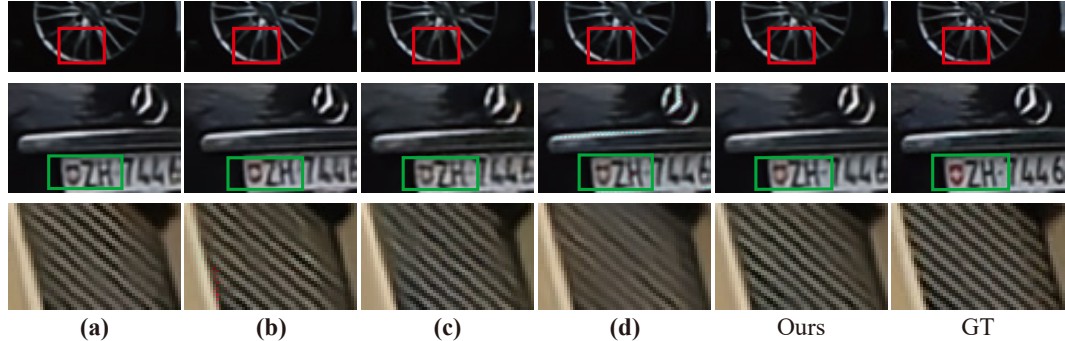

| **(a)** | **(b)** | **(c)** | **(d)** | Ours | GT |

Figure 6: Visualization of the ablation for ×4 BISR on SyntheticBurst. Settings (a)-(d) are in Table 3.

Table 3: Ablation Study on ×4 SyntheticBurst

| Settings | PSNR | SSIM | Params.(M) |
|---|---|---|---|
| (a) Align with RT | 42.97 | 0.972 | 9.0 |
| (b) Align with DConv | 42.81 | 0.970 | 11.5 |
| (c) w/o Eq-CNN | 42.76 | 0.971 | 10.1 |
| (d) w/o T-mat. | 42.80 | 0.972 | 8.7 |
| Ours | 43.18 | 0.974 | 8.7 |

*RT: Restormer [15]
*DConv: Deformable convolution network [7]
*w/o Eq-CNN: Replacing Eq-CNN with V-CNN
*w/o T-mat: Removing the transformation matrix

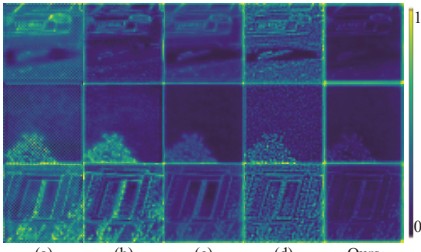

Figure 7: Error maps of aligned features for ablation studies on ×4 burst super-resolution using the SyntheticBurst dataset. Detailed settings of (a)-(d) can be referred to Table 3 and Section 4.2.

is that, though it does not perform well in the quantitative metrics, the visual results of this variant are comparable or even look better than that of other ablation variants as shown in Fig. 6, and the alignment error in features is also significantly smaller than that of variants (a) and (b) as depicted in Fig. 7. This can be due to the explicit alignment mechanism using the learnable transformation and the translation-equivariance of V-CNNs, which indirectly suggests the effectiveness of our approach.

**Effectiveness of the learnable transformation matrix (d).** We then remove the transformation matrix, denoted as "w/o T-mat." in Table 3, and such a variant can be seen as implementing implicit alignment with the Eq-CNN. It can be observed from Fig. 7 (d) that this leads to obvious feature misalignment and correspondingly inferior performance both in quantitative metrics and visual effects, highlighting the crucial role of explicit alignment.

# 5 Conclusion and Limitation

In this work, we have proposed a new method for BISR. The key consideration of our method is that we have designed a new effective alignment framework for the BISR task with Eq-CNN. Within the proposed alignment framework, by the equivariance property of Eq-CNN, the align transformation can be learned with explicit image domain supervision and directly applied in the feature domain in a theoretically sound way. In addition, we have introduced effective upsampling and fusion blocks using advanced neural architectures, including MDTA from Restormer and INR. Extensive experiments on two representative BISR benchmarks have been conducted, showing the effectiveness of the proposed method, both quantitatively and visually, against current state-of-the-art methods.

Despite its promising performance for BISR, our method still has limitations that need further investigation. For example, currently, the transformation considered in our model is restricted to rotation and translation due to the ability of existing Eq-CNNs, which may not be precise enough to characterize the relationship between the reference and source frame in complex real-world scenarios. Tackling this issue requires developing new techniques and theories for equivariance networks, which could not only enhance the availability of our method in real applications but also advance the study of equivariance in deep learning.

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

# A Theorem and Proofs

In this section, we present a comprehensive version of Theorem 1 and Proposition 1, which are briefly introduced in the main text, along with the related lemmas and proofs, aiming to provide a solid theoretical foundation for our proposed method.

It should be noted that we follow the previous works, and consider the equivariance on the orthogonal group $O(2)$[1]. Formally, $O(2) = \{A \in \mathbb{R}^{2\times2}|A^T A = I_{2\times2}\}$, which contains all rotation and reflection matrices. Without ambiguity, we use $A$ to parameterize $O(2)$. We consider the Euclidean group $E(2) = \mathbb{R}^2 \rtimes O(2)$ ($\rtimes$ is a semidirect-product), whose element is represented as $(x, A)$. Restricting the domain of $A$ and $x$, we can also use this representation to parameterize any subgroup of $E(2)$. The input image can be modeled as a function defined on $\mathbb{R}^2$, denoted as $r(x)$. The intermediate feature map can be modeled as a function defined on $E(2)$, denoted as $e(x, A)$. We denote the function spaces of $r$ and $e$ as $C^\infty(\mathbb{R}^2)$ and $C^\infty(E(2))$, respectively.

## A.1 Remark 1 and the Proof

**Notations.** For an input $r \in C^\infty(\mathbb{R}^2)$, transformations $\tilde{A} \in O(2)$ and $\tilde{b} \in \mathbb{R}^2$, $\tilde{A}$ acts on $r$ by

$$f_{\tilde{A}\tilde{b}}^R[r](x) = r(\tilde{A}^{-1}(x - \tilde{b})), \forall x \in \mathbb{R}^2. \tag{14}$$

For a feature map $e \in C^\infty(E(2))$, $E(2) = \mathbb{R}^2 \ltimes O(2)$, and a transformation $\tilde{A} \in O(2)$, $\tilde{A}$ act on $e$ by

$$f_{\tilde{A}\tilde{b}}^E[e](x, A, \tilde{b}) = e(\tilde{A}^{-1}(x - \tilde{b}), \tilde{A}^{-1}A), \forall (x, A) \in E(2). \tag{15}$$

Let $\Psi$ denote the convolution on the input layer, which maps an input $r \in C^\infty(\mathbb{R}^2)$ to a feature map defined on $E(2)$:

$$\Psi[r](x, A) = \int_{\mathbb{R}^2} \varphi_{in}\left(A^{-1}\delta\right) r(x - \delta)d\sigma(\delta), \ \forall(x, A) \in E(2), \tag{16}$$

where $\sigma$ is a measure on $\mathbb{R}^2$ and $\varphi$ is the proposed parameterized filter. $\Phi$ denotes the convolution on the intermediate layer, which maps a feature map $e \in C^\infty(E(2))$ to another feature map defined on $E(2)$:

$$\Phi[e](x, B) = \int_{O(2)}\int_{\mathbb{R}^2} \varphi_A\left(B^{-1}\delta\right) e(x - \delta, BA)d\sigma(\delta)dv(A), \ \forall(x, B) \in E(2), \tag{17}$$

where $v$ is a measure on $O(2)$, $A, B \in O(2)$ denote orthogonal transformations in the considered group, and $\varphi_{\tilde{A}}$ indicates the filter with respect to the channel of the feature map indexed by $\tilde{A}$, i.e., $e(x, A)|_{A=\tilde{A}}$. $\Upsilon$ denotes the convolution on the final layer, which maps a feature map $e \in C^\infty(E(2))$ to a function defined on $\mathbb{R}^2$:

$$\Upsilon[e](x) = \int_{O(2)}\int_{\mathbb{R}^2} \varphi_{out}\left(B^{-1}\delta\right) e(x - \delta, B)d\sigma(\delta)dv(B), \ \forall x \in \mathbb{R}^2. \tag{18}$$

Then we will prove Remark 1.

**Remark 1.** *For $r \in C^\infty(\mathbb{R}^2)$, $e \in C^\infty(E(2))$ and $\tilde{A} \in O(2)$, the following results are satisfied:*

$$
\begin{aligned}
\Psi\left[f_{\tilde{A}\tilde{b}}^R[r]\right] &= f_{\tilde{A}\tilde{b}}^E\left[\Psi[r]\right], \\
\Phi\left[f_{\tilde{A}\tilde{b}}^E[e]\right] &= f_{\tilde{A}\tilde{b}}^E\left[\Phi[e]\right], \\
\Upsilon\left[f_{\tilde{A}\tilde{b}}^E[e]\right] &= f_{\tilde{A}\tilde{b}}^R\left[\Upsilon[e]\right],
\end{aligned}
\tag{19}
$$

*where $f_{\tilde{A}\tilde{b}}^R$, $f_{\tilde{A}\tilde{b}}^E$, $\Psi$, $\Phi$ and $\Upsilon$ are defined by (14), (15), (16), (17) and (18), respectively.*

---

[1]The rotation group $S$ represents a subgroup of $O(2)$, and it is also regarded as the discretization of $O(2)$ in this paper.

443    *Proof.* (1) For any $x \in \mathbb{R}^2$, $A \in O(2)$, and $\tilde{b} \in \mathbb{R}^2$ we can obtain

$$\Psi \left[ f_{\tilde{A}\tilde{b}}^{R} [r] \right] (x, A)$$
$$= \int_{\mathbb{R}^2} \varphi_{in} \left( A^{-1}\delta \right) f_{\tilde{A}\tilde{b}}^{R} [r] (x - \delta) d\sigma(\delta) \tag{20}$$
$$= \int_{\mathbb{R}^2} \varphi_{in} \left( A^{-1}\delta \right) r(\tilde{A}^{-1}(x - \delta - \tilde{b})) d\sigma(\delta).$$

444    Let $\hat{\delta} = \tilde{A}^{-1}\delta$, since $|det(\tilde{A})| = 1$, and we have

$$\int_{\mathbb{R}^2} \varphi_{in} \left( A^{-1}\delta \right) r(\tilde{A}^{-1}(x - \delta - \tilde{b})) d\sigma(\delta)$$
$$= \int_{\mathbb{R}^2} \varphi_{in} \left( A^{-1}\tilde{A}\hat{\delta} \right) r(\tilde{A}^{-1}(x - \tilde{b}) - \hat{\delta})) d\sigma(\hat{\delta})$$
$$= \int_{\mathbb{R}^2} \varphi_{in} \left( (\tilde{A}^{-1}A)^{-1}\hat{\delta} \right) r(\tilde{A}^{-1}(x - \tilde{b}) - \hat{\delta})) d\sigma(\hat{\delta}) \tag{21}$$
$$= \Psi[r](\tilde{A}^{-1}(x - \tilde{b}), \tilde{A}^{-1}A)$$
$$= f_{\tilde{A}\tilde{b}}^{E} [\Psi[r]] (x, A, \tilde{b}).$$

445    This proves that $\Psi \left[ f_{\tilde{A}\tilde{b}}^{R} [r] \right] = f_{\tilde{A}\tilde{b}}^{E} [\Psi [r]]$.

446    (2) Similar to the proof in (1), for any $x \in \mathbb{R}^2$, $B \in O(2)$, we can obtain

$$\Phi \left[ f_{\tilde{A}\tilde{b}}^{E} [e] \right] (x, B)$$
$$= \int_{\mathbb{R}^2} \int_{O(2)} \varphi_A \left( B^{-1}\delta \right) f_{\tilde{A}\tilde{b}}^{E} [e] (x - \delta, BA, \tilde{b}) d\sigma(\delta) v(A)$$
$$= \int_{\mathbb{R}^2} \int_{O(2)} \varphi_A \left( B^{-1}\delta \right) e(\tilde{A}^{-1}(x - \delta - \tilde{b}), \tilde{A}^{-1}BA) d\sigma(\delta) v(A)$$
$$= \int_{\mathbb{R}^2} \int_{O(2)} \varphi_A \left( B^{-1}\tilde{A}\hat{\delta} \right) e(\tilde{A}^{-1}(x - \tilde{b}) - \hat{\delta}, \tilde{A}^{-1}BA) d\sigma(\hat{\delta}) v(A) \tag{22}$$
$$= \int_{\mathbb{R}^2} \int_{O(2)} \varphi_A \left( (\tilde{A}^{-1}B)^{-1}\hat{\delta} \right) e(\tilde{A}^{-1}(x - \tilde{b}) - \hat{\delta}, \tilde{A}^{-1}BA) d\sigma(\hat{\delta}) v(A)$$
$$= \Phi [e] (\tilde{A}^{-1}(x - \tilde{b}), \tilde{A}^{-1}B)$$
$$= f_{\tilde{A}\tilde{b}}^{E} [\Phi [e]] (x, B, \tilde{b}).$$

447    (3) For any $x \in \mathbb{R}^2$, we can deduce that

$$\Upsilon \left[ f_{\tilde{A}\tilde{b}}^{E} [e] \right] (x)$$
$$= \int_{\mathbb{R}^2} \int_{O(2)} \varphi_{out} \left( B^{-1}\delta \right) f_{\tilde{A}\tilde{b}}^{E} [e] (x - \delta, B, \tilde{b}) d\sigma(\delta) v(B)$$
$$= \int_{\mathbb{R}^2} \int_{O(2)} \varphi_{out} \left( B^{-1}\delta \right) e(\tilde{A}^{-1}(x - \delta - \tilde{b}), \tilde{A}^{-1}B) d\sigma(\delta) v(B) \tag{23}$$
$$= \int_{\mathbb{R}^2} \int_{O(2)} \varphi_{out} \left( \left( \tilde{A}^{-1}B \right)^{-1} \hat{\delta} \right) e(\tilde{A}^{-1}(x - \tilde{b}) - \hat{\delta}, \tilde{A}^{-1}B) d\sigma(\hat{\delta}) v(B).$$
$$= \Upsilon[e](\tilde{A}^{-1}(x - \tilde{b}))$$
$$= f_{\tilde{A}\tilde{b}}^{R} [\Upsilon[e]] (x).$$

448    This proves that $\Upsilon \left[ f_{\tilde{A}\tilde{b}}^{E} [e] \right] = f_{\tilde{A}\tilde{b}}^{R} [\Upsilon [e]]$.      $\square$

 **A.2   Remark 2 and the Proof**

 **Notations.** We assume that an image $I \in R^{n \times n}$ represents a two-dimensional grid function obtained
 by discretizing a smooth function, i.e., for $i, j = 1, 2, \cdots, n$,

$$I_{ij} = r(\delta_{ij}), \tag{24}$$

 where $\delta_{ij} = \left( \left( i - \frac{n+1}{2} \right) h, \left( j - \frac{n+1}{2} \right) h \right)^T$. We represent $Z$ as a three-dimensional grid function
 sampled from a smooth function $e : \mathbb{R}^2 \times S \to \mathbb{R}$, i.e., for $i, j = 1, 2, \cdots, n$,

$$Z_{ij}^{A,\tilde{b}} = e(\delta_{ij}, A, \tilde{b}), \tag{25}$$

 where $\delta_{ij} = \left( \left( i - \frac{n+1}{2} \right) h, \left( j - \frac{n+1}{2} \right) h \right)^T$ and $A \in S$, $S$ is a subgroup of $O(2)$, and $\tilde{b} \in \mathbb{R}^2$ is
 translation. For $i, j = 1, 2, \cdots, p$, and $A, B \in S$, we have

$$\begin{aligned}
\tilde{\Psi}_{ij}^A &= \varphi_{in} \left( A^{-1} \delta_{ij} \right), \\
\tilde{\Phi}_{ij}^{B,A} &= \varphi_A \left( B^{-1} \delta_{ij} \right), \\
\tilde{\Upsilon}_{ij}^A &= \varphi_{out} \left( A^{-1} \delta_{ij} \right),
\end{aligned} \tag{26}$$

 where $\delta_{ij} = ((i - (p+1)/2) h, (j - (p+1)/2) h)^T$, $\varphi_{in}$, $\varphi_{out}$ and $\varphi_A$ are parameterized filters. Let

$$\begin{aligned}
\delta_{ij} &= \left( \left( i - \frac{p+1}{2} \right) h, \left( j - \frac{p+1}{2} \right) h \right)^T, \\
x_{ij} &= \left( \left( i - \frac{n+p+2}{2} \right) h, \left( j - \frac{n+p+2}{2} \right) h \right)^T.
\end{aligned} \tag{27}$$

 For $\forall A \in S$ and $i, j = 1, 2, \cdots, n$, the convolution of $\tilde{\Psi}$ and $I$ is

$$\left( \tilde{\Psi} \star I \right)_{ij}^A = \sum_{(\tilde{i},\tilde{j}) \in \Lambda} \varphi_{in} \left( A^{-1} \delta_{\tilde{i}\tilde{j}} \right) r \left( x_{ij} - \delta_{\tilde{i}\tilde{j}} \right), \tag{28}$$

 where $\Lambda$ is a set of indexes, denoted as $\Lambda = \{(i, j) | i, j = 1, 2, \cdots, p\}$. For any $B \in S$ and
 $i, j = 1, 2, \cdots, n$, the convolution of $\tilde{\Phi}$ and $Z$ is

$$\left( \tilde{\Phi} \star Z \right)_{ij}^B = \sum_{(\tilde{i},\tilde{j}) \in \Lambda, A \in S} \varphi_A \left( B^{-1} \delta_{\tilde{i}\tilde{j}} \right) e \left( x_{ij} - \delta_{\tilde{i}\tilde{j}}, BA \right), \tag{29}$$

 where $\Lambda = \{(i, j) | i, j = 1, 2, \cdots, p\}$. For $i, j = 1, 2, \cdots, n$, the convolution of $\tilde{\Upsilon}$ and $Z$ is

$$\left( \tilde{\Upsilon} \star Z \right)_{ij} = \sum_{(\tilde{i},\tilde{j}) \in \Lambda, B \in S} \varphi_{out} \left( B^{-1} \delta_{\tilde{i}\tilde{j}} \right) e \left( x_{ij} - \delta_{\tilde{i}\tilde{j}}, B \right) \tag{30}$$

 where $\Lambda = \{(i, j) | i, j = 1, 2, \cdots, p\}$.

 The transformations on $I$ and $Z$ are defined by

$$\begin{aligned}
\left( \tilde{f}_{A\tilde{b}}^R(I) \right)_{ij} &= f_{A\tilde{b}}^R[r](x_{ij}), \left( \tilde{f}_{A\tilde{b}}^{\tilde{E}}(Z) \right)_{ij}^{A\tilde{b}} = f_{A\tilde{b}}^E[e](x_{ij}, A, \tilde{b}), \\
&\forall i, j = 1, 2, \cdots, n, \forall A, \tilde{A} \in S.
\end{aligned} \tag{31}$$

 Then we will prove the Remark 2. We firstly introduce the following necessary lemma.

 **Lemma 1.** *For smooth functions* $r : \mathbb{R}^2 \to \mathbb{R}$ *and* $\varphi : \mathbb{R}^2 \to \mathbb{R}$, *if for* $\delta \in \mathbb{R}^2$, *the follow conditions*
 *are satisfied:*

$$\begin{aligned}
|r(\delta)| &\le F_1, |\varphi(\delta)| \le F_2, \\
\|\nabla r(\delta)\| &\le G_1, \|\nabla \varphi(\delta)\| \le G_2, \\
\|\nabla^2 r(\delta)\| &\le H_1, \|\nabla^2 \varphi(\delta)\| \le H_2, \\
\forall \|\delta\| &\ge (p+1/2)h, \varphi(\delta) = 0,
\end{aligned} \tag{32}$$

where $p, h > 0$, $\nabla$ and $\nabla^2$ denote the operators of gradient and Hessian matrix, respectively, then, $\forall \tilde{A} \in S, y \in \mathbb{R}$ the following results are satisfied:

$$\left| \int_{R^2} \varphi\left(\tilde{A}^{-1}\delta\right) r(x - \delta)\, d\sigma(\delta) - \sum_{i,j \in \Lambda} \varphi\left(\tilde{A}^{-1}\delta_{ij}\right) r(x - \delta_{ij})\, h^2 \right| \leq \frac{(p+1)^2 C}{4} h^4, \quad (33)$$

where $\Lambda = \{(i,j) | i, j = 1, 2, \cdots, p\}$, $\delta_{ij} = ((i - {}^{(p+1)}\!/\!_2)\, h, (j - {}^{(p+1)}\!/\!_2)\, h)^T$ and $C = F_1 H_2 + F_2 H_1 + 2 G_1 G_2$.

The specific proof of lemma 1 can be referred to [11]. Based on lemma 1, let us prove Remark 2.

**Remark 2.** *Assume that an image $I \in \mathbb{R}^{n \times n}$ is discretized from the smooth function $r : \mathbb{R}^2 \to \mathbb{R}$ by (24), a feature map $Z \in \mathbb{R}^{n \times n \times t}$ is discretized from the smooth function $e : \mathbb{R}^2 \times S \to \mathbb{R}$ by (25), $|S| = t$, and filters $\tilde{\Psi}$, $\tilde{\Phi}$ and $\tilde{\Upsilon}$ are generated from $\varphi_{in}$, $\varphi_{out}$ and $\varphi_A, \forall A \in S$, by (26), respectively. If for any $A \in S$, $x \in \mathbb{R}^2$, the following conditions are satisfied:*

$$\begin{aligned}
&|r(x)|, |e(x, A)| \leq F_1, \\
&\|\nabla r(x)\|, \|\nabla e(x, A)\| \leq G_1, \\
&\|\nabla^2 r(x)\|, \|\nabla^2 e(x, A)\| \leq H_1, \\
&|\varphi_{in}(x)|, |\varphi_A(x)|, |\varphi_{out}(x)| \leq F_2, \\
&\|\nabla \varphi_{in}(x)\|, \|\nabla \varphi_A(x)\|, \|\nabla \varphi_{out}(x)\| \leq G_2, \\
&\|\nabla^2 \varphi_{in}(x)\|, \|\nabla^2 \varphi_A(x)\|, \|\nabla^2 \varphi_{out}(x)\| \leq H_2, \\
&\forall \|x\| \geq {}^{(p+1)h}\!/\!_2, \ \varphi_{in}(x), \varphi_A(x), \varphi_{out}(x) = 0,
\end{aligned} \qquad (34)$$

*where $p$ is the filter size, $h$ is the mesh size, and $\nabla$ and $\nabla^2$ denote the operators of gradient and Hessian matrix, respectively, then for any $\tilde{A} \in S$, the following results are satisfied:*

$$\begin{aligned}
&\left\| \tilde{\Psi} \star \tilde{f}^R_{\tilde{A}\tilde{b}}(I) - \tilde{f}^{\tilde{E}}_{\tilde{A}\tilde{b}}\left(\tilde{\Psi} \star I\right) \right\|_\infty \leq \frac{C}{2}(p+1)^2 h^2, \\
&\left\| \tilde{\Phi} \star \tilde{f}^{\tilde{E}}_{\tilde{A}\tilde{b}}(Z) - \tilde{f}^{\tilde{E}}_{\tilde{A}\tilde{b}}\left(\tilde{\Phi} \star Z\right) \right\|_\infty \leq \frac{C}{2}(p+1)^2 h^2 t, \\
&\left\| \tilde{\Upsilon} \star \tilde{f}^{\tilde{E}}_{\tilde{A}\tilde{b}}(Z) - \tilde{f}^R_{\tilde{A}\tilde{b}}\left(\tilde{\Upsilon} \star Z\right) \right\|_\infty \leq \frac{C}{2}(p+1)^2 h^2 t,
\end{aligned} \qquad (35)$$

*where $C = F_1 H_2 + F_2 H_1 + 2 G_1 G_2$, $\tilde{f}^R_{\tilde{A}\tilde{b}}, \tilde{f}^{\tilde{E}}_{\tilde{A}\tilde{b}}, \tilde{\Psi}, \tilde{\Phi}$ and $\tilde{\Upsilon}$ are defined by (26) and (31), respectively. The operators $\star$ involved in Eq. (35) are defined in (28), (29) and (30), respectively, and $\|\cdot\|_\infty$ represents the infinity norm.*

*Proof.* For any $x \in \mathbb{R}, A, B \in S$, let

$$\hat{\Psi}[r](x, A) = \sum_{(\tilde{i}, \tilde{j}) \in \Lambda} \varphi_{in}\left(A^{-1}\delta_{\tilde{i}\tilde{j}}\right) r\left(x - \delta_{\tilde{i}\tilde{j}}\right), \qquad (36)$$

where $\Lambda = \{(\tilde{i}, \tilde{j}) | \tilde{i}, \tilde{j} = 1, 2, \cdots, p\}$. Then, for any $A \in S$, we can obtain

$$\hat{\Psi}[r](x_{ij}, A) = \left(\tilde{\Psi} \star I\right)^A_{ij}. \qquad (37)$$

1) By **Remark 1**, we know that $\Psi\left[f^R_{\tilde{A}\tilde{b}}[r]\right] = f^E_{\tilde{A}\tilde{b}}[\Psi[r]]$. Thus for any $A \in S$, we have

$$\begin{aligned}
&\left| \left(\tilde{\Psi} \star \tilde{f}^R_{\tilde{A}\tilde{b}}(I) - \tilde{f}^{\tilde{E}}_{\tilde{A}\tilde{b}}\left(\tilde{\Psi} \star I\right)\right)^A_{ij} \right| \\
&= \left| \hat{\Psi}\left[f^R_{\tilde{A}\tilde{b}}[r]\right](x_{ij}, A) - f^E_{\tilde{A}\tilde{b}}\left[\hat{\Psi}[r]\right](x_{ij}, A) \right| \\
&\leq \left| \hat{\Psi}\left[f^R_{\tilde{A}\tilde{b}}[r]\right](x_{ij}, A) - \frac{1}{h^2}\Psi\left[f^R_{\tilde{A}\tilde{b}}[r]\right](x_{ij}, A) \right| \\
&\quad + \left| f^E_{\tilde{A}\tilde{b}}\left[\hat{\Psi}[r]\right](x_{ij}, A) - \frac{1}{h^2} f^E_{\tilde{A}\tilde{b}}\left[\Psi[r]\right](x_{ij}, A) \right|.
\end{aligned} \qquad (38)$$

Let $\hat{r} = f_{\tilde{A}\tilde{b}}^R[r]$, and then it is easy to deduce that $\hat{r}$ satisfies the conditions in **lemma 1**. Then, by **lemma 1**,

$$
\begin{aligned}
&\left| \hat{\Psi}\left[ f_{\tilde{A}\tilde{b}}^R[r] \right] (x_{ij}, A) - \frac{1}{h^2} \Psi\left[ f_{\tilde{A}\tilde{b}}^R[r] \right] (x_{ij}, A) \right| \\
&= \frac{1}{h^2} \left| \hat{\Psi}\left[ f_{\tilde{A}\tilde{b}}^R[r] \right] (x_{ij}, A) h^2 - \Psi\left[ f_{\tilde{A}\tilde{b}}^R[r] \right] (x_{ij}, A) \right| \\
&= \frac{1}{h^2} \left| \sum_{(i,j) \in \Lambda} \varphi_{in}\left( A^{-1} \delta_{ij} \right) \hat{r}\left( x_{ij} - \delta_{ij} \right) h^2 - \int_{\mathbb{R}^2} \varphi_{in}\left( A^{-1} \delta \right) \hat{r}(x_{ij} - \delta) d\sigma(\delta) \right| \\
&\leq \frac{(p+1)^2 C}{4} h^2.
\end{aligned}
\tag{39}
$$

Besides, let $\hat{A} = \tilde{A}^{-1} A$ and $\hat{x}_{ij} = \tilde{A}^{-1}(x_{ij} - \tilde{b})$, and by **lemma 1**, we can also achieve,

$$
\begin{aligned}
&\left| f_{\tilde{A}\tilde{b}}^E\left[ \hat{\Psi}[r] \right] (x_{ij}, A) - \frac{1}{h^2} f_{\tilde{A}\tilde{b}}^E\left[ \Psi[r] \right] (x_{ij}, A) \right| \\
&= \frac{1}{h^2} \left| f_{\tilde{A}\tilde{b}}^E\left[ \hat{\Psi}[r] \right] (x_{ij}, A) h^2 - f_{\tilde{A}\tilde{b}}^E\left[ \Psi[r] \right] (x_{ij}, A) \right| \\
&= \frac{1}{h^2} \left| \left[ \hat{\Psi}[r] \right] (\tilde{A}^{-1}(x_{ij} - \tilde{b}), \tilde{A}^{-1} A) h^2 - \left[ \Psi[r] \right] (\tilde{A}^{-1}(x_{ij} - \tilde{b}), \tilde{A}^{-1} A) \right| \\
&= \frac{1}{h^2} \left| \sum_{(i,j) \in \Lambda} \varphi_{in}\left( A^{-1} \tilde{A} \delta_{ij} \right) r\left( \tilde{A}^{-1}(x_{ij} - \tilde{b}) - \delta_{ij} \right) h^2 - \int_{\mathbb{R}^2} \varphi_{in}\left( A^{-1} \tilde{A} \delta \right) r(\tilde{A}^{-1}(x_{ij} - \tilde{b}) - \delta) d\sigma(\delta) \right| \\
&= \frac{1}{h^2} \left| \sum_{(i,j) \in \Lambda} \varphi_{in}\left( \hat{A}^{-1} \delta_{ij} \right) r\left( \hat{x}_{ij} - \delta_{ij} \right) h^2 - \int_{\mathbb{R}^2} \varphi_{in}\left( \hat{A}^{-1} \delta \right) r(\hat{x}_{ij} - \delta) d\sigma(\delta) \right| \\
&\leq \frac{(p+1)^2 C}{4} h^2.
\end{aligned}
\tag{40}
$$

Thus, combining (38), (39) and (40), we can achieve

$$
\left| \hat{\Psi}\left[ f_{\tilde{A}\tilde{b}}^R[r] \right](x_{ij}, A, \tilde{b}) - f_{\tilde{A}\tilde{b}}^E\left[ \hat{\Psi}[r] \right](x_{ij}, A, \tilde{b}) \right| \leq \frac{C}{2}(p+1)^2 h^2.
\tag{41}
$$

In other word,

$$
\left| \left( \tilde{\Psi} \star \tilde{f}_{\tilde{A}\tilde{b}}^R(I) - \tilde{f}_{\tilde{A}\tilde{b}}^{\tilde{E}}\left( \tilde{\Psi} \star I \right) \right)_{ij}^A \right| \leq \frac{C}{2}(p+1)^2 h^2.
\tag{42}
$$

This proves the first inequality in (35).

2) For any $A, B \in S$, let $\hat{B} = \tilde{A}^{-1} B$, $r_A(x) = e(x, A)$, and $\hat{\Psi}_A$ be a operator defined in the formulation of (36), while correlated to $\varphi_A$. Then, for any $i, j = 1, 2, \cdots, n$, $B \in S$,

$$
\begin{aligned}
&\left| \left( \tilde{\Phi} \star \tilde{f}_{\tilde{A}\tilde{b}}^{\tilde{E}}(Z) - \tilde{f}_{\tilde{A}\tilde{b}}^{\tilde{E}}\left( \tilde{\Phi} \star Z \right) \right)_{ij}^B \right| \\
&= \left| \sum_{(\tilde{i}, \tilde{j}) \in \Lambda, A \in S} \varphi_A\left( B^{-1} \delta_{\tilde{i}\tilde{j}} \right) e\left( \tilde{A}^{-1}\left( x_{ij} - \delta_{\tilde{i}\tilde{j}} - \tilde{b} \right), \tilde{A}^{-1} B A \right) - \sum_{(\tilde{i}, \tilde{j}) \in \Lambda, A \in S} \varphi_A\left( B^{-1} \tilde{A} \delta_{\tilde{i}\tilde{j}} \right) e\left( \tilde{A}^{-1}(x_{ij} - \tilde{b}) - \delta_{\tilde{i}\tilde{j}}, \tilde{A}^{-1} B A \right) \right| \\
&\leq \sum_{A \in S} \left| \sum_{(\tilde{i}, \tilde{j}) \in \Lambda} \varphi_A\left( B^{-1} \delta_{\tilde{i}\tilde{j}} \right) r_{\hat{B}A}\left( \tilde{A}^{-1}(x_{ij} - \delta_{\tilde{i}\tilde{j}} - \tilde{b}) \right) - \sum_{(\tilde{i}, \tilde{j}) \in \Lambda} \varphi_A\left( B^{-1} \tilde{A} \delta_{\tilde{i}\tilde{j}} \right) r_{\hat{B}A}\left( \tilde{A}^{-1}(x_{ij} - \tilde{b}) - \delta_{\tilde{i}\tilde{j}} \right) \right| \\
&= \sum_{A \in S} \left| \sum_{(\tilde{i}, \tilde{j}) \in \Lambda} \varphi_A\left( B^{-1} \delta_{\tilde{i}\tilde{j}} \right) f_{\tilde{A}\tilde{b}}^R[r_{\hat{B}A}]\left( x_{ij} - \delta_{\tilde{i}\tilde{j}} \right) - \sum_{(\tilde{i}, \tilde{j}) \in \Lambda} \varphi_A\left( B^{-1} \tilde{A} \delta_{\tilde{i}\tilde{j}} \right) r_{\hat{B}A}\left( \tilde{A}^{-1}(x_{ij} - \tilde{b}) - \delta_{\tilde{i}\tilde{j}} \right) \right| \\
&= \sum_{A \in S} \left| \hat{\Psi}_A\left[ f_{\tilde{A}\tilde{b}}^R[r_{\hat{B}A}] \right] (x_{ij}, B, \tilde{b}) - f_{\tilde{A}\tilde{b}}^E\left[ \hat{\Psi}_A[r_{\hat{B}A}] \right] (x_{ij}, B, \tilde{b}) \right|.
\end{aligned}
$$

Then by (41), we can achieve that $\forall i, j = 1, 2, \cdots, n, B \in S$,

$$\left| \left( \tilde{\Phi} \star \tilde{f}_{\tilde{A}\tilde{b}}^{\tilde{E}} (Z) - \tilde{f}_{\tilde{A}\tilde{b}}^{\tilde{E}} \left( \tilde{\Phi} \star Z \right) \right)_{ij}^{B} \right| \le \frac{C}{2} (p+1)^2 h^2 t. \tag{43}$$

This proves the second inequality in (35).

3) For any $A, B \in S$, let $\hat{B} = \tilde{A}^{-1} B$, $r_A(x) = e(x, A)$, and $\hat{\Psi}_{out}$ be a operator defined in the formulation of (36), while correlated to $\varphi_{out}$.

Then, we have that $\forall i, j = 1, 2, \cdots, n$,

$$\left| \left( \tilde{\Upsilon} \star \tilde{f}_{\tilde{A}\tilde{b}}^{\tilde{E}} (Z) - \tilde{f}_{\tilde{A}\tilde{b}}^{\tilde{E}} \left( \tilde{\Upsilon} \star Z \right) \right)_{ij} \right|$$

$$= \left| \sum_{(\tilde{i}, \tilde{j}) \in \Lambda, B \in S} \varphi_{out} \big( B^{-1} \delta_{\tilde{i}\tilde{j}} \big) e \big( \tilde{A}^{-1} \big( x_{ij} - \delta_{\tilde{i}\tilde{j}} - \tilde{b} \big), \tilde{A}^{-1} B \big) - \sum_{(\tilde{i}, \tilde{j}) \in \Lambda, B \in S} \varphi_{out} \big( B^{-1} \tilde{A} \delta_{\tilde{i}\tilde{j}} \big) e \big( \tilde{A}^{-1} (x_{ij} - \tilde{b}) - \delta_{\tilde{i}\tilde{j}}, \tilde{A}^{-1} B \big) \right|$$

$$\le \sum_{B \in S} \left| \sum_{(\tilde{i}, \tilde{j}) \in \Lambda} \varphi_{out} \big( B^{-1} \delta_{\tilde{i}\tilde{j}} \big) r_{\hat{B}} \big( \tilde{A}^{-1} \big( x_{ij} - \delta_{\tilde{i}\tilde{j}} - \tilde{b} \big) \big) - \sum_{(\tilde{i}, \tilde{j}) \in \Lambda} \varphi_{out} \big( B^{-1} \tilde{A} \delta_{\tilde{i}\tilde{j}} \big) r_{\hat{B}} \big( \tilde{A}^{-1} (x_{ij} - \tilde{b}) - \delta_{\tilde{i}\tilde{j}} \big) \right|$$

$$= \sum_{B \in S} \left| \sum_{(\tilde{i}, \tilde{j}) \in \Lambda} \varphi_{out} \big( B^{-1} \delta_{\tilde{i}\tilde{j}} \big) f_{\tilde{A}\tilde{b}}^{R} [r_{\hat{B}}] \big( x_{ij} - \delta_{\tilde{i}\tilde{j}} \big) - \sum_{(\tilde{i}, \tilde{j}) \in \Lambda} \varphi_{out} \big( B^{-1} \tilde{A} \delta_{\tilde{i}\tilde{j}} \big) r_{\hat{B}} \big( \tilde{A}^{-1} (x_{ij} - \tilde{b}) - \delta_{\tilde{i}\tilde{j}} \big) \right|$$

$$= \sum_{B \in S} \left| \hat{\Psi}_{out} \left[ f_{\tilde{A}\tilde{b}}^{R} [r_{\hat{B}}] \right] (x_{ij}, B, \tilde{b}) - f_{\tilde{A}\tilde{b}}^{E} \left[ \hat{\Psi}_{out} [r_{\hat{B}}] \right] (x_{ij}, B, \tilde{b}) \right|.$$

Then by (41), we can achieve that $\forall i, j = 1, 2, \cdots, n$,

$$\left| \left( \tilde{\Upsilon} \star \tilde{f}_{\tilde{A}\tilde{b}}^{\tilde{E}} (Z) - \tilde{f}_{\tilde{A}\tilde{b}}^{\tilde{E}} \left( \tilde{\Upsilon} \star Z \right) \right)_{ij} \right| \le \frac{C}{2} (p+1)^2 h^2 t. \tag{44}$$

This proves the third inequality in (35).

$\square$

## A.3  Theorem 1 and the Proof

**Notations.** In the following, we provide the corresponding formulations, just like [11, 38]. It should be noted that for convenience in subsequent proofs, $|A| \le a$ indicates that all elements of $A$ are less than $a$, and $|A| \le |B|$ implies that the value of any element $a_{ij}$ at position $(i, j)$ in $A$ is less than the value of the corresponding element $b_{ij}$ in $B$.

For an input $r \in C^{\infty}(\mathbb{R}^2)$, translation $b \in \mathbb{R}^2$ and a degree $\theta \in [0, 2\pi]$, $A_\theta \in O(2)$ is the rotation matrix $\begin{bmatrix} \cos\theta, -\sin\theta \\ \sin\theta, \cos\theta \end{bmatrix}$. $A_\theta$ acts on $r$ by

$$f_{\theta b}^{R}[r](x) = r(A_\theta^{-1}(x - b)), \forall x \in \mathbb{R}^2. \tag{45}$$

For a feature map $e \in C^{\infty}(E(2))$, $E(2) = \mathbb{R}^2 \ltimes O(2)$, and a degree $\theta \in [0, 2\pi]$. $A_\theta$ acts on $e$ by

$$f_{\theta b}^{E}[e](x, A, b) = e(A_\theta^{-1}(x - b), A_\theta^{-1} A), \forall (x, A, b) \in E(2). \tag{46}$$

Considering an multi-channel image $I \in \mathbb{R}^{H \times W \times C}$ as input, which can be naturally represented by a two-dimensional grid function. Suppose the filter is of size $p \times p$, then, the mesh grids for filter and image can be respectively represented as follows:

$$\delta_{kl} = \left( \left( k - \frac{p+1}{2} \right) h, \left( l - \frac{p+1}{2} \right) h \right)^T, \ x_{kl} = \left( \left( k - \frac{H+1}{2} \right) h, \left( l - \frac{W+1}{2} \right) h \right)^T. \tag{47}$$

Then, each channel of $I$ can be obtained by discretizing a smooth function, i.e., for $k = 1, 2, \cdots, W$, $l = 1, 2, \cdots, H$, and $c = 1, 2, \cdots, C$,

$$I_{kl}^c = r_c(x_{kl}), \tag{48}$$

where $r_c$ is latent function for the $c^{th}$ channel.

We denote the equivariant number as $t$ and the correlated rotation group of the equivariant convolution as $S$, respectively. Then, $|S| = t$ and $S = \{A_\theta | \theta = 2\pi k/t, k = 1, 2, \cdots, t\}$. We represent the feature map of equivariant convolution as $Z \in \mathbb{R}^{H \times W \times t \times C}$. $Z$ is a four-dimensional grid function, whose $c^{th}$ channel is sampled from a smooth function $e_c : \mathbb{R}^2 \times S \to \mathbb{R}$, i.e., for $k = 1, 2, \cdots, W$ and $l = 1, 2, \cdots, H$,

$$Z_{kl}^{A,c} = e_c(x_{kl}, A), \tag{49}$$

where $A \in S$.

**Input layer.** The filter of the input multi-channel convolution layer can be represented as

$$\tilde{\Psi}_{kl}^{A,c,d} = \varphi_{cd}\left(A^{-1}\delta_{kl}\right), \tag{50}$$

where $\varphi_{cd}$ is the parameterized filter, $A \in S$, $c = 1, 2, \cdots, n_c$, $d = 1, 2, \cdots, n_d$, $n_c$ and $n_d$ are the input and output channel numbers, respectively. Denoting multi-channel convolution of $\tilde{\Psi}$ and $I$ in the input layer as $Z = \hat{\Psi}(I)$, then it can be calculated by

$$\hat{\Psi}(I)^{A,d} = \sum_c \tilde{\Psi}^{A,c,d} * I^c, \tag{51}$$

where $*$ denotes the 2-D convolution operation. It can be also rewritten in the following more detailed formulation:

$$\hat{e}_d(x_{kl}, A) = \sum_{c,\delta \in \Lambda} \varphi_{cd}\left(A^{-1}\delta\right) r_c(x_{kl} - \delta), \tag{52}$$

where $\Lambda$ is a set of indexes, denoted as $\Lambda = \{\delta_{\hat{k}\hat{l}} | \hat{k}, \hat{l} = 1, 2, \cdots, p\}$, $A \in S$, $k = 1, 2, \cdots, W$ and $l = 1, 2, \cdots, H$.

**Intermediate layer.** The filter of the intermediate multi-channel convolution layer can be represented as

$$\tilde{\Phi}_{kl}^{A,B,c,d} = \varphi_{Acd}\left(B^{-1}\delta_{kl}\right), \tag{53}$$

where $\varphi_{Acd}$ is the parameterized filter, $A, B \in S$, $c = 1, 2, \cdots, n_c$, $d = 1, 2, \cdots, n_d$, $n_c$ and $n_d$ are the input and output channel numbers, respectively. Denoting the multi-channel convolution of $\tilde{\Phi}$ and $Z$ in the intermediate layer as $\hat{Z} = \hat{\Phi}(Z)$, then it can be calculated by

$$\hat{\Phi}(Z)^{B,d} = \sum_{c,A} \tilde{\Phi}^{A,B,c,d} * Z^{A,c}. \tag{54}$$

It can also be rewritten in the following more detailed formulation:

$$\hat{e}_d(x_{kl}, B) = \sum_{c,A,\delta \in \Lambda} \varphi_{Acd}\left(B^{-1}\delta\right) e_c(x_{kl} - \delta, BA). \tag{55}$$

**Output layer.** The filter of the output multi-channel convolution layer can be represented as

$$\tilde{\Upsilon}_{kl}^{B,c,d} = \varphi_{cd}\left(B^{-1}\delta_{kl}\right), \tag{56}$$

where $\varphi_{cd}$ is the parameterized filter, $B \in S$, $c = 1, 2, \cdots, n_c$, $d = 1, 2, \cdots, n_d$, $n_c$ and $n_d$ are the input and output channel numbers, respectively. Denoting the multi-channel convolution of $\tilde{\Upsilon}$ and $Z$ in the output layer as $\hat{Y} = \hat{\Upsilon}(Z)$, then it can be calculated by

$$\hat{\Upsilon}(Z)^d = \sum_{c,B} \tilde{\Upsilon}^{B,c,d} * Z^{B,c}. \tag{57}$$

It can be also rewritten in the following more detailed formulation:

$$\hat{r}_d(x_{kl}) = \sum_{c,B,\delta \in \Lambda} \varphi_{cd}\left(B^{-1}\delta\right) e_c(x_{kl} - \delta, B). \tag{58}$$

The transformations on each channel of the input image and the feature map are defined by

$$\left(\tilde{f}^R_{\theta b}(I)\right)^c_{kl} = f^R_{\theta b}[r_c](x_{kl}), \; \left(\tilde{f}^{\tilde{E}}_{\theta b}(Z)\right)^{A,c}_{kl} = f^E_{\theta b}[e_c](x_{kl}, A, b),$$
$$\forall k = 1, 2, \cdots, H, l = 1, 2, \cdots, W, c = 1, 2, \cdots, C, \forall A \in S, \theta \in [0, 2\pi]. \tag{59}$$

For expression conciseness we further denote

$$\tilde{f}_{\theta b}[x] = \begin{cases} \tilde{f}^R_{\theta b}[x] & \text{if} \quad \forall x \in \mathbb{R}^{H \times W \times C} \\ \tilde{f}^{\tilde{E}}_{\theta b}[x] & \text{if} \quad \forall x \in \mathbb{R}^{H \times W \times t \times C} \end{cases}. \tag{60}$$

Following the [38], for a feature map $Z \in \mathbb{R}^{H \times W \times t \times C}$, we say the channel number of the correlated convolution layer is $tC$, due to the fact that $Z$ is usually reshaped into the shape of $H \times W \times tC$ for implementation convenience, and the flop of the correlated equivariant convolution layer is similar to a $tC$-channel convolution layer.

Then we will prove the Theorem 1. Before this, we first present the following necessary lemmas and the specific proof can be referred to [38].

**Lemma 2.** *For an image $I$ with size $H \times W \times n_0$, and a $N$-layer rotation equivariant CNN network $g(\cdot)$, whose channel number of the $i^{th}$ layer is $n_i$, rotation equivariant subgroup is $S \leqslant O(2)$, $|S| = t$, and activation function is set as ReLU. If the latent continuous function of the $c^{th}$ channel of $I$ denoted as $r_c : \mathbb{R}^2 \to \mathbb{R}$, and the latent continuous function of any convolution filters in the $i^{th}$ layer denoted as $\varphi^i : \mathbb{R}^2 \to \mathbb{R}$, where $i \in \{1, \cdots, N\}$, $c \in \{1, \cdots, n_0\}$, for any $x \in \mathbb{R}^2$, the following conditions are satisfied:*

$$|r_c(x)| \leq F_0, \|\nabla r_c(x)\| \leq G_0, \|\nabla^2 r_c(x)\| \leq H_0,$$
$$|\varphi^i(x)| \leq F_i, \|\nabla \varphi^i(x)\| \leq G_i, \|\nabla^2 \varphi^i(x)\| \leq H_i, \tag{61}$$
$$\forall \|x\| \geq \,^{(p+1)h}/_2, \; \varphi_i(x) = 0,$$

*where $p$ is the filter size, $h$ is the mesh size, and $\nabla$ and $\nabla^2$ denote the operators of gradient and Hessian matrix, respectively. Denote*

$$e^i_d(x, B) = \begin{cases} \sum_{c, \delta \in \Lambda} \varphi^1_{cd}(B^{-1}\delta) r_c(x - \delta) & \text{if} \quad i = 1, \\ \sum_{c, A, \delta \in \Lambda} \varphi^i_{Acd}(B^{-1}\delta) e^{i-1}_c(x - \delta, BA) & \text{if} \quad i \neq 1, N \end{cases} \tag{62}$$

*where $\Lambda = \left\{ \left(\left(k - \frac{p+1}{2}\right)h, \left(l - \frac{p+1}{2}\right)h\right)^T | k, l = 1, 2, \cdots, p \right\}$, $\varphi^1_{cd}$ and $\varphi^i_{Acd}$ are filters in the first layer and other layers respectively. Then, for $\forall B \in S$ the following results are satisfied:*

$$\left|e^i_d(x, B)\right| \leq F_0 \mathcal{F}_i, \tag{63}$$

$$\left|\nabla e^i_d(x, B)\right| \leq \left(\sum_{m=1}^{i} \frac{G_m F_0}{F_m} + G_0\right) \mathcal{F}_i, \tag{64}$$

$$\left|\nabla^2 e^i_d(x, B)\right| \leq \left(\sum_{m=1}^{i} \frac{H_m F_0}{F_m} + 2\sum_{l=1}^{i} \frac{G_l}{F_l} \sum_{m=1}^{l-1} \frac{G_m F_0}{F_m} + 2\sum_{m=1}^{i} \frac{G_m G_0}{F_m} + H_0\right) \mathcal{F}_i, \tag{65}$$

*where $\mathcal{F}_i = \prod_{k=1}^{i} n_{k-1} p^2 F_k, \forall i = 1, 2, \cdots, N - 1$.*

**Lemma 3.** *For an image $I$ with size $H \times W \times n_0$, and a $N$-layer rotation equivariant CNN network $g(\cdot)$, whose channel number of the $i^{th}$ layer is $n_i$, rotation equivariant subgroup is $S \leqslant O(2)$, $|S| = t$, and activation function is set as ReLU. If the latent continuous function of the $c^{th}$ channel of $I$ denoted as $r_c : \mathbb{R}^2 \to \mathbb{R}$, and the latent continuous function of any convolution filters in the $i^{th}$ layer denoted as $\varphi^i : \mathbb{R}^2 \to \mathbb{R}$, where $i \in \{1, \cdots, N\}$, $c \in \{1, \cdots, n_0\}$, for any $x \in \mathbb{R}^2$, the following conditions are satisfied:*

$$|r_c(x)| \leq F_0, \|\nabla r_c(x)\| \leq G_0, \|\nabla^2 r_c(x)\| \leq H_0,$$
$$|\varphi^i(x)| \leq F_i, \|\nabla \varphi^i(x)\| \leq G_i, \|\nabla^2 \varphi^i(x)\| \leq H_i, \tag{66}$$
$$\forall \|x\| \geq \,^{(p+1)h}/_2, \; \varphi_i(x) = 0,$$

where $p$ is the filter size, $h$ is the mesh size, $\nabla$ and $\nabla^2$ denote the operators of gradient and Hessian matrix, respectively. For an arbitrary $\theta \in [0, 2\pi]$, $A_\theta$ denotes the rotation matrix. If $F(\theta) = \mathrm{g}\left[\tilde{f}_{\theta b}\right](I) = \hat{\Upsilon}\left[\hat{\Phi}_{N-1} \cdots \hat{\Phi}_{i+1}\left[\hat{\Phi}_i \cdots \hat{\Phi}_2\left[\hat{\Psi}\left[\tilde{f}_{\theta b}\right]\right]\right]\cdots\right](I)$, then the following result is satisfied:

$$|F'(\theta)| \leq \mathcal{F}\left(\max\{H, W\} + N\left(p+1\right)\right)hG_0, \tag{67}$$

where $\mathcal{F} = \prod\limits_{k=1}^{N} n_{k-1}p^2 F_k$.

**Lemma 4.** *Under the same conditions with lemma 3,*

*If* $F(\theta) = \tilde{f}_{\theta b}\left[\mathrm{g}\right](I) = \tilde{f}_{\theta b}\left[\hat{\Upsilon}\left[\hat{\Phi}_{N-1}\cdots\hat{\Phi}_{i+1}\left[\hat{\Phi}_i\cdots\hat{\Phi}_2\left[\hat{\Psi}\right]\cdots\right]\right]\right](I)$, *and then the following result is satisfied:*

$$|F'(\theta)| \leq \mathcal{F}\max\{H, W\}hG_0, \tag{68}$$

where $\mathcal{F} = \prod\limits_{k=1}^{N} n_{k-1}p^2 F_k$.

Then, let us give Theorem 1 and prove it based on the aforementioned Lemmas.

**Theorem 1.** *For an image $I$ with size $H \times W \times n_0$, and a $N$-layer rotation-translation equivariant CNN network $\mathrm{g}(\cdot)$, whose channel number of the $i^{th}$ layer is $n_i$, rotation equivariant subgroup is $S \leqslant O(2)$, $|S| = t$, and activation function is set as ReLU. If the latent continuous function of the $c^{th}$ channel of $I$ denoted as $r_c : \mathbb{R}^2 \to \mathbb{R}$, and the latent continuous function of any convolution filters in the $i^{th}$ layer denoted as $\varphi^i : \mathbb{R}^2 \to \mathbb{R}$, where $i \in \{1, \cdots, N\}$, $c \in \{1, \cdots, n_0\}$, for any $x \in \mathbb{R}^2$, the following conditions are satisfied:*

$$\begin{aligned}
&|r_c(x)| \leq F_0, \|\nabla r_c(x)\| \leq G_0, \|\nabla^2 r_c(x)\| \leq H_0, \\
&|\varphi^i(x)| \leq F_i, \|\nabla\varphi^i(x)\| \leq G_i, \|\nabla^2\varphi^i(x)\| \leq H_i, \\
&\forall\|x\| \geq {(p+1)h}/{2}, \ \varphi_i(x) = 0,
\end{aligned} \tag{69}$$

*where $p$ is the filter size, $h$ is the mesh size, $\nabla$ and $\nabla^2$ denote the operators of gradient and Hessian matrix, respectively. For an arbitrary $0 \leq \theta \leq 2\pi$, $A_\theta \in S$ denotes the rotation matrix, $b \in \mathbb{R}^2$ denotes the translation, and the following result is satisfied:*

$$\left\|\tilde{f}_{\theta b}^{-1}\,\mathrm{g}\left[\tilde{f}_{\theta b}\right](I) - [\mathrm{g}](I)\right\|_\infty \leq C_1 h^2 + C_2 pht^{-1}, \tag{70}$$

*where $\tilde{f}_{\theta b}$ is defined in Eq. (60) and*

$$\begin{aligned}
C_1 &= 2N\mathcal{F}\cdot\sum_{i=1}^{N}\left(\frac{H_i F_0}{F_i} + 2\frac{G_i}{F_i}\sum_{m=1}^{i-1}\frac{G_m F_0}{F_m} + 2\frac{G_i G_0}{F_i} + H_0\right), \\
C_2 &= 2\pi G_0\mathcal{F}\left(2\max\{H, W\}p^{-1} + 2N\right), \mathcal{F} = \prod\nolimits_{i=1}^{N} n_{i-1}p^2 F_i.
\end{aligned} \tag{71}$$

*Proof.* Let $\hat{I} = \tilde{f}_{\theta b}I$, we can split the left part of Eq. (70) as

$$\begin{aligned}
&\left|\tilde{f}_{\theta b}^{-1}\,\mathrm{g}\left[\tilde{f}_{\theta b}\right](I) - [\mathrm{g}](I)\right| \\
&= \left|\tilde{f}_{\theta b}^{-1}\,\mathrm{g}(\hat{I}) - \mathrm{g}\left[\tilde{f}_{\theta b}^{-1}\right](\hat{I})\right| \\
&\leq \underbrace{\left|\mathrm{g}\left[\tilde{f}_{\theta b}^{-1}\right](\hat{I}) - \mathrm{g}\left[\tilde{f}_{\theta_k b}^{-1}\right](\hat{I})\right|}_{\langle 1 \rangle} + \underbrace{\left|\mathrm{g}\left[\tilde{f}_{\theta_k b}^{-1}\right](\hat{I}) - \tilde{f}_{\theta_k b}^{-1}\left[\mathrm{g}\right](\hat{I})\right|}_{\langle 2 \rangle} + \underbrace{\left|\tilde{f}_{\theta_k b}^{-1}\left[\mathrm{g}\right](\hat{I}) - \tilde{f}_{\theta b}^{-1}\left[\mathrm{g}\right](\hat{I})\right|}_{\langle 3 \rangle},
\end{aligned} \tag{72}$$

where $\theta = \theta_k + \delta$, where $k = 1, 2, \cdots, t$, $0 \leq \delta \leq {2\pi}/{t}$. Next, we need to estimate the error bounds of the above three items, separately. It should be noted that the following proof is deduced without the ReLU activation function for concise. However, the conclusions are all still correct for networks with the ReLU activation function, since ReLU does not disturb the equivariance or amplify the error bound.

Firstly, we prove the following inequality for the part $\langle 1 \rangle$ of Eq. (72).

$$\left| \mathrm{g}\left[ \tilde{f}_{\theta b}^{-1} \right](\hat{I}) - \mathrm{g}\left[ \tilde{f}_{\theta_k b}^{-1} \right](\hat{I}) \right| \le \frac{2\pi}{t} \mathcal{F}\left( \max\{H, W\} + N\left(p+1\right) \right) hG_0. \tag{73}$$

Let us denote $F_1(\theta) = \mathrm{g}\left[ \tilde{f}_{\theta b} \right](\hat{I})$. Obviously the function $F_1(\theta)$ is continuous with respect to $\theta$, so we have the following conclusion by the Lagrange Mean Value Theorem [39]

$$\begin{aligned}
\left| \mathrm{g}\left[ \tilde{f}_{\theta b} \right](\hat{I}) - \mathrm{g}\left[ \tilde{f}_{\theta_k b} \right](\hat{I}) \right| &= |F_1(\theta) - F_1(\theta_k)| \\
&\le |F_1'(\xi_1)| \, \delta \\
&\le \frac{2\pi}{t} |F_1'(\xi_1)|,
\end{aligned} \tag{74}$$

where $0 < \xi_1 < \delta$ and by lemma 3 we have $|F_1'(\xi_1)| \le \mathcal{F}\left( \max\{H, W\} + N\left(p+1\right) \right) hG_0$. Then we can prove Eq. (73).

Secondly, we prove the following inequality for the part $\langle 3 \rangle$ of Eq. (72).

$$\left| \tilde{f}_{\theta_k b} \left[ \mathrm{g} \right](\hat{I}) - \tilde{f}_{\theta b} \left[ \mathrm{g} \right](\hat{I}) \right| \le \frac{2\pi}{t} \mathcal{F} \max\{H, W\} hG_0. \tag{75}$$

Let us denote $F_2(\theta) = \tilde{f}_{\theta b} \left[ \mathrm{g} \right](\hat{I})$. Obviously the function $F_2(\theta)$ is continuous with respect to $\theta$, so we have the following conclusion by the Lagrange Mean Value Theorem [39]

$$\begin{aligned}
\left| \tilde{f}_{\theta_k b} \left[ \mathrm{g} \right](\hat{I}) - \tilde{f}_{\theta b} \left[ \mathrm{g} \right](\hat{I}) \right| &= |F_2(\theta) - F_2(\theta_k)| \\
&\le |F_2'(\xi_2)| \, \delta \\
&\le \frac{2\pi}{t} |F_2'(\xi_2)|,
\end{aligned} \tag{76}$$

where $0 < \xi_2 < \delta$ and by lemma 4 we have $|F_2'(\xi_2)| \le \mathcal{F} \max\{H, W\} hG_0$. Then we can easily achieve Eq. (75).

Thirdly, we now prove the following inequality:

$$\left| \mathrm{g}\left[ \tilde{f}_{\theta_k b}^{-1} \right](\hat{I}) - \tilde{f}_{\theta_k b}^{-1} \left[ \mathrm{g} \right](\hat{I}) \right| \le 2\mathcal{F} \sum_{i=1}^{N} \left( \sum_{m=1}^{i} \frac{H_m F_0}{F_m} + 2 \sum_{l=1}^{i} \frac{G_l}{F_l} \sum_{m=1}^{l-1} \frac{G_m F_0}{F_m} + 2 \sum_{m=1}^{i} \frac{G_m G_0}{F_m} + H_0 \right) h^2. \tag{77}$$

$\mathrm{g}(\cdot)$, an N-layer rotation equivariant CNN network, usually includes 1 input layer, $N-2$ intermediate layers, and 1 output layer. We can formally define it as :

$$\mathrm{g}(\cdot) = \hat{\Upsilon}\left[ \hat{\Phi}_{N-1} \cdots \hat{\Phi}_{i+1} \left[ \hat{\Phi}_i \cdots \hat{\Phi}_2 \left[ \hat{\Psi} \right] \cdots \right] \right](\cdot). \tag{78}$$

Then we have

$$\begin{aligned}
&\left| \mathrm{g}\left[ \tilde{f}_{\theta_k b}^{-1} \right](\hat{I}) - \tilde{f}_{\theta_k b}^{-1} \left[ \mathrm{g} \right](\hat{I}) \right| \\
&= \left| \hat{\Upsilon}\left[ \hat{\Phi}_{N-1} \cdots \hat{\Phi}_{i+1}\left[ \hat{\Phi}_i \cdots \hat{\Phi}_2\left[ \hat{\Psi}\left[ \tilde{f}_{\theta_k b}^{-1} \right] \right] \cdots \right] \right](\hat{I}) - \tilde{f}_{\theta_k b}^{-1}\left[ \hat{\Upsilon}\left[ \hat{\Phi}_{N-1} \cdots \hat{\Phi}_{i+1}\left[ \hat{\Phi}_i \cdots \hat{\Phi}_2\left[ \hat{\Psi} \right] \cdots \right] \right] \right](\hat{I}) \right| \\
&\le \left| \hat{\Upsilon}\left[ \hat{\Phi}_{N-1} \cdots \hat{\Phi}_{i+1}\left[ \hat{\Phi}_i \cdots \hat{\Phi}_2\left[ \hat{\Psi}\left[ \tilde{f}_{\theta_k b}^{-1} \right] \right] \cdots \right] \right](\hat{I}) - \hat{\Upsilon}\left[ \hat{\Phi}_{N-1} \cdots \hat{\Phi}_{i+1}\left[ \hat{\Phi}_i \cdots \hat{\Phi}_2\left[ \tilde{f}_{\theta_k b}^{-1}\left[ \hat{\Psi} \right] \right] \cdots \right] \right](\hat{I}) \right| \\
&\quad + \left| \hat{\Upsilon}\left[ \hat{\Phi}_{N-1} \cdots \hat{\Phi}_{i+1}\left[ \hat{\Phi}_i \cdots \hat{\Phi}_2\left[ \tilde{f}_{\theta_k b}^{-1}\left[ \hat{\Psi} \right] \right] \cdots \right] \right](\hat{I}) - \hat{\Upsilon}\left[ \hat{\Phi}_{N-1} \cdots \hat{\Phi}_{i+1}\left[ \hat{\Phi}_i \cdots \tilde{f}_{\theta_k b}^{-1}\left[ \hat{\Phi}_2\left[ \hat{\Psi} \right] \right] \cdots \right] \right](\hat{I}) \right| \\
&\qquad \cdots \\
&\quad + \left| \hat{\Upsilon}\left[ \hat{\Phi}_{N-1}\left[ \tilde{f}_{\theta_k b}^{-1} \cdots \hat{\Phi}_{i+1}\left[ \hat{\Phi}_i \cdots \hat{\Phi}_2\left[ \hat{\Psi} \right] \right] \cdots \right] \right](\hat{I}) - \hat{\Upsilon}\left[ \tilde{f}_{\theta_k b}^{-1}\left[ \hat{\Phi}_{N-1} \cdots \hat{\Phi}_{i+1}\left[ \hat{\Phi}_i \cdots \hat{\Phi}_2\left[ \hat{\Psi} \right] \right] \cdots \right] \right](\hat{I}) \right| \\
&\quad + \left| \hat{\Upsilon}\left[ \tilde{f}_{\theta_k b}^{-1}\left[ \hat{\Phi}_{N-1} \cdots \hat{\Phi}_{i+1}\left[ \hat{\Phi}_i \cdots \hat{\Phi}_2\left[ \hat{\Psi} \right] \right] \cdots \right] \right](\hat{I}) - \tilde{f}_{\theta_k b}^{-1}\left[ \hat{\Upsilon}\left[ \hat{\Phi}_{N-1} \cdots \hat{\Phi}_{i+1}\left[ \hat{\Phi}_i \cdots \hat{\Phi}_2\left[ \hat{\Psi} \right] \right] \cdots \right] \right](\hat{I}) \right|.
\end{aligned} \tag{79}$$

604 We denote $\delta_1$, $\delta_i$ $(i = 2, 3, \cdots, N-1)$, and $\delta_N$ as the filter indexes of the input layer, $i^{th}$ intermediate
605 layer and output layer, respectively. The input channel number of the $i^{th}$ layer is set as $c_{i-1} = n_{i-1}$.

606 1) For the input layer, with Eqs. (52), (55) and (58), let $x$ denote the coordinate of position $(k, l)$,
607 then we have

$$
\left| \left( \hat{\Upsilon}\Big[ \hat{\Phi}_{N-1} \cdots \hat{\Phi}_{i+1}\Big[ \hat{\Phi}_i \cdots \hat{\Phi}_2\Big[ \hat{\Psi}\Big[ \tilde{f}_{\theta_k b}^{-1} \Big] \Big] \cdots \Big] \Big](\hat{I}) - \hat{\Upsilon}\Big[ \hat{\Phi}_{N-1} \cdots \hat{\Phi}_{i+1}\Big[ \hat{\Phi}_i \cdots \hat{\Phi}_2\Big[ \tilde{f}_{\theta_k b}^{-1}\big[ \hat{\Psi} \big] \Big] \cdots \Big] \Big](\hat{I}) \right)_{ij}^{c_N} \right|
$$

$$
= \left| \sum_{\substack{c_{N-1} \\ B_{N-1} \in S \\ \delta_N \in \Lambda}} \cdots \sum_{\substack{c_1 \\ A \in S \\ \delta_2 \in \Lambda}} \sum_{\substack{c_0 \\ \delta_1 \in \Lambda}} \varphi_{c_{N-1} c_N}^N (B_{N-1}^{-1}\delta_N) \cdots \varphi_{c_0 c_1}^1 (A^{-1}\delta_1)\, r_{c_0}\, (A_{\theta_k}(x - \delta_N - \cdots - \delta_2 - \delta_1 + b)) \right.
$$

$$
\left. - \sum_{\substack{c_{N-1} \\ B_{N-1} \in S \\ \delta_N \in \Lambda}} \cdots \sum_{\substack{c_1 \\ A \in S \\ \delta_2 \in \Lambda}} \sum_{\substack{c_0 \\ \delta_1 \in \Lambda}} \varphi_{c_{N-1} c_N}^N (B_{N-1}^{-1}\delta_N) \cdots \varphi_{c_0 c_1}^1 (A^{-1}A_{\theta_k}^{-1}\delta_1)\, r_{c_0}(A_{\theta_k}(x - \delta_N - \cdots - \delta_2 + b) - \delta_1) \right|
$$

$$
= \left| \sum_{\substack{c_{N-1} \\ B_{N-1} \in S \\ \delta_N \in \Lambda}} \cdots \sum_{\substack{c_1 \\ A \in S \\ \delta_2 \in \Lambda}} \varphi_{c_{N-1} c_N}^N (B_{N-1}^{-1}\delta_N) \cdots \varphi_{A c_1 c_2}^2 (B_2^{-1}\delta_2) \right.
$$

$$
\left. \left( \sum_{\substack{c_0 \\ \delta_1 \in \Lambda}} \varphi_{c_0 c_1}^1 (A^{-1}\delta_1) r(A_{\theta_k}(x - \delta_N - \cdots \delta_1 + b)) - \sum_{\substack{c_0 \\ \delta_1 \in \Lambda}} \varphi_{c_0 c_1}^1 (A^{-1}A_{\theta_k}^{-1}\delta_1)\, r_{c_0}(A_{\theta_k}(x - \delta_N - \cdots \delta_2 + b) - \delta_1) \right) \right|
$$

$$
\le \sum_{\substack{c_{N-1} \\ B_{N-1} \in S \\ \delta_N \in \Lambda}} \cdots \sum_{\substack{c_1 \\ A \in S \\ \delta_2 \in \Lambda}} \left| \varphi_{c_{N-1} c_N}^N (B_{N-1}^{-1}\delta_N) \right| \cdots \left| \varphi_{A c_1 c_2}^2 (B_2^{-1}\delta_2) \right|
$$

$$
\left| \sum_{\substack{c_0 \\ \delta_1 \in \Lambda}} \varphi_{c_0 c_1}^1 (A^{-1}\delta_1) r(A_{\theta_k}(x - \delta_N - \cdots - \delta_1 + b)) - \sum_{\substack{c_0 \\ \delta_1 \in \Lambda}} \varphi_{c_0 c_1}^1 (A^{-1}A_{\theta_k}^{-1}\delta_1)\, r_{c_0}(A_{\theta_k}(x - \delta_N - \cdots - \delta_2 + b) - \delta_1) \right|
$$

$$
\le \sum_{\substack{c_{N-1} \\ B_{N-1} \in S \\ \delta_N \in \Lambda}} \cdots \sum_{\substack{c_1 \\ A \in S \\ \delta_2 \in \Lambda}} \sum_{c_0} F_N \cdots F_2 \left| \sum_{\delta_1 \in \Lambda} \varphi_{c_0 c_1}^1 (A^{-1}\delta_1)\, r\, (A_{\theta_k}(x - \delta_N \cdots - \delta_1 + b)) \right.
$$

$$
\left. - \sum_{\delta_1 \in \Lambda} \varphi_{c_0 c_1}^1 (A^{-1}A_{\theta_k}^{-1}\delta_1)\, r_{c_0}\, (A_{\theta_k}(x - \delta_N - \cdots - \delta_2 + b) - \delta_1) \right|.
$$

$$
\tag{80}
$$

Let us denote $\hat{x} = x - \delta_N - \delta_{N-1} - \cdots - \delta_2 + b$. Utilizing Eq. (35) from Remark 2 for the input layer, we can deduce the following result:

$$
\left| \sum_{\delta_1 \in \Lambda} \varphi^1_{c_0 c_1} \left( A^{-1}\delta_1 \right) r \left( A_{\theta_k}(x - \delta_N \cdots - \delta_1 + b) \right) \right.
$$

$$
\left. - \sum_{\delta_1 \in \Lambda} \varphi^1_{c_0 c_1} \left( A^{-1} A^{-1}_{\theta_k}\delta_1 \right) r_{c_0} \left( A_{\theta_k}(x - \delta_N - \cdots - \delta_2 + b) - \delta_1 \right) \right|
$$

$$
= \left| \sum_{\delta_1 \in \Lambda} \varphi^1_{c_0 c_1} \left( A^{-1}\delta_1 \right) r \left( A_{\theta_k}(\hat{x} - \delta_1) \right) \right.
$$

$$
\left. - \sum_{\delta_1 \in \Lambda} \varphi^1_{c_0 c_1} \left( A^{-1} A^{-1}_{\theta_k}\delta_1 \right) r_{c_0} \left( A_{\theta_k}\hat{x} - \delta_1 \right) \right|
$$

$$
\leq n_0 \frac{C_1}{2}(p + 1)^2 h^2,
\tag{81}
$$

where we do not specifically indicate the numbers of input and output channels, i.e. $\varphi^1(x) = \varphi^1_{c_0 c_1}(x)$, and we have $C_1 = H_1 F_0 + F_1 H_0 + 2 G_1 G_0$.

Therefore, according to Eqs. (80) and (81), we have

$$
\left| \left( \hat{\Upsilon} \left[ \hat{\Phi}_{N-1} \cdots \hat{\Phi}_{i+1} \left[ \hat{\Phi}_i \cdots \hat{\Phi}_2 \left[ \hat{\Psi} \left[ \tilde{f}^{-1}_{\theta_k b} \right] \right] \cdots \right] \right] \right)(\hat{I}) \right.
$$

$$
\left. - \hat{\Upsilon} \left[ \hat{\Phi}_{N-1} \cdots \hat{\Phi}_{i+1} \left[ \hat{\Phi}_i \cdots \hat{\Phi}_2 \left[ \tilde{f}^{-1}_{\theta_k b} \left[ \hat{\Psi} \right] \right] \cdots \right] \right](\hat{I}) \right)^{c_N}_{ij} \right|
$$

$$
\leq n_{N-1} p^2 F_N n_{N-2} p^2 F_{N-1} \cdots n_1 p^2 F_2 n_0 \frac{C_1}{2}(p + 1)^2 h^2
$$

$$
\leq \left( \prod_{k=2}^{N} n_{k-1} p^2 F_k \right) n_0 \frac{(p + 1)^2 h^2}{2} (H_1 F_0 + F_1 H_0 + 2 G_1 G_0)
$$

$$
\leq 2\mathcal{F} \left( \frac{H_1}{F_1} F_0 + H_0 + 2 \frac{G_1}{F_1} G_0 \right) h^2.
\tag{82}
$$

2) For the any $i^{th}$ intermediate layer, $1 < i < N$. We have:

$$
\left| \left( \hat{\Upsilon} \left[ \hat{\Phi}_{N-1} \cdots \hat{\Phi}_i \left[ \tilde{f}^{-1}_{\theta_k b} \left[ \hat{\Phi}_{i-1} \cdots \hat{\Phi}_2 \left[ \hat{\Psi} \right] \cdots \right] \right] \right] \right)(\hat{I}) \right.
$$

$$
\left. - \hat{\Upsilon} \left[ \hat{\Phi}_{N-1} \cdots \hat{\Phi}_{i+1} \left[ \tilde{f}^{-1}_{\theta_k b} \left[ \hat{\Phi}_i \cdots \hat{\Phi}_2 \left[ \hat{\Psi} \right] \cdots \right] \right] \right](\hat{I}) \right)^{c_N}_{ij} \right|
$$

$$
= \left| \sum_{\substack{c_{N-1} \\ B_{N-1} \in S \\ \delta_N \in \Lambda}} \cdots \sum_{\substack{c_{i-1} \\ B_{i-1} \in S \\ \delta_i \in \Lambda}} \varphi^N_{c_{N-1} c_N} (B^{-1}_{N-1}\delta_N) \cdots \varphi^i_{B_{i-1} c_{i-1} c_i} (B^{-1}_i \delta_i) \right.
$$

$$
e^{i-1}_{c_{i-1}} \left( A_{\theta_k}(x - \delta_N - \cdots - \delta_i + b), A_{\theta_k} B_{i-1} \right)
$$

$$
- \sum_{\substack{c_{N-1} \\ B_{N-1} \in S \\ \delta_N \in \Lambda}} \cdots \sum_{\substack{c_{i-1} \\ B_{i-1} \in S \\ \delta_i \in \Lambda}} \varphi^N_{c_{N-1} c_N} (B^{-1}_{N-1}\delta_N) \cdots \varphi^i_{B_{i-1} c_{i-1} c_i} (B^{-1}_i A^{-1}_{\theta_k}\delta_i)
$$

$$
\left. e^{i-1}_{c_{i-1}} \left( A_{\theta_k}(x - \delta_N - \cdots - \delta_{i+1} + b) - \delta_i, B_{i-1} \right) \right|
\tag{83}
$$

$$\leq \left| \sum_{\substack{c_{N-1} \\ B_{N-1} \in S \\ \delta_N \in \Lambda}} \cdots \sum_{\substack{c_i \\ B_i \in S \\ \delta_{i+1} \in \Lambda}} \varphi^N_{c_{N-1}c_N}(B_{N-1}^{-1}\delta_N)\varphi^{N-1}_{B_{N-2}c_{N-2}c_{N-1}}(B_{N-1}^{-1}\delta_{N-1})\cdots\varphi^{i+1}_{B_i c_i c_{i+1}}(B_{i+1}^{-1}\delta_{i+1}) \right.$$

$$\left( \left( \sum_{\substack{c_{i-1} \\ B_{i-1} \in S \\ \delta_i \in \Lambda}} \varphi^i_{B_{i-1}c_{i-1}c_i}(B_i^{-1}\delta_i)e^{i-1}_{c_{i-1}}(A_{\theta_k}(x-\delta_N-\cdots-\delta_{i+1}-\delta_i+b), A_{\theta_k}B_{i-1}) \right. \right.$$

$$\left. \left. \left. - \sum_{\substack{c_{i-1} \\ B_{i-1} \in S \\ \delta_i \in \Lambda}} \varphi^i_{B_{i-1}c_{i-1}c_i}(B_i^{-1}A_{\theta_k}^{-1}\delta_i)e^{i-1}_{c_{i-1}}(A_{\theta_k}(x-\delta_N-\cdots-\delta_{i+1}+b)-\delta_i, B_{i-1}) \right) \right) \right|$$

$$\leq \sum_{\substack{c_{N-1} \\ B_{N-1} \in S \\ \delta_N \in \Lambda}} \sum_{\substack{c_{N-2} \\ B_{N-2} \in S \\ \delta_{N-1} \in \Lambda}} \cdots \sum_{\substack{c_i \\ B_i \in S \\ \delta_{i+1} \in \Lambda}} F_N F_{N-1} \cdots F_{i+1}$$

$$\left| \left( \left( \sum_{\substack{c_{i-1} \\ B_{i-1} \in S \\ \delta_i \in \Lambda}} \varphi^i_{B_{i-1}c_{i-1}c_i}(B_i^{-1}\delta_i)e^{i-1}_{c_{i-1}}(A_{\theta_k}(x-\delta_N-\cdots-\delta_{i+1}-\delta_i+b), A_{\theta_k}B_{i-1}) \right. \right. \right.$$

$$\left. \left. \left. - \sum_{\substack{c_{i-1} \\ B_{i-1} \in S \\ \delta_i \in \Lambda}} \varphi^i_{B_{i-1}c_{i-1}c_i}(B_i^{-1}A_{\theta_k}^{-1}\delta_i)e^{i-1}_{c_{i-1}}(A_{\theta_k}(x-\delta_N-\cdots-\delta_{i+1}+b)-\delta_i, B_{i-1}) \right) \right) \right|.$$

615  Let us denote $\hat{x} = x - \delta_N - \cdots - \delta_{i+1} + b$. Then, by Eq. (35) in Remark 2, we have:

$$\left| \sum_{\substack{c_{i-1} \\ B_{i-1} \in S \\ \delta_i \in \Lambda}} \varphi^i_{B_{i-1}c_{i-1}c_i}(B_i^{-1}\delta_i)e^{i-1}_{c_{i-1}}\left(A_{\theta_k}(\hat{x}-\delta_i), A_{\theta_k}B_{i-1}\right) \right.$$

$$\left. - \sum_{\substack{c_{i-1} \\ B_{i-1} \in S \\ \delta_i \in \Lambda}} \varphi^i_{B_{i-1}c_{i-1}c_i}(B_i^{-1}A_{\theta_k}^{-1}\delta_i)e^{i-1}_{c_{i-1}}\left(A_{\theta_k}\hat{x}-\delta_i, B_{i-1}\right) \right|$$

$$= \sum_{c_{i-1}} \left| \sum_{\substack{B_{i-1} \in S \\ \delta_i \in \Lambda}} \varphi^i_{B_{i-1}c_{i-1}c_i}(B_i^{-1}\delta_i)e^{i-1}_{c_{i-1}}\left(A_{\theta_k}(\hat{x}-\delta_i), A_{\theta_k}B_{i-1}\right) \right.$$

$$\left. - \sum_{\substack{B_{i-1} \in S \\ \delta_i \in \Lambda}} \varphi^i_{B_{i-1}c_{i-1}c_i}(B_i^{-1}A_{\theta_k}^{-1}\delta_i)e^{i-1}_{c_{i-1}}\left(A_{\theta_k}\hat{x}-\delta_i, B_{i-1}\right) \right|$$

$$\leq n_{i-1}\frac{C_i}{2}(p+1)^2 h^2,$$

$$(84)$$

where we do not specifically indicate the numbers of input and output channels, i.e. $\varphi^i(x) = \varphi^i_{A c_{i-1} c_i}(x)$, and we have

$$C_i = \sup \left( \left\| \nabla^2 \varphi^i(x) \right\| \left| e^{i-1}_{c_{i-1}}(x, B) \right| + \left| \varphi^i(x) \right| \left\| \nabla^2 e^{i-1}_{c_{i-1}}(x, B) \right\| + 2 \left\| \nabla \varphi^i(x) \right\| \left\| \nabla e^{i-1}_{c_{i-1}}(x, B) \right\| \right).$$

Therefore, according to Eqs. (83) and (84), we have

$$\left| \left( \hat{\Upsilon} \left[ \hat{\Phi}_{N-1} \cdots \hat{\Phi}_i \left[ \tilde{f}^{-1}_{\theta_k b} \left[ \hat{\Phi}_{i-1} \cdots \hat{\Phi}_2 \left[ \hat{\Psi} \right] \cdots \right] \right] \right] \right) (\hat{I}) \right.$$

$$\left. - \hat{\Upsilon} \left[ \hat{\Phi}_{N-1} \cdots \hat{\Phi}_{i+1} \left[ \tilde{f}^{-1}_{\theta_k b} \left[ \hat{\Phi}_i \cdots \hat{\Phi}_2 \left[ \hat{\Psi} \right] \cdots \right] \right] \right] (\hat{I}) \right)^{c_N}_{ij} \right|$$

$$\leq n_{N-1} p^2 F_N \cdots n_i p^2 F_{i+1} n_{i-1} \frac{C_i}{2} (p+1)^2 h^2$$

$$= \left( \prod_{k=i+1}^{N} n_{k-1} p^2 F_k \right) n_{i-1} \frac{C_i}{2} (p+1)^2 h^2.$$

(85)

Substituting Eqs. (63), (64) and (65) into Eq. (85), denoting $\mathcal{F}_{i-1} = \prod_{k=1}^{i-1} n_{k-1} p^2 F_k$, then we have

$$\left| \left( \hat{\Upsilon} \left[ \hat{\Phi}_{N-1} \cdots \hat{\Phi}_i \left[ \tilde{f}^{-1}_{\theta_k b} \left[ \hat{\Phi}_{i-1} \cdots \hat{\Phi}_2 \left[ \hat{\Psi} \right] \cdots \right] \right] \right] (\hat{I}) - \hat{\Upsilon} \left[ \hat{\Phi}_{N-1} \cdots \hat{\Phi}_{i+1} \left[ \tilde{f}^{-1}_{\theta_k b} \left[ \hat{\Phi}_i \cdots \hat{\Phi}_2 \left[ \hat{\Psi} \right] \cdots \right] \right] \right] (\hat{I}) \right)^{c_N}_{ij} \right|$$

$$\leq \left( \prod_{k=i+1}^{N} n_{k-1} p^2 F_k \right) \frac{n_{i-1}(p+1)^2 h^2}{2}$$

$$\left( \left\| \nabla^2 \varphi^i(x) \right\| \left| e^{i-1}_{c_{i-1}}(x, B) \right| + \left| \varphi^i(x) \right| \left\| \nabla^2 e^{i-1}_{c_{i-1}}(x, B) \right\| + 2 \left\| \nabla \varphi^i(x) \right\| \left\| \nabla e^{i-1}_{c_{i-1}}(x, B) \right\| \right)$$

$$\leq 2 \left( \prod_{k=i}^{N} n_{k-1} p^2 F_k \right) \left( \frac{H_i}{F_i} \left| e^{i-1}_{c_{i-1}}(x, B) \right| + \left\| \nabla^2 e^{i-1}_{c_{i-1}}(x, B) \right\| + 2 \frac{G_i}{F_i} \left\| \nabla e^{i-1}_{c_{i-1}}(x, B) \right\| \right) h^2$$

$$\leq 2 \left( \prod_{k=i}^{N} n_{k-1} p^2 F_k \right) \mathcal{F}_{i-1}$$

$$\left( \frac{H_i F_0}{F_i} + \sum_{m=1}^{i-1} \frac{H_m F_0}{F_m} + 2 \sum_{l=1}^{i-1} \frac{G_l}{F_l} \sum_{m=1}^{l-1} \frac{G_m F_0}{F_m} + 2 \sum_{m=1}^{i-1} \frac{G_m G_0}{F_m} + H_0 + \sum_{m=1}^{i-1} 2 \frac{G_i G_m F_0}{F_i F_m} + 2 \frac{G_i G_0}{F_i} \right) h^2$$

$$\leq 2 \mathcal{F} \left( \sum_{m=1}^{i} \frac{H_m F_0}{F_m} + 2 \sum_{l=1}^{i} \frac{G_l}{F_l} \sum_{m=1}^{l-1} \frac{G_m F_0}{F_m} + 2 \sum_{m=1}^{i} \frac{G_m G_0}{F_m} + H_0 \right) h^2.$$

(86)

3) For the output layer, with Eqs. (52), (55) and (58) we have

$$\left| \left( \hat{\Upsilon} \left[ \tilde{f}^{-1}_{\theta_k b} \left[ \hat{\Phi}_{N-1} \cdots \hat{\Phi}_{i+1} \left[ \hat{\Phi}_i \cdots \hat{\Phi}_2 \left[ \hat{\Psi} \right] \cdots \right] \right] \right] (\hat{I}) - \tilde{f}^{-1}_{\theta_k b} \left[ \hat{\Upsilon} \left[ \hat{\Phi}_{N-1} \cdots \hat{\Phi}_{i+1} \left[ \hat{\Phi}_i \cdots \hat{\Phi}_2 \left[ \hat{\Psi} \right] \cdots \right] \right] \right] (\hat{I}) \right)^{c_N}_{ij} \right|$$

$$= \left| \sum_{\substack{c_{N-1} \\ B_{N-1} \in S \\ \delta_N \in \Lambda}} \varphi^N_{c_{N-1} c_N} (B^{-1}_{N-1} \delta_N) e^{N-1}_{c_{N-1}} \left( A_{\theta_k} (x - \delta_N + b), A_{\theta_k} B_{N-1} \right) \right.$$

$$\left. - \sum_{\substack{c_{N-1} \\ B_{N-1} \in S \\ \delta_N \in \Lambda}} \varphi^N_{c_{N-1} c_N} (B^{-1}_{N-1} A^{-1}_{\theta_k} \delta_N) e^{N-1}_{c_{N-1}} \left( A_{\theta_k} (x + b) - \delta_N, B_{N-1} \right) \right|.$$

Then, by Eq. (35) from Remark 2 for the output, we have:

$$
\left|\left(\hat{\Upsilon}\Big[\tilde{f}_{\theta_k b}^{-1}\Big[\hat{\Phi}_{N-1}\cdots\hat{\Phi}_{i+1}\Big[\hat{\Phi}_i\cdots\hat{\Phi}_2\big[\hat{\Psi}\big]\cdots\Big]\Big]\Big](\hat{I})-\tilde{f}_{\theta_k b}^{-1}\Big[\hat{\Upsilon}\Big[\hat{\Phi}_{N-1}\cdots\hat{\Phi}_{i+1}\Big[\hat{\Phi}_i\cdots\hat{\Phi}_2\big[\hat{\Psi}\big]\cdots\Big]\Big]\Big](\hat{I})\right)_{ij}^{c_N}\right|
$$

$$
\leq \sum_{c_{N-1}}\left|\sum_{\substack{B_{N-1}\in S\\ \delta_N\in\Lambda}}\varphi_{c_{N-1}c_N}^{N}(B_{N-1}^{-1}\delta_N)e_{c_{N-1}}^{N-1}\left(DA_{\theta_k}\left(x-\delta_N+b\right),A_{\theta_k}B_{N-1}\right)\right.
$$

$$
\left.-\sum_{\substack{B_{N-1}\in S\\ \delta_N\in\Lambda}}\varphi_{c_{N-1}c_N}^{N}(B_{N-1}^{-1}A_{\theta_k}^{-1}\delta_N)e_{c_{N-1}}^{N-1}\left(A_{\theta_k}(x+b)-\delta_N,B_{N-1}\right)\right|
$$

$$
\leq n_{N-1}\frac{C_N}{2}(p+1)^2h^2, \tag{87}
$$

where we do not specifically indicate the numbers of input and output channels, i.e. $\varphi^N(x)=\varphi_{c_{N-1}c_N}^{N}(x)$, and we have

$$
C_N=\sup\left(\left\|\nabla^2\varphi^N(x)\right\|\left|e_{c_{N-1}}^{N-1}(x,B)\right|+\left|\varphi^N(x)\right|\left\|\nabla^2 e_{c_{N-1}}^{N-1}(x,B)\right\|+2\left\|\nabla\varphi^N(x)\right\|\left\|\nabla e_{c_{N-1}}^{N-1}(x,B)\right\|\right).
$$

Substituting Eqs. (63), (64) and (65) into Eq. (87), denoting $\mathcal{F}_{N-1}=\prod_{k=1}^{N-1}n_{k-1}p^2F_k$, then we have

$$
\left|\left(\hat{\Upsilon}\Big[\tilde{f}_{\theta_k b}^{-1}\Big[\hat{\Phi}_{N-1}\cdots\hat{\Phi}_{i+1}\Big[\hat{\Phi}_i\cdots\hat{\Phi}_2\big[\hat{\Psi}\big]\cdots\Big]\Big]\Big](\hat{I})-\tilde{f}_{\theta_k b}^{-1}\Big[\hat{\Upsilon}\Big[\hat{\Phi}_{N-1}\cdots\hat{\Phi}_{i+1}\Big[\hat{\Phi}_i\cdots\hat{\Phi}_2\big[\hat{\Psi}\big]\cdots\Big]\Big]\Big](\hat{I})\right)_{ij}^{c_N}\right|
$$

$$
\leq\frac{n_{N-1}(p+1)^2h^2}{2}\left(\left\|\nabla^2\varphi^N(x)\right\|\left|e_{c_{N-1}}^{N-1}(x,B)\right|+\left|\varphi^N(x)\right|\left\|\nabla^2 e_{c_{N-1}}^{N-1}(x,B)\right\|+2\left\|\nabla\varphi^N(x)\right\|\left\|\nabla e_{c_{N-1}}^{N-1}(x,B)\right\|\right)
$$

$$
\leq 2n_{N-1}p^2F_N\left(\frac{H_N}{F_N}\left|e_{c_{N-1}}^{N-1}(x,B)\right|+\left\|\nabla^2 e_{c_{N-1}}^{N-1}(x,B)\right\|+2\frac{G_N}{F_N}\left\|\nabla e_{c_{N-1}}^{N-1}(x,B)\right\|\right)h^2
$$

$$
\leq 2n_{N-1}p^2F_N\mathcal{F}_{N-1}
$$

$$
\left(\frac{H_N}{F_N}F_0+\sum_{m=1}^{N-1}\frac{H_mF_0}{F_m}+2\sum_{l=1}^{N-1}\frac{G_l}{F_l}\sum_{m=1}^{l-1}\frac{G_mF_0}{F_m}+2\sum_{m=1}^{N-1}\frac{G_mG_0}{F_m}+H_0+\sum_{m=1}^{N-1}2\frac{G_NG_mF_0}{F_NF_m}+2\frac{G_NG_0}{F_N}\right)h^2
$$

$$
\leq 2\mathcal{F}\left(\sum_{m=1}^{N}\frac{H_mF_0}{F_m}+2\sum_{l=1}^{N}\frac{G_l}{F_l}\sum_{m=1}^{l-1}\frac{G_mF_0}{F_m}+2\sum_{m=1}^{N}\frac{G_mG_0}{F_m}+H_0\right)h^2. \tag{88}
$$

4) Substituting Eqs. (82), (86) and (88) into Eq. (79) we can get:

$$\left| \mathrm{g} \left[ \tilde{f}_{\theta_k b}^{-1} \right] (\hat{I}) - \tilde{f}_{\theta_k b}^{-1} [\mathrm{g}] (\hat{I}) \right|$$

$$\leq 2\mathcal{F} \left( \frac{H_1}{F_1} F_0 + H_0 + 2 \frac{G_1}{F_1} G_0 \right) h^2$$

$$+ \sum_{i=2}^{N-1} 2\mathcal{F} \left( \sum_{m=1}^{i} \frac{H_m F_0}{F_m} + 2 \sum_{l=1}^{i} \frac{G_l}{F_l} \sum_{m=1}^{l-1} \frac{G_m F_0}{F_m} + 2 \sum_{m=1}^{i} \frac{G_m G_0}{F_m} + H_0 \right) h^2$$

$$+ 2\mathcal{F} \left( \sum_{m=1}^{N} \frac{H_m F_0}{F_m} + 2 \sum_{l=1}^{N} \frac{G_l}{F_l} \sum_{m=1}^{l-1} \frac{G_m F_0}{F_m} + 2 \sum_{m=1}^{N} \frac{G_m G_0}{F_m} + H_0 \right) h^2$$

$$\leq 2\mathcal{F} \sum_{i=1}^{N} \left( \sum_{m=1}^{i} \frac{H_m F_0}{F_m} + 2 \sum_{l=1}^{i} \frac{G_l}{F_l} \sum_{m=1}^{l-1} \frac{G_m F_0}{F_m} + 2 \sum_{m=1}^{i} \frac{G_m G_0}{F_m} + H_0 \right) h^2. \tag{89}$$

Therefore, we achieve Eq. (77).

Finally, we will provide the error analysis for the N-layer rotation equivariant CNN network. Substituting Eqs. (73), (75) and (77) into Eq. (72), we can get:

$$\left| \tilde{f}_{\theta b}^{-1} \, \mathrm{g} \left[ \tilde{f}_{\theta b} \right] (I) - [\mathrm{g}](I) \right|$$

$$\leq \frac{2\pi}{t} \mathcal{F} \left( \max \{H, W\} + N(p+1) \right) h G_0 + \frac{2\pi}{t} \mathcal{F} \max \{H, W\} h G_0$$

$$+ 2\mathcal{F} \sum_{i=1}^{N} \left( \sum_{m=1}^{i} \frac{H_m F_0}{F_m} + 2 \sum_{l=1}^{i} \frac{G_l}{F_l} \sum_{m=1}^{l-1} \frac{G_m F_0}{F_m} + 2 \sum_{m=1}^{i} \frac{G_m G_0}{F_m} + H_0 \right) h^2$$

$$\leq \frac{2\pi}{t} \mathcal{F} \left( 2 \max \{H, W\} + N(p+1) \right) h G_0$$

$$+ 2\mathcal{F} \sum_{i=1}^{N} \left( \sum_{m=1}^{i} \frac{H_m F_0}{F_m} + 2 \sum_{l=1}^{i} \frac{G_l}{F_l} \sum_{m=1}^{l-1} \frac{G_m F_0}{F_m} + 2 \sum_{m=1}^{i} \frac{G_m G_0}{F_m} + H_0 \right) h^2 \tag{90}$$

Next, in order to get a more concise form, we further scale the entire error bound. By Eq. (90), we have:

$$\left| \tilde{f}_{\theta b}^{-1} \, \mathrm{g} \left[ \tilde{f}_{\theta b} \right] (I) - [\mathrm{g}] (I) \right|$$

$$= \frac{2\pi}{t} \mathcal{F} \left( 2 \max \{H, W\} p^{-1} + N(p+1) p^{-1} \right) p h G_0$$

$$+ 2\mathcal{F} \sum_{i=1}^{N} (N + 1 - i) \left( \frac{H_i F_0}{F_i} + 2 \frac{G_i}{F_i} \sum_{m=1}^{i-1} \frac{G_m F_0}{F_m} + 2 \frac{G_i G_0}{F_i} \right) h^2 + 2\mathcal{F} N H_0 h^2$$

$$\leq \frac{2\pi}{t} \mathcal{F} \left( 2 \max \{H, W\} p^{-1} + 2N \right) p h G_0 \tag{91}$$

$$+ 2\mathcal{F} N \sum_{i=1}^{N} \left( \frac{H_i F_0}{F_i} + 2 \frac{G_i}{F_i} \sum_{m=1}^{i-1} \frac{G_m F_0}{F_m} + 2 \frac{G_i G_0}{F_i} \right) h^2 + 2\mathcal{F} N H_0 h^2$$

$$\leq 2N\mathcal{F} \sum_{i=1}^{N} \left( \frac{H_i F_0}{F_i} + 2 \frac{G_i}{F_i} \sum_{m=1}^{i-1} \frac{G_m F_0}{F_m} + 2 \frac{G_i G_0}{F_i} + H_0 \right) h^2$$

$$+ 2\pi G_0 \mathcal{F} \left( 2 \max \{H, W\} p^{-1} + 2N \right) p h t^{-1}$$

If we denote

$$C_1 = 2N\mathcal{F} \cdot \sum_{i=1}^{N} \left( \frac{H_i F_0}{F_i} + 2 \frac{G_i}{F_i} \sum_{m=1}^{i-1} \frac{G_m F_0}{F_m} + 2 \frac{G_i G_0}{F_i} + H_0 \right),$$

$$C_2 = 2\pi G_0 \mathcal{F} \left( 2 \max\{H,W\} p^{-1} + 2N \right). \tag{92}$$

636 Then we can obtain the error bound of the N-layer rotation equivariant CNN network as Eq. (70)

637 $\square$

638 **Lemma 5.** *Based on the same conditions, for an arbitrary $0 \leq \theta \leq 2\pi$, $A_\theta \in S$ denotes the rotation*
639 *matrix, $b \in \mathbb{R}^2$ denotes the translation, the following result is satisfied:*

$$\left\| \mathrm{g} \left[ \tilde{f}_{\theta b} \right] (I) - \tilde{f}_{\theta b} \left[ \mathrm{g} \right] (I) \right\|_\infty \leq C_1 h^2 + C_2 pht^{-1}, \tag{93}$$

640 *where $C_1$ and $C_2$ are defined in (92).*

641 The proof of Lemma 5 is similar to Theorem 1 of [38]. Please refer to [38] for more details.

## A.4 Proposition 1 and the Proof

643 Based on Theorem 1, we proceed to present the Proposition 1 in the main text and its proof.

644 **Proposition 1.** *For images $I_0$ and $I_j$ with size $H \times W \times n_0$, and a N-layer rotation-translation*
645 *equivariant CNN network $\mathrm{g}(\cdot)$, whose channel number of the $i^{th}$ layer is $n_i$, rotation equivariant*
646 *subgroup is $S \leqslant O(2)$, $|S| = t$, and activation function is set as ReLU. If the latent continuous*
647 *function of the $c^{th}$ channel of $I_j$ and $I_0$ are denoted as $r_c : \mathbb{R}^2 \to \mathbb{R}$ and $\tilde{r}_c : \mathbb{R}^2 \to \mathbb{R}$, respectively,*
648 *and the latent continuous function of any convolution filters in the $i^{th}$ layer is denoted as $\varphi^i : \mathbb{R}^2 \to \mathbb{R}$,*
649 *where $i \in \{1, \cdots, N\}$, $c \in \{1, \cdots, n_0\}$, for any $x \in \mathbb{R}^2$, the following conditions are satisfied:*

$$|r_c(x)|, |\tilde{r}_c(x)| \leq F_0, \|\nabla r_c(x)\|, \|\nabla \tilde{r}_c(x)\| \leq G_0, \|\nabla^2 r_c(x)\|, \|\nabla^2 \tilde{r}_c(x)\| \leq H_0,$$
$$|\varphi^i(x)| \leq F_i, \|\nabla \varphi^i(x)\| \leq G_i, \|\nabla^2 \varphi^i(x)\| \leq H_i, \tag{94}$$
$$\forall \|x\| \geq {}^{(p+1)h}\!/_2, \ \varphi_i(x) = 0,$$

650 *where $p$ is the filter size, $h$ is the mesh size, $\nabla$ and $\nabla^2$ denote the operators of gradient and Hessian*
651 *matrix, respectively. For an arbitrary $0 \leq \theta \leq 2\pi$ and a feature map of equivariant convolution*
652 *$Z = \mathrm{g}(I)$ with size $H \times W \times tC$, $A_\theta \in S$ denotes the rotation matrix, $b \in \mathbb{R}^2$ denotes the translation,*
653 *the following result is satisfied:*

$$\left\| \tilde{f}_{\theta b}^{-1}(Z_j) - Z_0 \right\|_\infty \leq C_3 \left\| \tilde{f}_{\theta b}^{-1}(I_j) - I_0 \right\|_2 + C_1 h^2 + C_2 pht^{-1}, \tag{95}$$

654 *where $\tilde{f}_{\theta b}$ is defined in Eq. (60) and*

$$C_1 = 2N\mathcal{F} \cdot \sum_{i=1}^{N} \left( \frac{H_i F_0}{F_i} + 2 \frac{G_i}{F_i} \sum_{m=1}^{i-1} \frac{G_m F_0}{F_m} + 2 \frac{G_i G_0}{F_i} + H_0 \right),$$
$$C_2 = 2\pi G_0 \mathcal{F} \left( 2 \max\{H, W\} p^{-1} + 2N \right), \tag{96}$$
$$C_3 = \prod_{k=1}^{N} n_{k-1} p^2 F_k.$$

655 *Proof.* We can deduce that

$$\begin{aligned} & \left\| \tilde{f}_{\theta b}^{-1}(Z_j) - Z_0 \right\|_\infty \\ = & \left\| \tilde{f}_{\theta b}^{-1} \left[ \mathrm{g} \right] (I_j) - \mathrm{g} \left[ \tilde{f}_{\theta b}^{-1} \right] (I_j) + \mathrm{g} \left[ \tilde{f}_{\theta b}^{-1} \right] (I_j) - \mathrm{g}(I_0) \right\|_\infty \\ \leq & \underbrace{\left\| \tilde{f}_{\theta b}^{-1} \left[ \mathrm{g} \right] (I_j) - \mathrm{g} \left[ \tilde{f}_{\theta b}^{-1} \right] (I_j) \right\|_\infty}_{\langle 1 \rangle} + \underbrace{\left\| \mathrm{g} \left[ \tilde{f}_{\theta b}^{-1} \right] (I_j) - \mathrm{g}(I_0) \right\|_\infty}_{\langle 2 \rangle} \end{aligned} \tag{97}$$

656 For the part $\langle 1 \rangle$, by exploiting Lemma 5, we have

$$\left\| \tilde{f}_{\theta b}^{-1} \left[ \mathrm{g} \right] (I_j) - \mathrm{g} \left[ \tilde{f}_{\theta b}^{-1} \right] (I_j) \right\|_\infty \leq C_1 h^2 + C_2 pht^{-1}, \tag{98}$$

where $C_1$ and $C_2$ are defined in (92). For the part $\langle 2 \rangle$, consistent with the mathematical notation defined in Section 1.3., $x_{kl}$ denotes the coordinates of the point $(k, l)$. Let $\hat{r_c}(x) = r_c(A_\theta x + b)$, then $\hat{r_c}(x)$ is the latent function of $\tilde{f}_{\theta b}^{-1}(I_j)$. Besides,

$$|\hat{r_c}(x)| \leq F_0, \|\nabla \hat{r_c}(x)\| \leq G_0, \|\nabla^2 \hat{r_c}(x)\| \leq H_0. \tag{99}$$

Then we can deduce that

$$\left| \left( g\left[\tilde{f}_{\theta b}^{-1}\right](I_j) - g(I_0) \right)_{kl} \right|$$

$$= \sum_{\substack{c_{N-1} \\ B_{N-1} \in S \\ \delta_N \in \Lambda}} \cdots \sum_{\substack{c_1 \\ A \in S \\ \delta_2 \in \Lambda}} \sum_{\substack{c_0 \\ \delta_1 \in \Lambda}} \varphi_{c_{N-1}c_N}^N (B_{N-1}^{-1} \delta_N) \cdots \varphi_{c_0 c_1}^1 \left( A^{-1} \delta_1 \right)$$

$$(\hat{r_c}(x_{kl} - \delta_N - \cdots - \delta_1) - \tilde{r_c}(x_{kl} - \delta_N - \cdots - \delta_1))$$

$$= \sum_{\substack{c_{N-1} \\ B_{N-1} \in S \\ \delta_N \in \Lambda}} \cdots \sum_{\substack{c_1 \\ A \in S \\ \delta_2 \in \Lambda}} \sum_{\substack{c_0 \\ \delta_1 \in \Lambda}} \left| \varphi_{c_{N-1}c_N}^N (B_{N-1}^{-1} \delta_N) \right| \cdots \left| \varphi_{c_0 c_1}^1 \left( A^{-1} \delta_1 \right) \right|$$

$$\left| \hat{r_c}(x_{kl} - \delta_N - \cdots - \delta_1) - \tilde{r_c}(x_{kl} - \delta_N - \cdots - \delta_1) \right|$$

$$\leq \left( \prod_{k=1}^N n_{k-1} p^2 F_k \right) \left\| \tilde{f}_{\theta b}^{-1}(I_j) - I_0 \right\|_\infty$$

$$\leq \left( \prod_{k=1}^N n_{k-1} p^2 F_k \right) \left\| \tilde{f}_{\theta b}^{-1}(I_j) - I_0 \right\|_2.$$

$$\tag{100}$$

Finally, by substituting Eqs. (98) and (100) into Eq. (97), we have

$$\left\| \tilde{f}_{\theta b}^{-1}(Z_j) - Z_0 \right\|_\infty$$

$$\leq C_1 h^2 + C_2 p h t^{-1} + \left( \prod_{k=1}^N n_{k-1} p^2 F_k \right) \left\| \tilde{f}_{\theta b}^{-1}(I_j) - I_0 \right\|_2. \tag{101}$$

If we denote

$$C_3 = \prod_{k=1}^N n_{k-1} p^2 F_k, \tag{102}$$

then we can obtain Eq. (95), the proof is then completed. $\qquad\square$

# B  Supplementary Results

In this section, we first provide full-size visualized results of comparison experiments and ablation study in the main text. Fig. 8-11 are the full-size version of comparison results on x4 SyntheticBurst [3] dataset. Fig. 12-15 are the full-size version of comparison results on x4 BurstSR [23] dataset. Fig. 16-18 are the full-size visualized results of ablation study on x4 SyntheticBurst dataset.

In addition, we provide comprehensive multi-scale super-resolution (SR) comparisons on the SyntheticBurst and BurstSR datasets for both ×2 and ×3 scaling factors to further validate the robustness and generalization of our method. Among the compared methods, only BurstM [9] supports multi-scale SR, while other methods exhibit inferior performance compared to both BurstM and our approach. Therefore, we focus our comparison on BurstM as the primary baseline.

The quantitative results are in Table 1 in the main text. The qualitative results on the x2 and x3 SyntheticBurst dataset [3] are presented in Fig. 19 and Fig. 20. The error maps clearly demonstrate that our method achieves superior reconstruction quality, recovering finer details than BurstM. For the x2 and x3 BurstSR dataset [23], as shown in Fig. 21 and Fig. 22, we provide enlarged patches for visual comparison due to the absence of ground truth images. The results highlight that our method preserves more structural information and produces visually sharper results compared to BurstM, further validating its effectiveness in real-world burst SR scenarios. These consistent improvements across datasets and scaling factors underscore the robustness and generalizability of our approach.

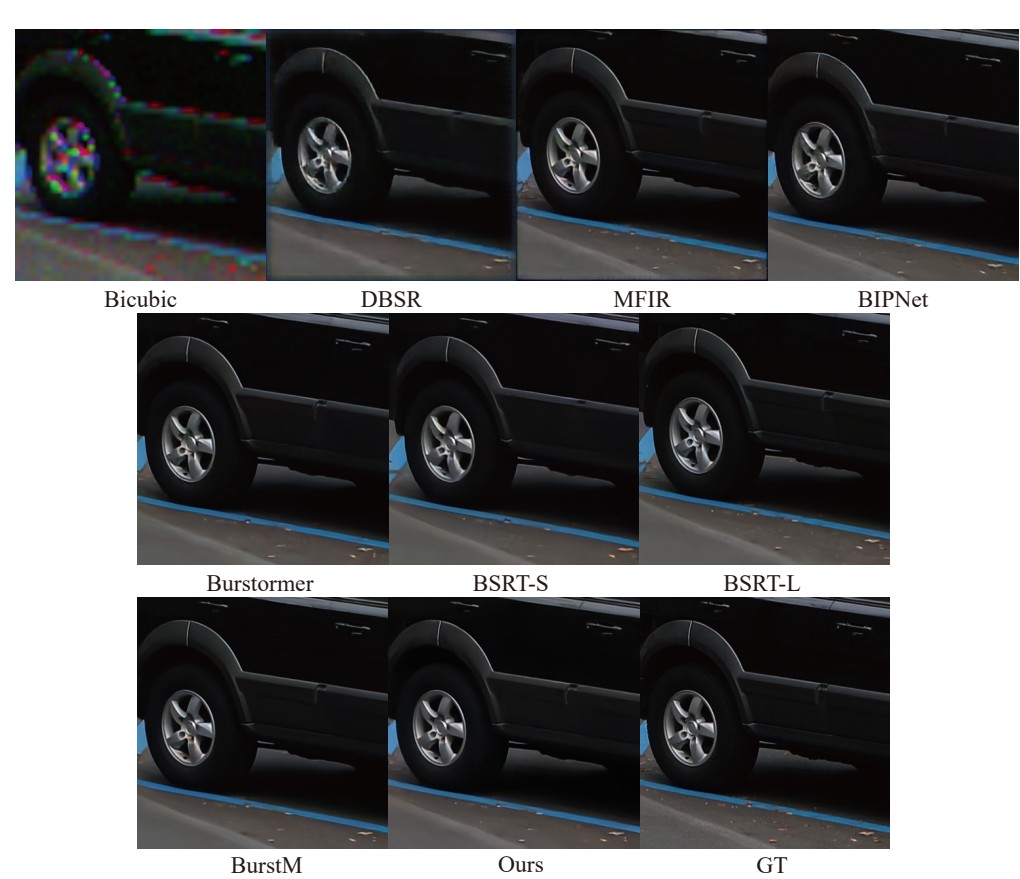

Figure 8: Visual comparison of x4 BISR on the SyntheticBurst dataset.

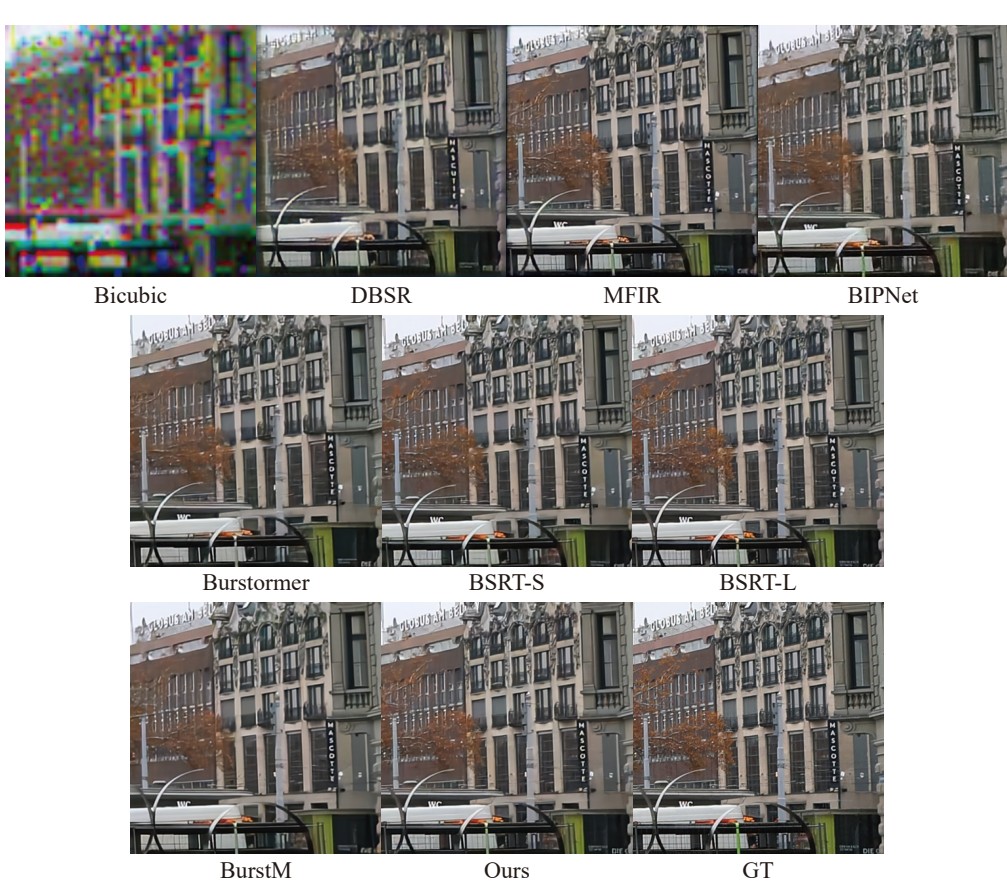

Figure 9: Visual comparison of x4 BISR on the SyntheticBurst dataset.

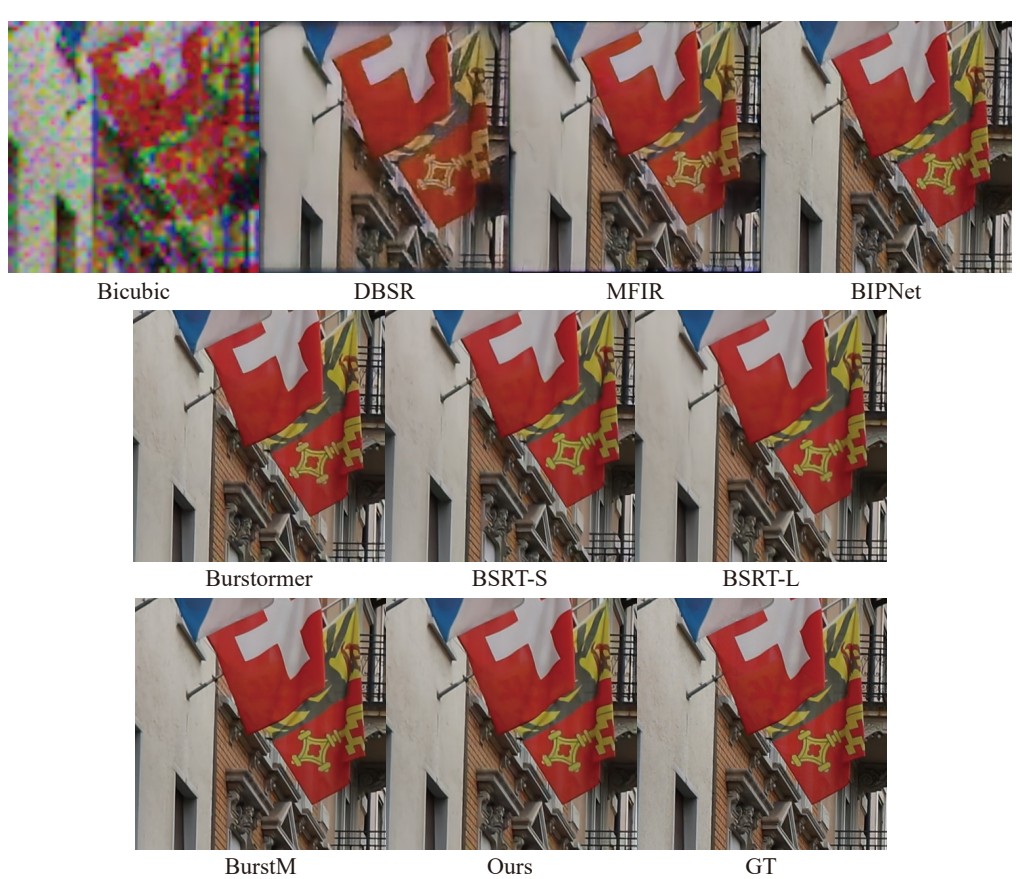

Figure 10: Visual comparison of x4 BISR on the SyntheticBurst dataset.

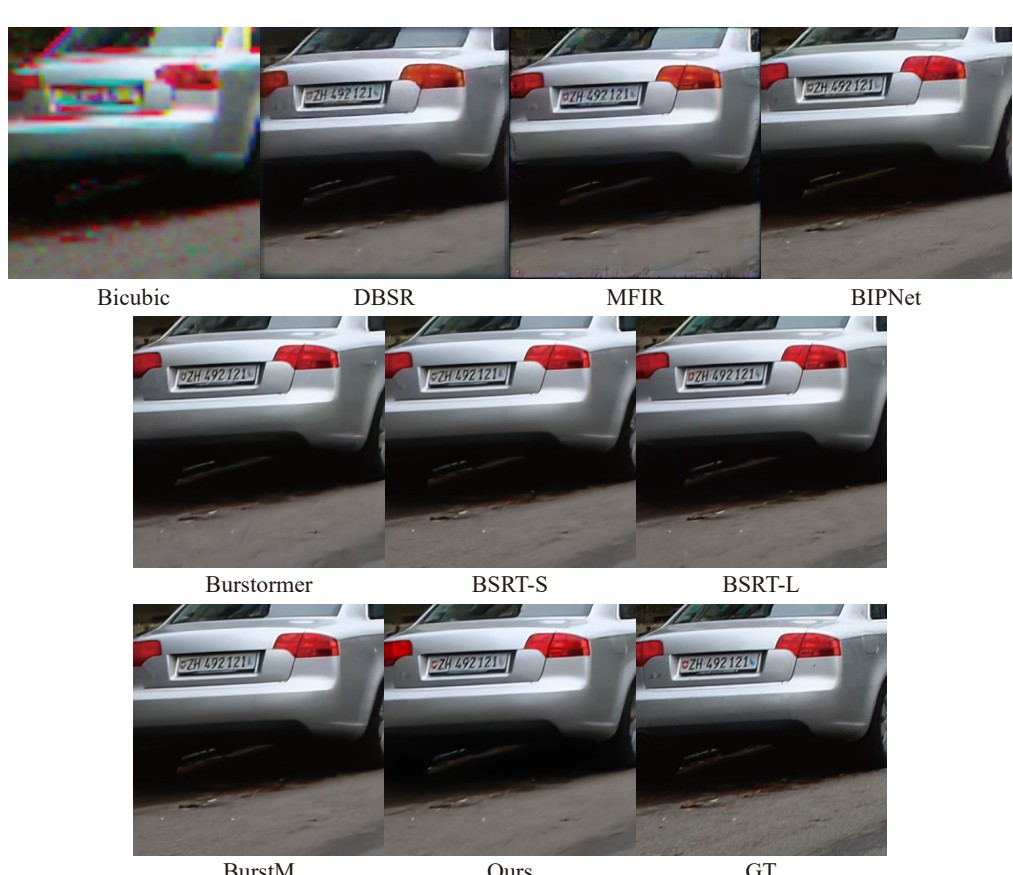

Figure 11: Visual comparison of x4 BISR on the SyntheticBurst dataset.

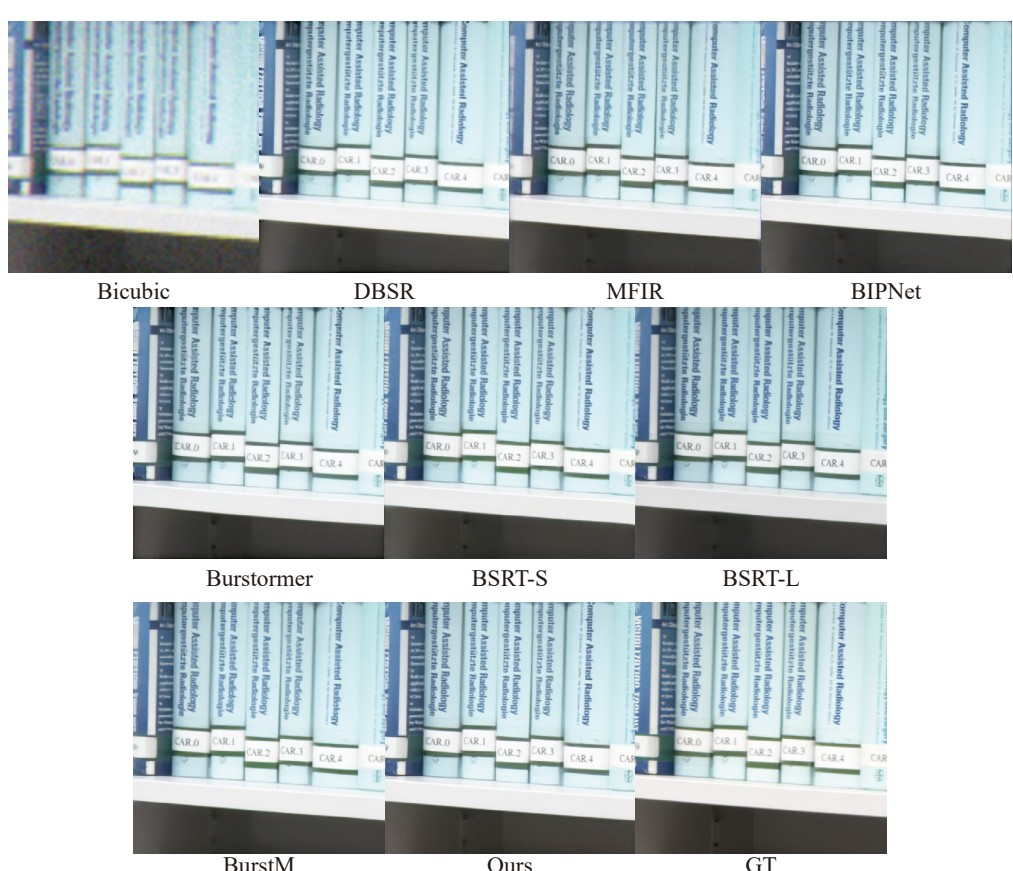

Figure 12: Visual comparison of x4 BISR on the BurstSR dataset.

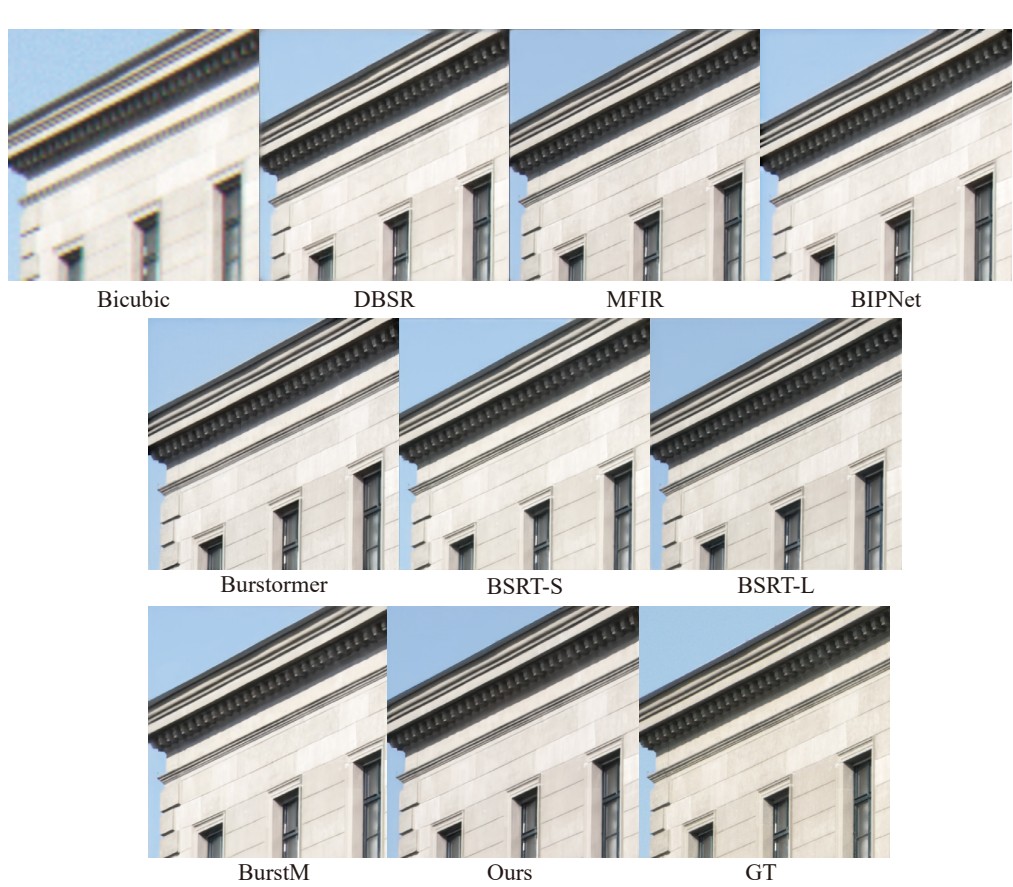

Figure 13: Visual comparison of x4 BISR on the BurstSR dataset.

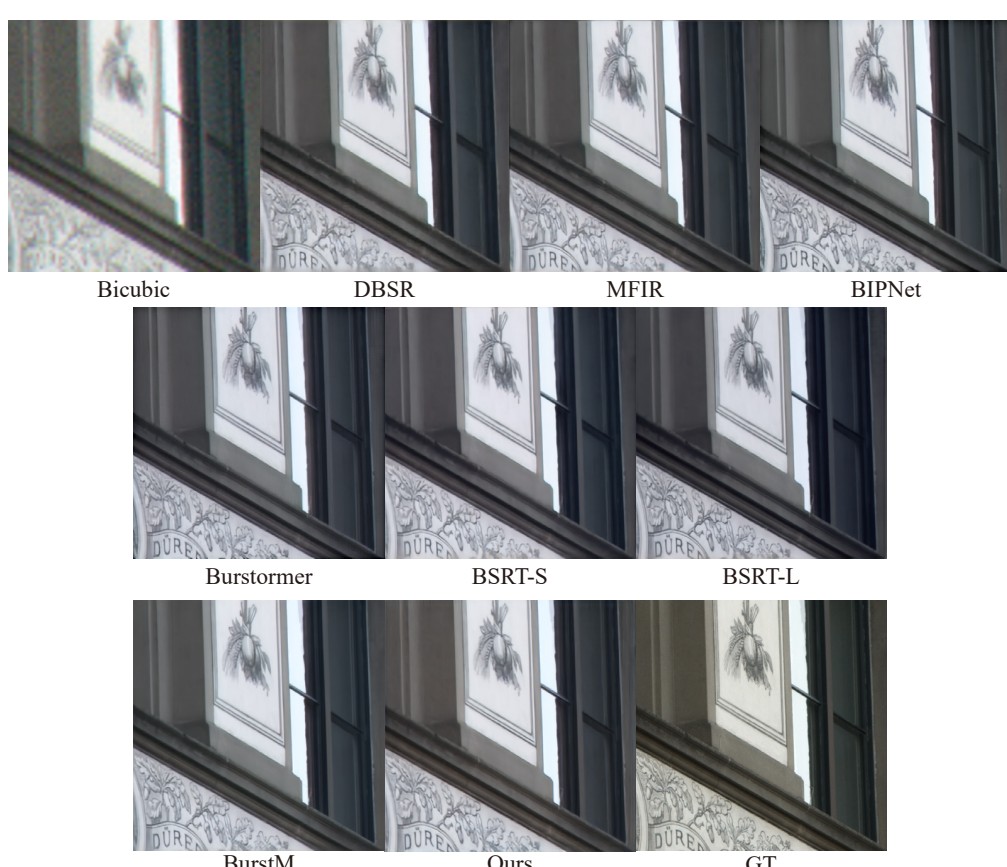

| Bicubic | DBSR | MFIR | BIPNet |
| Burstormer | BSRT-S | BSRT-L | |
| BurstM | Ours | GT | |

Figure 14: Visual comparison of x4 BISR on the BurstSR dataset.

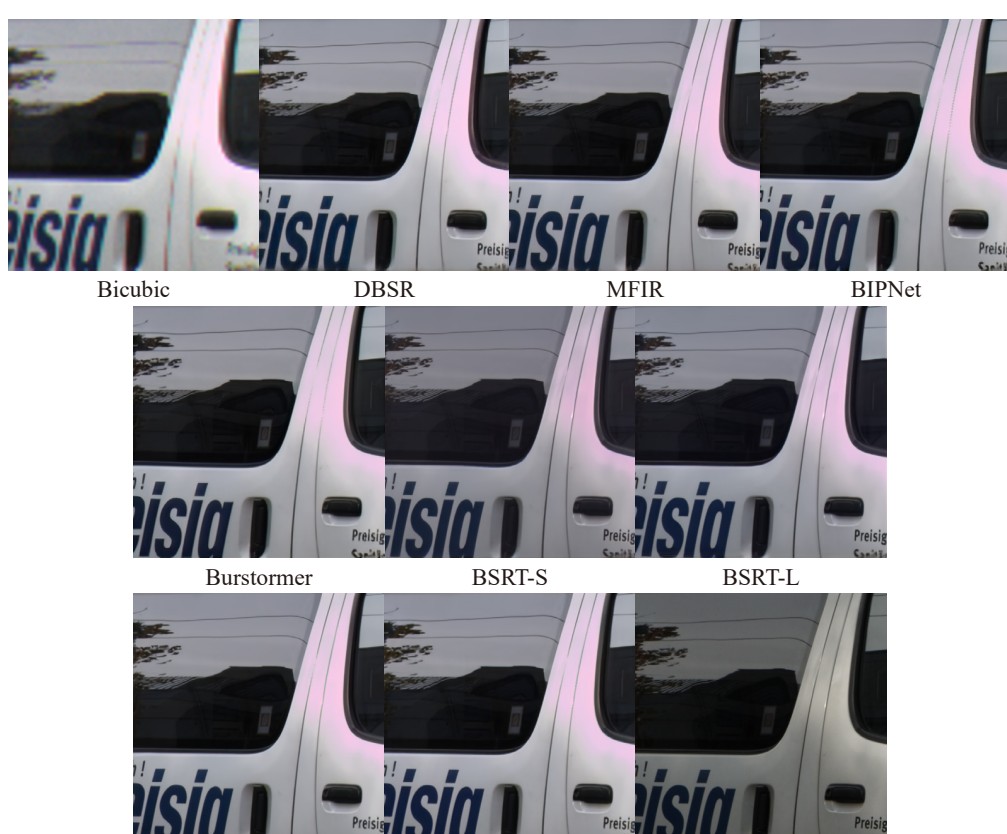

Figure 15: Visual comparison of x4 BISR on the BurstSR dataset.

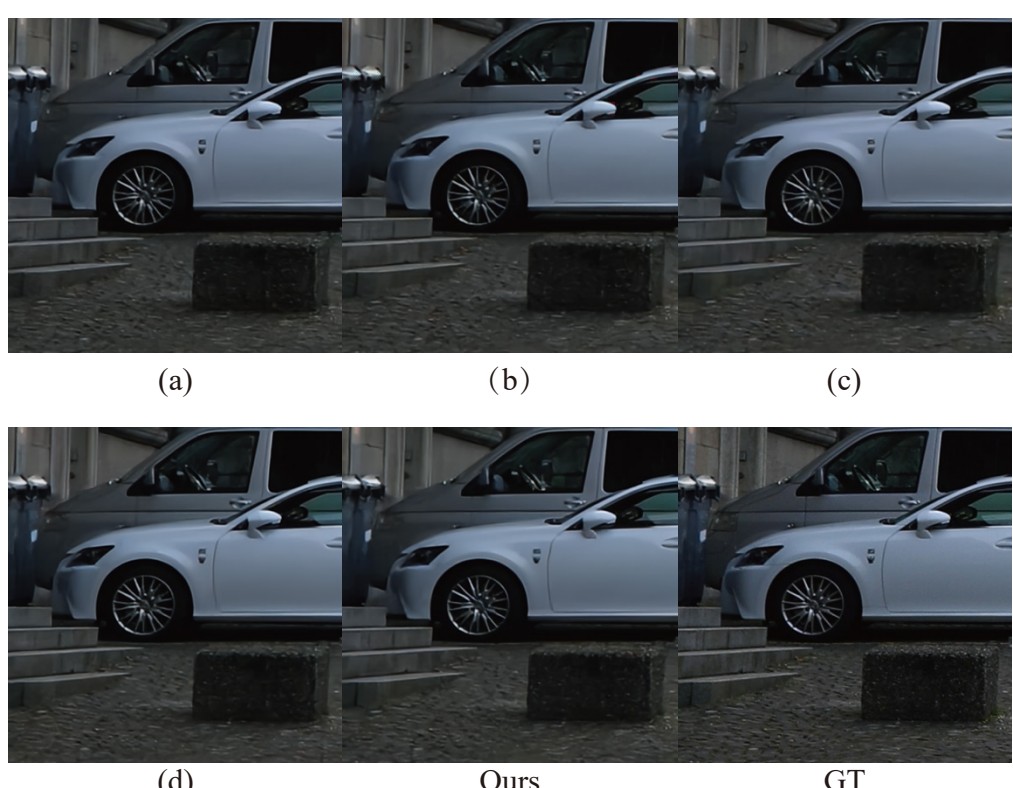

Figure 16: Visual comparison of ablation study for x4 BISR on the SyntheticBurst dataset.

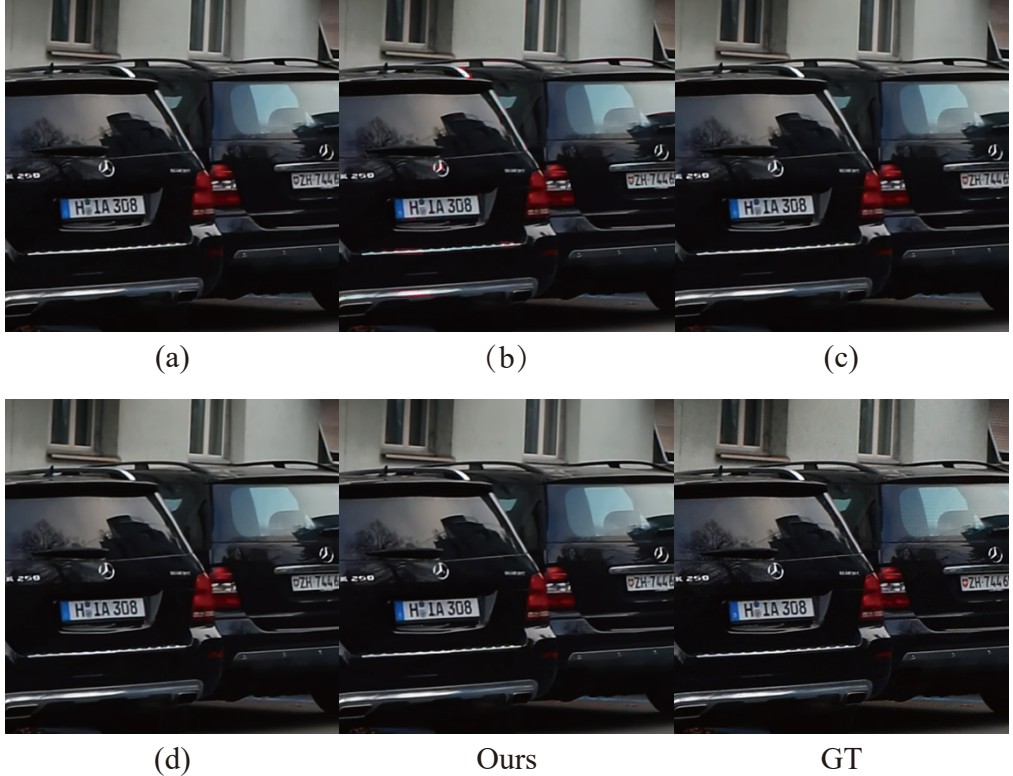

Figure 17: Visual comparison of ablation study for x4 BISR on the SyntheticBurst dataset.

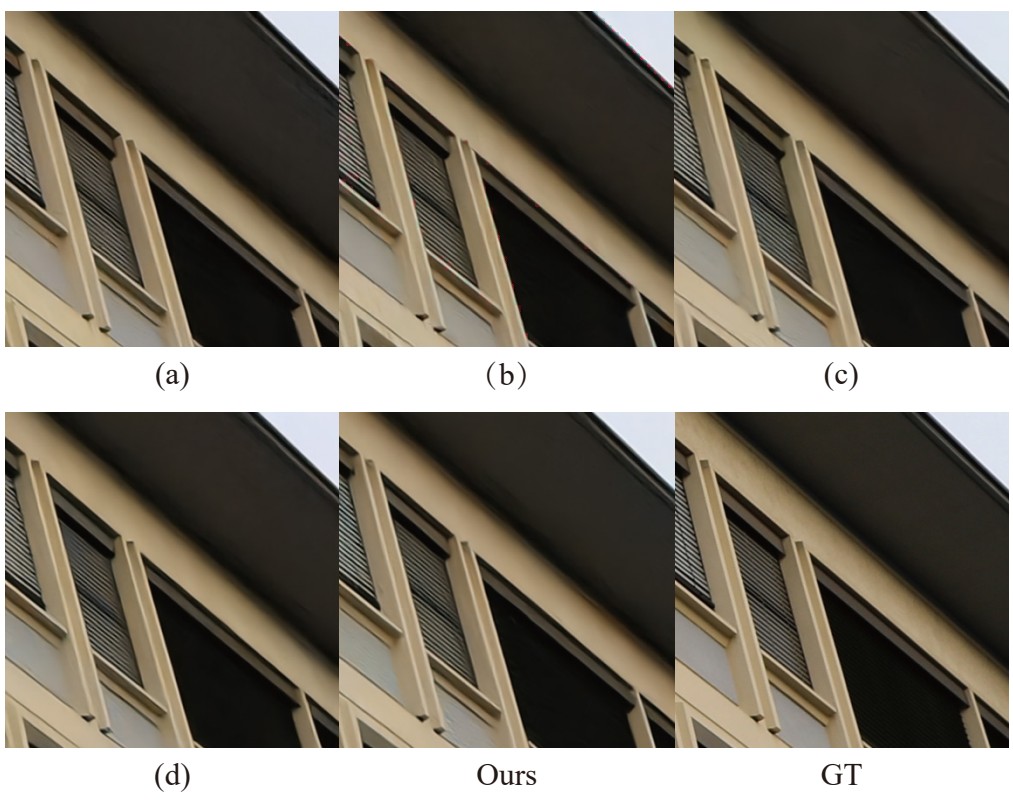

(a)          (b)          (c)

(d)          Ours          GT

Figure 18: Visual comparison of ablation study for x4 BISR on the SyntheticBurst dataset.

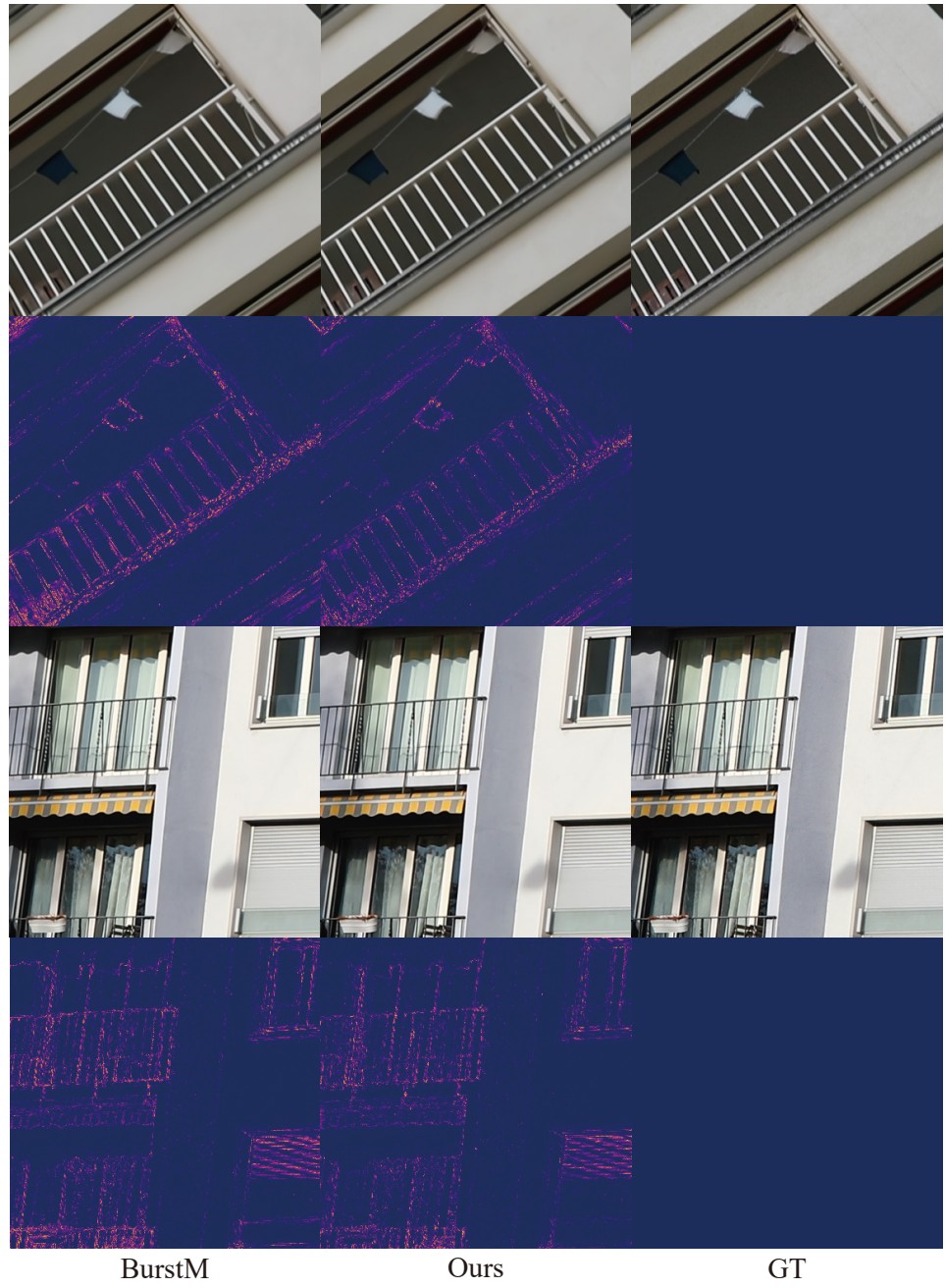

BurstM                    Ours                    GT

Figure 19: Visual comparison x2 BISR on the SyntheticBurst dataset.

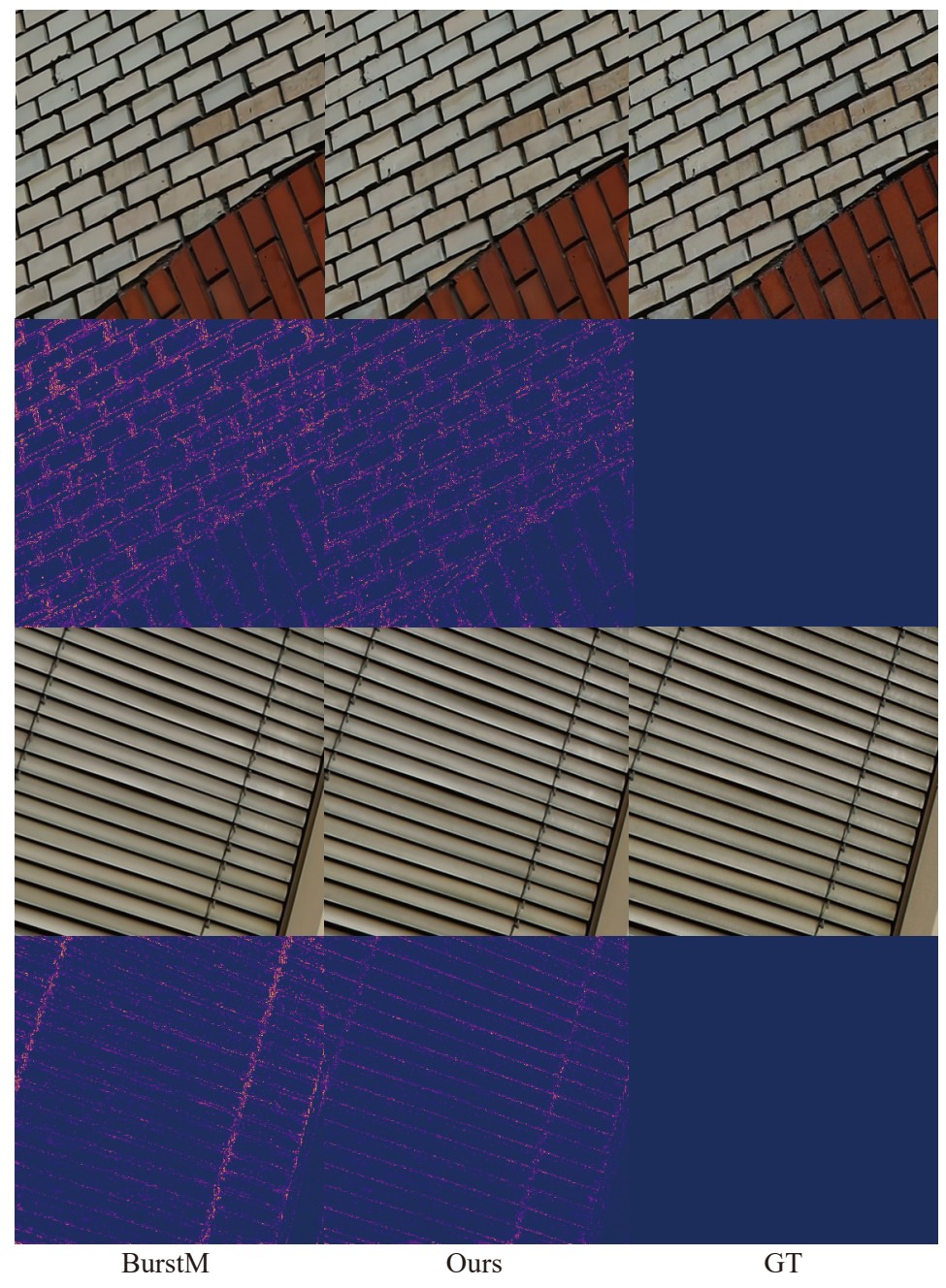

BurstM                    Ours                    GT

Figure 20: Visual comparison x3 BISR on the SyntheticBurst dataset.

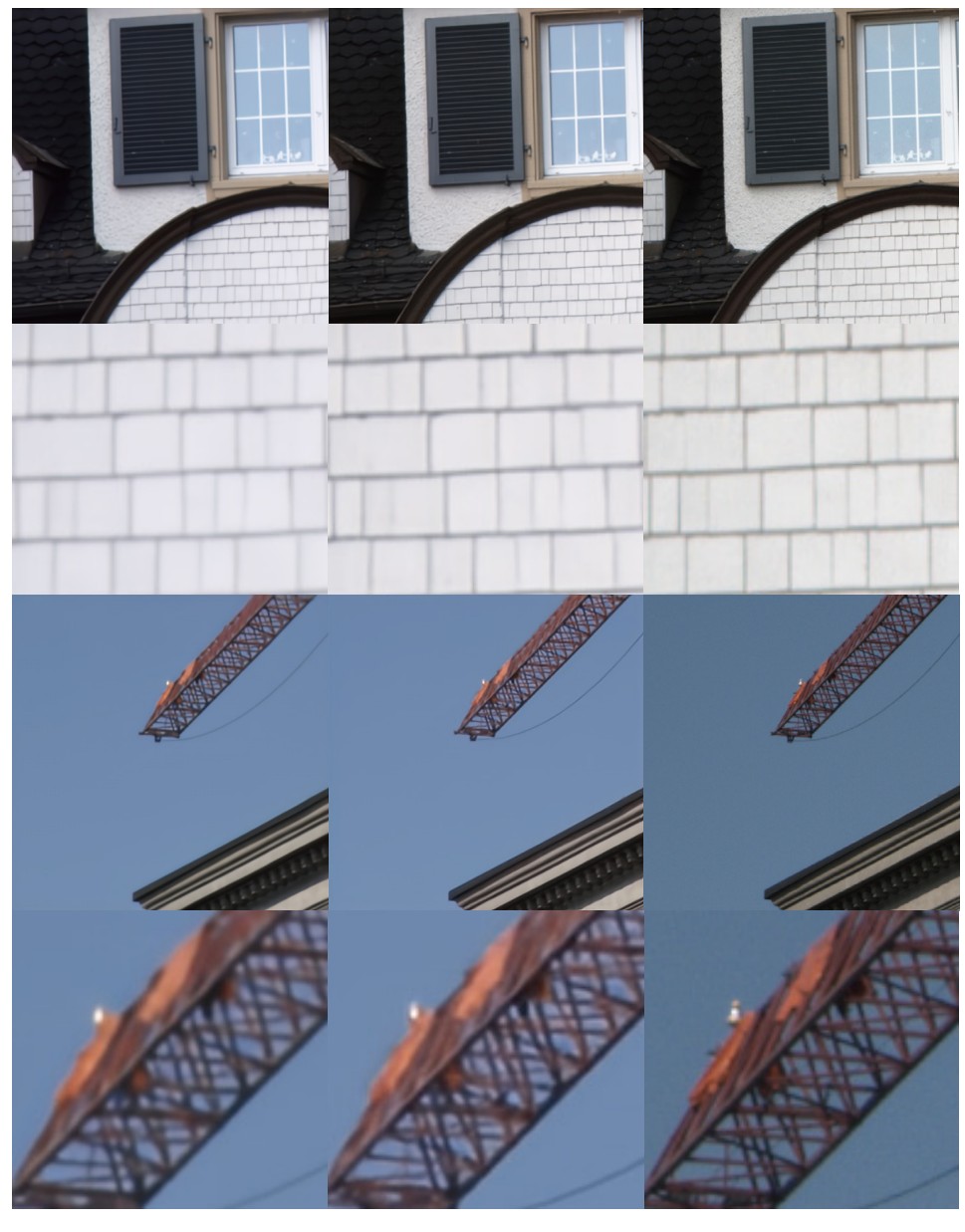

BurstM                        Ours                          GT

Figure 21: Visual comparison x2 BISR on the BurstSR dataset.

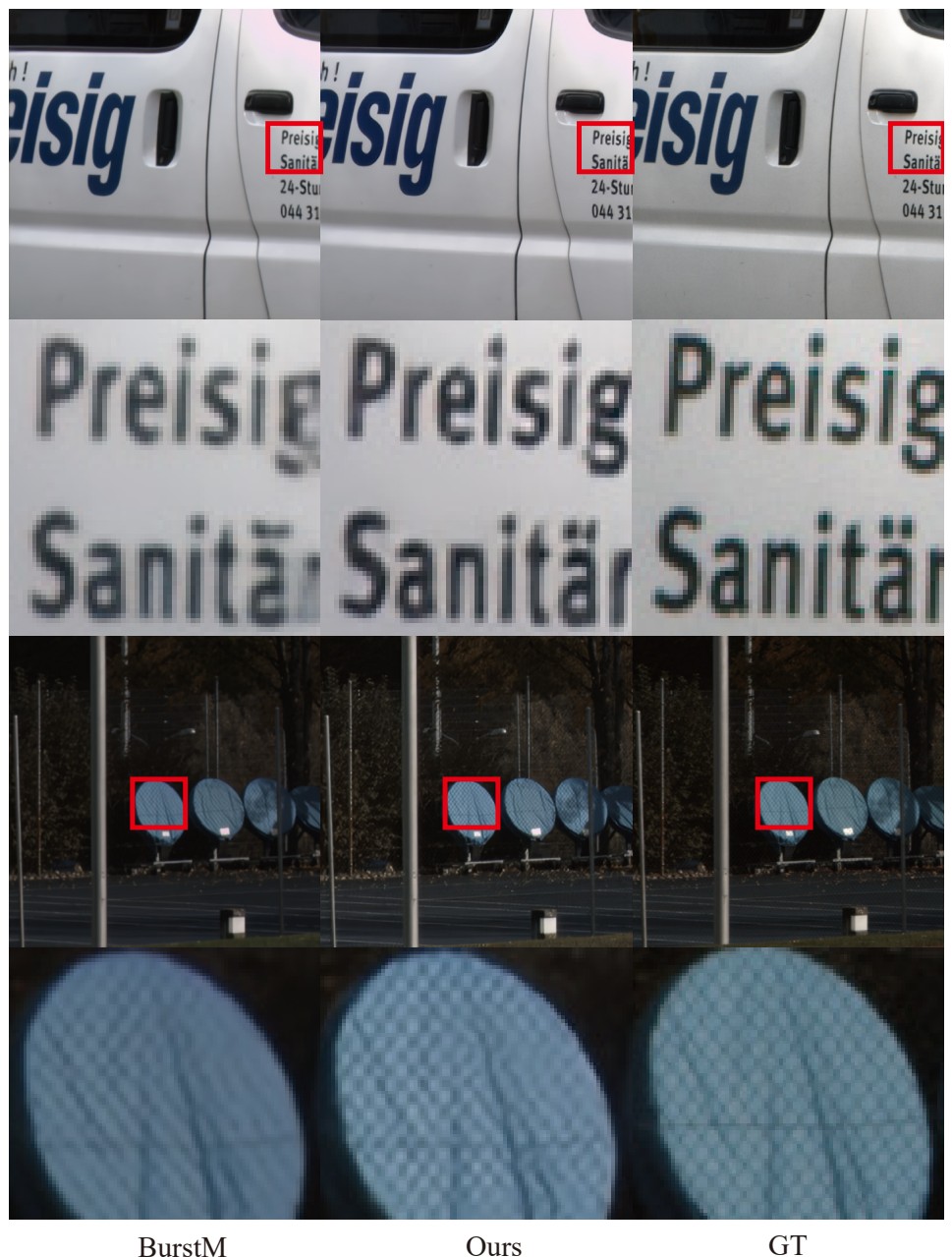

|     |      |    |
|-----|------|----|
| BurstM | Ours | GT |

Figure 22: Visual comparison x3 BISR on the BurstSR dataset.

