# OpenReview forum: "Feature Alignment with Equivariant Convolutions for Burst Image Super-Resolution"
_NeurIPS.cc/2025/Conference — Submitted to NeurIPS 2025_

### Official Review · Reviewer_ZHcB · 2025-07-01

**Clarity:** 3
**Significance:** 2
**Originality:** 2
**Rating:** 3
**Confidence:** 5

**Summary:**

This paper proposes an equivariant convolution-based alignment method for burst image super-resolution. The approach provides consistent transformations between the image and feature domains, enabling pixel-level supervised alignments to be effectively transferred to the feature space. As a result, the proposed method achieves more accurate alignment and improves overall reconstruction performance. Additionally, the reconstruction module is enhanced through the integration of components such as MDTA and INR. The effectiveness of the proposed framework is validated on standard benchmark datasets.

**Questions:**

Impact of Frame Distance from Reference Frame:
- In burst image sequences, the content difference between the reference frame and nearby frames is typically smaller than that with distant frames. Has the model's performance been analyzed in terms of frame distance from the reference? It would be insightful to understand whether the alignment module performs consistently across all frames, or whether distant frames degrade performance due to increased motion or occlusion.

Effect of Number of Input Frames (N = 14):

- The proposed method uses 14 frames as input. Is this number empirically determined to be optimal? Intuitively, increasing the number of frames may improve performance up to a certain point, after which gains are marginal or even negative due to noise, misalignment, or computational overhead. An ablation study or saturation analysis on the number of input frames would strengthen the paper and offer practical insights into the trade-off between performance and computational cost.

**Ethical Concerns:**

["NO or VERY MINOR ethics concerns only"]

**Final Justification:**

Thank you very much for the authors’ additional clarifications. While I understand the intention behind the proposed method and acknowledge the authors' claim of novelty, I still believe that the work does not meet the standards expected of a top-tier conference. So, I keep my original rating.

**Limitations:**

While the proposed method assumes that the transformation between burst image frames can be effectively modeled using rotation and translation, it remains unclear whether this assumption holds in all cases. In real-world scenarios, especially when handheld devices or rolling shutter effects are involved, more complex transformations such as affine or even projective warping may occur. It would be helpful for the authors to clarify whether the model can adequately handle such cases, or to provide empirical justification that rotation-translation equivariance is sufficient for the majority of burst image sequences.

**Paper Formatting Concerns:**

The spacing between Figure 4 and the surrounding text appears to be too narrow, which may affect readability and visual clarity. It is recommended to adjust the formatting to provide more appropriate separation between the figure, its caption, and the main text.

**Quality:**

3

**Strengths And Weaknesses:**

Strengths

- The proposed use of equivariant convolution for alignment addresses a critical challenge in burst image super-resolution (BISR), where accurate alignment significantly impacts reconstruction quality.

- The reconstruction module is enhanced by integrating MDTA and INR, leading to performance improvements.

- The inclusion of theoretical justification for the equivariant design is a strength, lending analytical support to the proposed method.

Weaknesses and Suggestions for Improvement

- While the application of equivariant convolution to BISR appears to be novel, the approach itself mainly adapts existing techniques. As such, the methodological contribution may be seen as incremental rather than innovative.

- The reconstruction module relies on the integration of established components (e.g., MDTA, INR) in an ad hoc manner. This fusion lacks a principled justification and further diminishes the perceived novelty of the work.

- Among the contributions listed at the end of the introduction, the final point refers to empirical performance improvements. Since this is an experimental observation rather than a technical contribution, it does not strengthen the core claims. Thus, the paper essentially presents two main contributions: (1) the application of equivariant convolution to BISR, and (2) a reconstruction module with known techniques. Given this, the overall contribution volume may be considered limited.

- In Equation (1), the transformation function $f(\cdot)$ is defined as rotation and translation. This raises a question: does the method fail to capture more general transformations such as affine or projective ones? Is invariance to only rotation and translation sufficient for real-world burst frames? If so, further empirical or theoretical justification is needed.

- The paper lacks evaluation using perceptual metrics such as LPIPS or FID, which are especially relevant for assessing visual quality in image generation tasks.

- While the proposed method shows improved quantitative results (e.g., PSNR, SSIM), the qualitative improvements are less convincing. As shown in both the main paper and supplementary material, the visual differences between the proposed method and baselines are subtle and often only apparent when images are heavily zoomed in.


- Given that alignment is the core contribution of this paper, relying solely on PSNR and SSIM may be insufficient to evaluate its full effectiveness. For example, recent reference-based SR methods like CSSR include edge-based or structure-aware metrics(such as gradient magnitude similarity deviation (GMSD)) to better capture alignment quality. Incorporating similar evaluations could help highlight the strengths of the proposed alignment module.

  - [CSSR] CSSR: Cross-and Self-feature Transformer with High-Frequency Feature Alignment for Reference-Based Super-Resolution
  - [GMSD] Gradient magnitude similarity deviation: A highly efficient perceptual image quality index. IEEE TIP 2013

---

> ### Author Rebuttal · Authors · 2025-07-30
>
> >**Q1:** While the application ... innovative.
>
> **A1:** While our work builds upon the established principles of Eq-CNN, its application to the BISR task is both novel and non-trivial. Importantly, we reveal that present alignment strategies adopted in BISR lack theoretical grounding and, in some cases, may be theoretically invalid as demonstrated in Figure 1 of the manuscript, leading to less accurate feature alignment (Table 1, Figure 2 of the manuscript).
> Furthermore, our theoretical analysis not only guide the design of our alignment framework but also advances the understanding of Eq-CNNs more broadly.
>
> >**Q2:** The reconstruction module ... work.
>
> **A2:** INR is essential for multi-scale SR, without which we can only handle single-scale SR, making it irreplaceable. MDTA introduces channel-wise self-attention via cross-covariance, effectively capturing global context and enhancing reference-source feature integration.
> To verify the necessity of MDTA, we added ablation experiments by removing MDTA while keeping INR. As shown in Table 1, replacing MDTA with a CNN results in a 0.15 drop in PSNR, highlighting its importance.
>
> **Table 1. Ablation Study on MDTA**
>
> | Settings             | PSNR  | SSIM  | Params. (M) |
> |----------------------|:-----:|:-----:|:-----------:|
> | MDTA->CNN          | 43.03 | 0.973 | 8.9         |
> | Ours             | **43.18** | **0.974** | **8.7**     |
>
> >**Q3:** Among the contributions ... limited.
>
> **A3:** As addressed in **A1**, our main contributions include a novel application of Eq-CNN to learn alignment in the image domain and transfer it to the feature domain, which is theoretically sound and previously unexplored in BISR. We also design a reconstruction module that combines this alignment with advanced upsampling and fusion techniques, as demonstrated in **A2**. Furthermore, our theoretical analysis advances the understanding of Eq-CNNs.
>
> As for the final point on empirical performance gains, while it may not be a technical novelty, it is crucial for validating the effectiveness of our method. Notably, prior works [2,3] also present performance improvements as a key contribution, particularly when achieving new SOTA. Nevertheless, we appreciate the suggestion and will revise the writing to better summarize and clarify our contributions.
>
> >**Q4:** In Equation (1) ... needed.
>
> **A4:** While our method may not capture all transformations, assuming rotation and translation is practical for BISR, as burst frames are typically captured in a short time, and this assumption aligns with common benchmarks [1].
>
> Table 4 in **A8** demonstrates the effectiveness of our alignment under real conditions. We also compare our method with optical flow, used in SOTA methods like BurstM [3], in both image and feature domains (see Table 1 and Figure 2 of the manuscript). Our method better handles misalignment, supporting the validity of our design. Besides, this limitation is discussed in the Limitation section of the manuscript.
>
> >**Q5:** The paper lacks ... tasks.
>
> **A5:** As suggested, we added LPIPS evaluation. For fairness, all methods were tested using their released ×4 pre-trained models.  As shown in Table 2, our method achieves consistently lower scores on both synthetic and real datasets. On BurstSR, it obtains the lowest score. On SyntheticBurst, it ranks second only behind BSRT-Large, a much larger and more computationally expensive model. Compared to the SOTA method BurstM, our method achieves a 0.004 lower LPIPS. These results further show the strength of our method in perceptual fidelity.
>
> **Table 2. Comparison of LPIPS**
> | Method      | SyntheticBurst   | BurstSR  |
> |-------------|:-----:|:-----:|
> | DBSR        | 0.081 | 0.029 |
> | MFIR        | 0.045 | 0.023 |
> | BIPNet      | 0.035 | 0.026 |
> | Burstormer  | 0.031 | 0.023 |
> | BSRT_S      | 0.031 | 0.021 |
> | BSRT_L      | **0.025** | 0.021 |
> | BurstM      | 0.032 | *0.020* |
> | Ours        | *0.028* | **0.019** |
>
>
> >**Q6:** While the proposed ... zoomed in.
>
> **A6:**  We first want to clarify that, in the image super-resolution tasks, fine-grained detail recovery is one of the primary goals, with improvements often apparent only in zoomed-in high-frequency regions. Therefore, zoom-in visualization is a standard practice in recent SOTA methods [2,3]. Following this, we adopted the same strategy to enable fair and meaningful comparison. As shown in Figures 4–6 of the manuscript, our method better preserves textures and reduces artifacts, demonstrating superior detail recovery.
>
> > **Q7:** Given that alignment ... module.
>
> **A7:** As suggested, we have added the GMSD metric to assess the alignment quality. As shown in Table 3, our method achieves the best scores across all compared methods, which highlights the effectiveness of our alignment module.
>
> **Table 3. Comparison of GMSD**
>
> | Method      | Syn   | Real  |
> |-------------|:-----:|:-----:|
> | DBSR        | 0.896 | 0.838 |
> | MFIR        | 0.956 | 0.830 |
> | BIPNet      | 0.962 | 0.849 |
> | Burstormer  | 0.966 | 0.847 |
> | BSRT_S      | 0.958 | 0.855 |
> | BSRT_L      | 0.961 | 0.855 |
> | BurstM      | 0.964 | 0.851 |
> | Ours    | **0.967** | **0.857** |
>
> > **Q8:** Impact of Frame ... occlusion.
>
> **A8:** We want to clarify that, in BISR, frames are typically captured under random motion, with no fixed temporal order or consistent correlation between frame index and motion magnitude (see Table 4 Input). Therefore, the concern by the reviewer will not be a practical issue in applications. In fact, it can be observed from Table 4 that our alignment remains consistently effective and stable across all frames.
>
> **Table 4. Frame-wise Comparison of GMSD**
>
> | Frame | SyntheticBurst Input | SyntheticBurst Image | SyntheticBurst Feature | BurstSR Input | BurstSR Image | BurstSR Feature |
> |-------|:--------------------:|:---------------------:|:------------------------:|:-------------:|:-------------:|:----------------:|
> | 1     | 0.858                | 0.910                 | 0.940                    | 0.866         | 0.915         | 0.954            |
> | 2     | 0.862                | 0.912                 | 0.940                    | 0.868         | 0.915         | 0.954            |
> | 3     | 0.856                | 0.909                 | 0.939                    | 0.869         | 0.916         | 0.955            |
> | 4     | 0.857                | 0.909                 | 0.939                    | 0.865         | 0.913         | 0.955            |
> | 5     | 0.858                | 0.911                 | 0.939                    | 0.868         | 0.915         | 0.955            |
> | 6     | 0.858                | 0.909                 | 0.940                    | 0.870         | 0.918         | 0.954            |
> | 7     | 0.858                | 0.909                 | 0.939                    | 0.870         | 0.918         | 0.954            |
> | 8     | 0.859                | 0.909                 | 0.940                    | 0.869         | 0.917         | 0.954            |
> | 9     | 0.858                | 0.909                 | 0.939                    | 0.869         | 0.917         | 0.954            |
> | 10    | 0.861                | 0.910                 | 0.939                    | 0.869         | 0.917         | 0.954            |
> | 11    | 0.859                | 0.909                 | 0.940                    | 0.868         | 0.916         | 0.954            |
> | 12    | 0.857                | 0.910                 | 0.939                    | 0.866         | 0.913         | 0.953            |
> | 13    | 0.859                | 0.911                 | 0.940                    | 0.867         | 0.916         | 0.954            |
>
> > **Q9:** Effect of Number ... cost.
>
> **A9:** We follow the standard setting of using 14 input frames (N = 14) as adopted in previous studies on SyntheticBurst and BurstSR for fair comparison. Prior studies [4] show that performance improves with more frames when N < 14. However, the use of more than 14 frames remains unexplored. Incorporating additional frames may introduce challenges such as more complex alignment and increased computational cost as mentioned by the reviewer. We leave this for future investigation by possibly collecting real burst sequences with more frames.
>
>
> > **Q10:** While the proposed ... sequences.
>
> **A10:** As addressed in **A4**, modeling BISR transformation with rotation and translation is reasonable and aligned with the benchmark [1].
> Although rolling shutter effects may occur in real scenarios, such degradations are not emphasized in this task and are typically addressed in a specific way [5], which is out of the domain of BISR.
> The effectiveness of our assumption is supported by the results in Table 4 of **A8** and the comparisons in Figure 2 and Table 1 of the manuscript. We have indeed discussed the limitation of transformation modeling in the Limitation section of the manuscript.
>
> ## Reference
> [1] Goutam Bhat, Martin Danelljan, and Radu Timofte. Ntire 2021 challenge on burst super-resolution: Methods and results. In Proceedings of the IEEE/CVF Conference on Computer Vision and Pattern Recognition (CVPR) Workshops, pages 613–626, 2021.
>
> [2] Di X, Peng L, Xia P, et al. Qmambabsr: Burst image super-resolution with query state space model[C]//Proceedings of the Computer Vision and Pattern Recognition Conference. 2025: 23080-23090.
>
> [3] Kang E G, Lee B, Im S, et al. Burstm: Deep burst multi-scale sr using fourier space with optical flow[C]//European Conference on Computer Vision. Cham: Springer Nature Switzerland, 2024: 459-477.
>
> [4] Wei P, Sun Y, Guo X, et al. Towards real-world burst image super-resolution: Benchmark and method[C]//Proceedings of the IEEE/CVF International Conference on Computer Vision. 2023: 13233-13242.
>
> [5] Yang Z, Li H, Cheng S, et al. Multi-Frame Rolling Shutter Correction with Diffusion Models[J]. IEEE Transactions on Circuits and Systems for Video Technology, 2025.

---

> > ### Comment · Reviewer_ZHcB · 2025-08-05
> > **Thank you for the feedback.**
> >
> > I appreciate the authors' efforts in addressing the concerns raised, but several key issues (such as the methodological novelty, robustness under real-world conditions, and adequacy of evaluation) remain insufficiently resolved. A deeper empirical or theoretical justification would be needed to fully strengthen the claims. Each follow-up response to the authors' reply is as follows.
> >
> > A1: I appreciate the attempt to clarify the contribution, but simply stating that the approach is "novel and non-trivial" does not sufficiently address my concern. Without clearer justification of why the method constitutes a significant advancement beyond prior work, I remain unconvinced that it meets the bar for a top-tier conference.
> >
> > A2: I appreciate the clarification on the roles of MDTA and INR, but the response does not fully address my concern regarding the ad hoc nature of their integration. Since both components are based on existing methods, a more principled justification for their specific combination is needed to support the novelty of the overall design.
> >
> > A3: While I appreciate the clarification, the response largely reiterates prior points without convincingly addressing my concern about the limited scope of the core contributions. In particular, the emphasis on empirical gains feels insufficient to strengthen the paper's novelty from a technical standpoint.
> >
> > A4: While I acknowledge that the method shows improved alignment performance over existing approaches like BurstM, this alone does not fully resolve my concern. The core issue is whether limiting the transformation to rotation and translation is sufficiently general for real-world burst frames, which still requires stronger theoretical or empirical justification.
> >
> > A5: While I appreciate the inclusion of LPIPS, the response would be stronger with a broader set of perceptual metrics, such as FID, and further discussion on the significance of the observed improvements. As it stands, the perceptual evaluation still feels somewhat limited.
> >
> > A6: While I understand that zoom-in visualizations are standard practice for evaluating fine details, the response does not fully address my concern. The qualitative differences remain subtle, and stronger evidence, such as more compelling visual examples or user studies, would be needed to convincingly support the claims.
> >
> > A7: I appreciate the inclusion of the GMSD metric, which is a helpful addition. However, since the performance differences are relatively small, a more detailed interpretation or analysis would help clarify the practical significance of the observed improvements.
> >
> > A8: While I appreciate the clarification and the inclusion of frame-wise results, the response does not fully address my concern. A more direct analysis based on actual motion or content difference from the reference frame (rather than frame index alone) would offer more insight into the robustness of the alignment module under challenging conditions.
> >
> > A9: I understand the use of 14 frames follows prior work, but without empirical analysis or ablation in this paper, it's difficult to assess whether this choice is optimal. A brief study on the impact of varying frame numbers, even within a limited range, would provide valuable insight into the performance–efficiency trade-off.
> >
> > A10: While I understand the rationale for focusing on rotation and translation, the response does not fully resolve my concern. A more detailed discussion or empirical analysis of how the method performs under more complex real-world distortions (such as affine or rolling shutter effects) would strengthen the robustness claims of the proposed approach.

---

> > > ### Author Response · Authors · 2025-08-05
> > >
> > > Thanks for your detailed reply to our response. Here, we want to make further clarifications regarding your comments.
> > >
> > > > A1: I appreciate ... conference.
> > >
> > > We want to kindly argue that, as mentioned in our response and thoroughly presented in our paper, the significance of this work has indeed been justified both theoretically and empirically.
> > >
> > > > A2: I appreciate ... design.
> > >
> > > We have discussed why we adopted these two modules in our previous response. Here, we want to further clarify that we did not intend to specifically combine them, but use them for independent reasons as mentioned in our previous response.
> > >
> > > > A3: While I appreciate ... standpoint.
> > >
> > > As mentioned before, the contributions of this work have been thoroughly presented and justified in our paper both theoretically and empirically. Such contributions have indeed been partially recognized as "Strengths" in the initial comments.
> > >
> > > > A4: While I acknowledge ... justification.
> > >
> > > In addition to our previous response, we want to further emphasize that the rotation-translation transformation is indeed sufficiently general for real-world burst frames. First, theoretically, since the burst frames are captured in a very short time, the transformation can indeed be approximately regarded as rotation-translation, and existing synthetic benchmarks have indeed adopted such an approximation. Second, empirically, our method has shown promising alignment results in the real-captured burst image dataset, verifying the generality of the rotation-translation transformation for real-world burst frames.
> > >
> > > > A5: While I appreciate ... limited.
> > >
> > > Firstly, we notice that in the initial comments, LPIPS and FID are mentioned with "or" for evaluating the perceptual quality. Secondly, LPIPS is generally a more preferable perceptual metric in image restoration tasks, such as super-resolution and deblurring. Specifically, it measures the perceptual distance between one restored image with its corresponding ground-truth at the image level. In contrast, FID measures the perceptual distance between two datasets at the distribution level. In image restoration tasks, since the ground-truth images are available, the image-level reconstruction is generally more desired. We believe that combining both the fidelity metric, such as PSNR, and the LPIPS perceptual metric, the empirical effectiveness of our method can be strongly substantiated.
> > >
> > > > A6: While I understand ... claims.
> > >
> > > As suggested, we will conduct a user study and incorporate it in the revision, though we notice that this requirement was not raised in the initial comments. Thanks again for this constructive suggestion.
> > >
> > > > A7: I appreciate ... improvements.
> > >
> > > We want to further clarify that our method is the only one that can have competitive GMSD on both datasets. Specifically, Burstormer has a GMSD close to that of ours on SyntheticBurst (Syn), but is less competitive on BurstSR (Real); and BSRT models give close GMSDs to that of ours on BurstSR, but are not that effective on SyntheticBurst. Therefore, we believe these results are significant in showing the effectiveness of our method.
> > >
> > > > A8: While I appreciate ... conditions.
> > >
> > > In our previous response, we have indeed shown that "the alignment module performs consistently across all frames". For the large "motion" issue, we want to kindly clarify again that in the burst SR setting, since the frames are captured in a very short time, the "motion" would not be too large, and can be handled by our method, as thoroughly evaluated by our experiments. We want to further kindly argue that, in this work, we aim at proposing a new method for an existing task under the standard setting, which is indeed common sense along this research line, but not trying to raise new problems or tasks.
> > >
> > > > A9: I understand ... trade-off.
> > >
> > > We want to emphasize again that the optimality of 14 frames has been substantiated by previous studies, and challenging this claim (in the performance-efficiency trade-off sense) is out of the scope of our work, though it could be an interesting direction.
> > >
> > > > A10: While I understand ... approach.
> > >
> > > We want to clarify that, in this work, we are just following the standard setting to address an existing image restoration task, and have achieved promising results. We know that our method may have limitations, but we want to kindly argue that it should not be blamed for not being able to solve the problem it is not designed for.

---

> > > > ### Comment · Reviewer_ZHcB · 2025-08-05
> > > > **Thank you for the feedback.**
> > > >
> > > > Thank you for your detailed clarifications. I now have a clearer understanding of the authors' intentions regarding each of the comments and questions.
> > > >
> > > > That said, the proposed approach primarily adapts existing techniques, and as such, the methodological contribution may be perceived as incremental rather than truly innovative. The reconstruction module, in particular, relies on the integration of established components (e.g., MDTA, INR). This further limits the perceived novelty of the work. In addition, the visual improvements over existing methods remain rather subtle(this one was pointed by other reviewer too).
> > > >
> > > > In conclusion, I believe the technical contributions do not sufficiently meet the bar for a top-tier conference, and the qualitative (and even quantitative) results do not significantly outperform prior work (marginally better).
> > > >
> > > > Therefore, I still consider this submission to be borderline. I would encourage the authors to further develop the ideas (addressing the limitations discussed here and incorporating aspects mentioned as future work) and consider resubmitting to a future venue.

---

> ### Author Response · Authors · 2025-08-05
>
> We want to emphasize again that our main contribution, which has been been motivated and discussed in detail the introduction part of the manuscript, is the **theoretically sound feature alignment**, but **NOT** the application existing MDTA or INR. Such a theretically sound feature alignment is the main focus of the paper, and has been thoroughly discussed and justified both theoretically and empirically justified. Specifically, we have proven that the alignment is **theoretically solid**, and empirically demonstrated its **superiority over existing alignment**.
>
> We know that assessing the contribution of a paper is sometimes subjuctive, but we kindly hope that you would be willing to check our paper again to verify our clarifications.

---

### Official Review · Reviewer_yD4L · 2025-07-02

**Clarity:** 2
**Significance:** 2
**Originality:** 3
**Rating:** 4
**Confidence:** 4

**Summary:**

This paper proposes a new framework for Burst Image Super-Resolution (BISR) that addresses the limitations of existing alignment methods. Instead of using deformable convolutions or optical flow, the authors introduce an equivariant convolution-based alignment approach that learns spatial transformations in the image domain and applies them consistently in the feature domain. This design improves alignment accuracy in a theoretically sound manner. The framework is further enhanced with a reconstruction module using MDTA blocks and Implicit Neural Representation (INR) for effective feature fusion and upsampling. Experiments on BISR benchmarks show that the proposed method outperforms existing approaches in both quantitative metrics and visual quality.

**Questions:**

1. Could you please revise the visual comparisons? In their current form, it is difficult to perceive clear differences between methods, which weakens the qualitative evaluation.
2. Most of the baselines are from before 2023. Would it be possible to include one or two more recent state-of-the-art BISR methods for a more up-to-date comparison?
3. In addition to FLOPs, could you also report the actual inference time on GPU? This would help better assess the practical efficiency of your method.

**Ethical Concerns:**

["NO or VERY MINOR ethics concerns only"]

**Final Justification:**

The authors have adequately addressed the concerns I raised in the initial review. The clarifications and additional experimental results strengthen the paper's contributions and validate its technical claims. In light of these improvements, I am willing to raise my overall score.

**Limitations:**

No negative societal impact.

**Paper Formatting Concerns:**

No formatting issue.

**Quality:**

2

**Strengths And Weaknesses:**

Strength:
1. The paper introduces a novel use of Equivariant Convolutional Networks (Eq-CNNs) for alignment, allowing spatial transformations learned in the image domain to be applied accurately in the feature domain.
2. It clearly identifies limitations in existing BISR alignment strategies and provides a well-justified motivation for using Eq-CNNs, supported by theoretical reasoning and intuitive illustrations.

Weakness:
1. The visualizations are not very informative—in several examples, it is difficult to perceive meaningful differences in image quality, which limits the interpretability of the visual comparisons.
2. The comparative baseline methods are relatively outdated, with most published before 2023. Including one or two more recent state-of-the-art BISR methods would strengthen the experimental validation.
3. While the paper reports FLOPs as a measure of efficiency, it would be more convincing to also provide real inference speed (e.g., runtime on GPU) to better reflect practical performance.

---

> ### Author Rebuttal · Authors · 2025-07-30
>
> >**Q1:** The visualizations are not very informative—in several examples, it is difficult to perceive meaningful differences in image quality, which limits the interpretability of the visual comparisons.
>
> **A1:** For interpretability, we have indeed zoomed in or highlighted local regions for each example, and provided textual descriptions in Lines 250–253, 264–266, and 288–289 of the manuscript. For instance, in Figure 4 of the manuscript, we zoom into local details to illustrate improvements in texture (Rows 1–3) and color (Row 4). In revision, we will further revise the figures to make the advantages of our method more visually apparent.
>
> >**Q2:** The comparative baseline methods are relatively outdated, with most published before 2023. Including one or two more recent state-of-the-art BISR methods would strengthen the experimental validation.
>
> **A2:** Considering that BISR is relatively a less hot research topic in image restoration, there are not many methods developed after 2023. In particular, to our knowledge, there are only two papers have been published in top conferences, i.e., BurstM in ECCV 2024 and QMambaBSR in CVPR 2025. We have indeed included BurstM in exmperiments. For QMambaBSR, since it has not released official code, we did not include it in our manuscript, while based on its reported metrics, our method still achieves better results (e.g., PSNR on  SyntheticBurst with the x4 setting: 43.18 dB for our method vs. 43.12 dB for QMambaBSR). We will include comparisons with new methods as they become public.
>
> >**Q3:** While the paper reports FLOPs as a measure of efficiency, it would be more convincing to also provide real inference speed (e.g., runtime on GPU) to better reflect practical performance.
>
> **A3:** We have provided inference time for additional measurement of efficiency in Table 1. It can be observed that, compared to the SOTA method BurstM, our model achieves lower FLOPs and faster inference speed. Furthermore, in comparison with large-scale model BSRT-Large (BSRT_L), our approach shows an apparent advantage in efficiency.
>
>
>
> **Table 1 Comparison of Efficiency**
> | Method      | Param (M) | FLOPs (G) | Time (s) |
> |-------------|:---------:|:---------:|:--------:|
> | DBSR        | 13.01     | 111.71    | 0.239    |
> | MFIR        | 12.13     | 121.01    | 0.398    |
> | BIPNet      | 6.70      | 326.47    | 0.023    |
> | Burstormer  | 2.50      | 38.33     | 0.029    |
> | BSRT_S      | 4.92      | 178.82    | 0.370    |
> | BSRT_L      | 20.71     | 362.63    | 0.499    |
> | BurstM      | 14.00     | 436.21    | 0.033    |
> | Ours        | 8.70      | 170.21    | 0.028    |

---

> > ### Comment · Reviewer_yD4L · 2025-08-06
> >
> > The authors have adequately addressed the concerns I raised in the initial review. The clarifications and additional experimental results strengthen the paper's contributions and validate its technical claims. In light of these improvements, I am willing to raise my overall score to borderline accept from borderline reject.

---

> > > ### Author Response · Authors · 2025-08-07
> > >
> > > Thank you for your comments and support!

---

### Official Review · Reviewer_UMQ2 · 2025-07-03

**Clarity:** 3
**Significance:** 2
**Originality:** 2
**Rating:** 3
**Confidence:** 3

**Summary:**

This paper addresses Burst Image Super-Resolution (BISR) by introducing a new alignment framework based on equivariant convolutional neural networks (Eq-CNNs), which enables theoretically principled alignment transformations learned in the image domain to be applied effectively in the feature domain. The authors propose a full pipeline incorporating this alignment scheme, combined with advanced modules for feature upsampling and fusion. Experimental results on standard benchmarks (SyntheticBurst and BurstSR) validate the method’s effectiveness, showing improvements in both quantitative metrics and visual quality over several state-of-the-art approaches. The paper also offers a theoretical analysis supporting the proposed alignment strategy.

**Questions:**

See weaknesses

**Ethical Concerns:**

["NO or VERY MINOR ethics concerns only"]

**Final Justification:**

The overall quality of the paper is not a major issue, but the novelty, significance, and improvement are too limited. While a 0.1 dB improvement can be meaningful and significant for some image restoration tasks where the baseline PSNR is around 20–30, in this paper, the baseline PSNR is already above 40, making the improvement very marginal.

**Limitations:**

yes

**Paper Formatting Concerns:**

no formatting concerns

**Quality:**

3

**Strengths And Weaknesses:**

Strength:

1. Introducing Eq-CNNs into burst super-resolution is novel, as no prior work has explored this direction, demonstrating a certain level of innovation.

2. The paper is clearly written and easy to follow. It provides theoretical justification for how Eq-CNNs enable better alignment, and clearly describes the task setting of burst image super-resolution.

3. The authors conduct thorough ablation and comparative experiments to demonstrate the effectiveness of the proposed method. Additionally, they visualize residuals to show that Eq-CNNs achieve lower reconstruction errors and more efficient feature extraction.

Weakness:

1. Though the approach is theoretically well-founded, the quantitative improvements reported in Table 2 over state-of-the-art methods (e.g., BurstM, BSRT-Large) are quite marginal, particularly on realistic benchmarks like BurstSR, where the PSNR gain is as small as 0.01 or less. Specifically, in Figure 5, the visual enhancement achieved by the proposed method appears limited, and there remains a noticeable color deviation compared to the ground truth, similar to other competing approaches.

2. The content of Proposition 1 largely overlaps with Section 3.3 "Theoretical Results," making it redundant. It is recommended to remove one of them to improve conciseness and avoid repetition.

3. The ablation studies (Table 3, Figures 6/7) are primarily on SyntheticBurst, not the real BurstSR data, making it unclear how robust the alignment strategy is under real-world acquisition artifacts.

4. It is suggested to compare with more recent burst super resolution methods such as QMambaBSR.

---

> ### Author Rebuttal · Authors · 2025-07-30
>
> >**Q1:** Though the approach is theoretically well-founded, the quantitative improvements reported in Table 2 over state-of-the-art methods (e.g., BurstM, BSRT-Large) are quite marginal, particularly on realistic benchmarks like BurstSR, where the PSNR gain is as small as 0.01 or less. Specifically, in Figure 5, the visual enhancement achieved by the proposed method appears limited, and there remains a noticeable color deviation compared to the ground truth, similar to other competing approaches.
>
> **A1:** We want to clarify that the PSNR improvement on the BurstSR dataset is 0.10 dB, rather than 0.01, which can be regarded significant enough in image restoration and enhancement studies.
>
> Moreover, to quantify the color deviation noticed by the reviewer, we additionally report the Delta E (ΔE\*) metric [1], which directly measures perceptual color differences, in Table 1. It can be seen that our method consistently achieves lower ΔE* values compared to prior methods, indicating its better color preservation.
> Regarding the color deviation in Figure 5, this might be partly attributable to inherent limitations in the RAW-to-sRGB conversion process used in dataset generation. Similar artifacts have been noted in prior studies such as [2] on the BurstSR dataset. Another possible reason is that, due to the limited amount of real-world aligned data, existing methods, including ours, are primarily first trained on SyntheticBurst and then fine-tuned on BurstSR, which may not sufficiently capture the complex color statistics in real-world scenes. In particular, as shown in Table 1, all methods exhibit relatively more severe color shifts on BurstSR than on SyntheticBurst. Nonetheless, we acknowledge this color deviation issue should be emphasized for the BusrSR task and will further investigate it.
>
>
> **Table 1. Comparison of ΔE\* on SyntheticBurst and BurstSR**
> | Method      | SyntheticBurst | BurstSR |
> |-------------|--------|---------|
> | DBSR        | 2.60   | 6.68    |
> | MFIR        | 2.44   | 6.47    |
> | BIPNet      | 2.29   | 6.33    |
> | Burstormer  | 2.12   | 6.27    |
> | BSRT_S      | 2.75   | 5.87    |
> | BSRT_L      | 2.15   | **5.57**    |
> | BurstM      | 2.06   | 6.00    |
> | Ours        | **2.03**   | **5.57**    |
>
> > **Q2:** The content of Proposition 1 largely overlaps with Section 3.3 "Theoretical Results," making it redundant. It is recommended to remove one of them to improve conciseness and avoid repetition.
>
> **A2:** Thanks for the suggestion, and we will revise the manuscript accordingly to improve the conciseness.
>
>
> >**Q3:** The ablation studies (Table 3, Figures 6/7) are primarily on SyntheticBurst, not the real BurstSR data, making it unclear how robust the alignment strategy is under real-world acquisition artifacts.
>
> **A3:** We want to clarify that our alignment strategy has indeed been evaluated on the real BurstSR data in the main experiments. Specifically, as shown in Table 1 and Figure 2 of the manuscript, our method demonstrates clear advantages over flow-based alignment on the real BurstSR dataset in both of the image domain and the feature domain. These results provide evidence of the robustness of our alignment module under real-world conditions.
>
> As for the ablation studies, we followed the same experimental settings as prior studies, which all conducted ablations on SyntheticBurst. Moreover, since models for BurstSR are typically first trained on SyntheticBurst and then fine-tuned (due to the limited availability of well-aligned real bursts), ablation on SyntheticBurst offers a more stable and interpretable evaluation framework.
>
> >**Q4:** It is suggested to compare with more recent burst super resolution methods such as QMambaBSR.
>
> **A4:** As QMambaBSR has not released official code so far, a direct comparison was not feasible. Nevertheless, based on the reported results in its paper, our method achieves comparable or even better performance (e.g., PSNR on  SyntheticBurst with the x4 setting: 43.18 dB for our method vs. 43.12 dB for QMambaBSR). We will add a direct comparison once the official code become publicly available.
>
> # Reference
>
> [1] Backhaus, Werner GK, Reinhold Kliegl, and John S. Werner, eds. Color vision: Perspectives from different disciplines. Walter de Gruyter, 2011.
>
> [2] EungGu Kang, Byeonghun Lee, Sunghoon Im, and Kyong Hwan Jin. Burstm: Deep burst multi-scale sr using fourier space with optical flow. In European Conference on Computer Vision, pages 459–477.Springer, 2025.

---

> > ### Comment · Reviewer_UMQ2 · 2025-08-06
> >
> > Sorry for the typo. Regarding the statement, "We want to clarify that the PSNR improvement on the BurstSR dataset is 0.10 dB"—while a 0.1 dB improvement can be meaningful and significant for some image restoration tasks where the baseline PSNR is around 20–30, in this paper, the baseline PSNR is already above 40, making the improvement very marginal here.

---

> ### Author Response · Authors · 2025-08-06
>
> Thanks for your reply.
>
> Firstly, we want to kindly clarify that in addition to the results on the BurstSR dataset, the PSNR improvement of our method over BurstM on the SyntheticBurst dataset with the x4 setting, which is a widely used setting in the burst SR studies, is **0.31dB**. In addition, the 0.1dB improvement is indeed non-trivial for the burst SR task. Specifically, in the original BurstM paper, the PSNR improvement of BurstM over Burstormer on the SyntheticBurst dataset with the x4 setting is only **0.04 dB**. Comprehensively considering all the results across all the datasets and settings in Table 2 of our paper, we believe the performance improvement of our method should not be considered as marginal.
>
> Secondly, in addition to the performance improvement, we want to kindly emphasize that the proposed theoretically sound feature alignment method,  which has been recognized as a strength in the initial comments, is of more significance. As thoroughly discussed and justified in our paper, such a theoretically sound feature alignment method and corresponding theoretical analysis not only advances the burst SR task itself but also contributes to Eq-CNN theories, and these contributions could be more important to the related research areas.

---

### Official Review · Reviewer_AJEB · 2025-07-03

**Clarity:** 3
**Significance:** 3
**Originality:** 3
**Rating:** 3
**Confidence:** 4

**Summary:**

This paper is interesting. Authors propose to estimate the alignment in image domain and apply the alignment in feature domain. As well as a new up-sampling network, new SOTA results are acheived.

**Questions:**

Will results be better with sub-pixel alignment?

**Ethical Concerns:**

["NO or VERY MINOR ethics concerns only"]

**Limitations:**

What will be the improvments if the movements are small in different burst images ?

**Quality:**

3

**Strengths And Weaknesses:**

A new idea to align in feature domain for burst image SR.
Similar idea has been used in other SR tasks.

Strengths
1. Theoretically Sound Alignment: By leveraging Eq-CNN, the framework ensures consistency between transformations in the image and feature domains. This allows alignment transformations learned via image-domain supervision to be directly applied to features, improving alignment accuracy (as shown by lower feature alignment errors in Table 1 and Figure 2).
2. Effective Feature Alignment: Compared to optical flow or deformable convolutions, the proposed method reduces feature misalignment, especially on real-world datasets (BurstSR), where it achieves a 1.94 relative error vs. 2.06 for flow-based methods (Table 1).
Advanced Reconstruction Module: Integrating MDTA (for capturing global inter-frame correlations) and INR (for multi-scale upsampling) enhances feature fusion and detail recovery, contributing to superior HR reconstruction.
3. Superior Experimental Performance: On both synthetic (SyntheticBurst) and real (BurstSR) datasets, the method outperforms SOTA methods (e.g., BurstM, BSRT-Large) in PSNR, SSIM, and visual quality, preserving finer textures and reducing artifacts (Tables 2, 4-5; Figures 4-5).

Weaknesses
1. Limited Transformation Modeling: The framework only models rotation and translation transformations due to the constraints of existing Eq-CNNs. This may not capture complex real-world motions (e.g., non-rigid deformations), limiting applicability in scenarios with intricate misalignments.
2. Dependence on Eq-CNN Capabilities: Performance is bounded by the equivariance of Eq-CNNs, which currently lack support for more general transformations (e.g., affine or projective transformations).
3. Potential Overhead in Feature Extraction: Eq-CNNs may introduce additional computational complexity compared to vanilla CNNs, though experiments show the model remains efficient relative to SOTA methods (Table 2).
4. Limited Generalization to Extreme Degradations: While effective on benchmark datasets, the method’s performance on highly noisy or severely blurred burst frames is not explicitly evaluated, leaving room for further validation.

---

> ### Author Rebuttal · Authors · 2025-07-30
>
> > Q1: Limited Transformation Modeling: The framework only models rotation and translation transformations due to the constraints of existing Eq-CNNs. This may not capture complex real-world motions (e.g., non-rigid deformations), limiting applicability in scenarios with intricate misalignments.
>
> **A1:** Though might not be perfect, we believe that our assumption is practical due to the nature of the burst super-resolution task, i.e., the burst frames are captured in a very short time, and the widely used benchmark has adopted the similar assumption [1]. In addition, we also compare our alignment results both in the image and feature domains with optical flow, which is stated as an effective alignment method in the SOTA method BurstM [2], as shown in Table 1 and Figure 2 of the main text. The results suggest that our method can indeed better deal with the misalignment issue than optical flow in real-world scenarios.
> Indeed, we have discussed this limitation in the manuscript (see Section 5) for future exploration.
>
> >Q2: Dependence on Eq-CNN Capabilities: Performance is bounded by the equivariance of Eq-CNNs, which currently lack support for more general transformations (e.g., affine or projective transformations).
>
> **A2:** We agree that the current design leverages Eq-CNNs primarily for their rotation and translation equivariance. As noted in **A1**, this transformation group already provides a sufficient modeling capacity for typical misalignments encountered in burst image sequences. At the same time, we have explicitly discussed this limitation in our manuscript (see Section 5) and consider the extension to broader transformation groups (e.g., affine or projective) a promising direction for future research.
>
> >Q3: Potential Overhead in Feature Extraction: Eq-CNNs may introduce additional computational complexity compared to vanilla CNNs, though experiments show the model remains efficient relative to SOTA methods (Table 2).
>
> **A3:** We want to clarify that Eq-CNNs do not introduce extra computational cost compared to vanilla CNNs during inference. The reason is that the convolution operations are exactly the same between the Eq-CNN and vanilla CNN in implementation during inference. Therefore, if the total numbers of channels are same in each layer between the two kinds of networks, the computational complexity will be theoretically of no difference (more details can be found in [3] and the corresponding code implementation). Such a theoretical property leads to the efficiency of the proposed method shown in Table 2 as noted by the reviewer.
>
> >Q4: Limited Generalization to Extreme Degradations: While effective on benchmark datasets, the method’s performance on highly noisy or severely blurred burst frames is not explicitly evaluated, leaving room for further validation.
>
> **A4:** Our current experiments are conducted on existing benchmark datasets, which are designed to provide fair and standardized comparisons for burst super-resolution. It is worth noting that these datasets already include realistic noise degradations, and our method demonstrates strong performance under such conditions. Scenarios involving other degradations, such as severe blur or intense noise, are typically studied under separate tasks (e.g., burst denoising or deblurring [4,5]) and often require task-specific architectures or loss functions, which is beyond of the scope of this work. Nevertheless, we acknowledge this as a valuable direction and will consider extending our framework to handle such challenging conditions in future work.
>
> >Q5: Will results be better with sub-pixel alignment?
>
> **A5:** Our framework already supports sub-pixel alignment through the feature transformation modeling inherent in the learned sub-pixel alignment and interpolation module. Nonetheless, integrating explicit sub-pixel alignment strategies could potentially yield further improvements, and would be an interesting direction of future work.
>
> >Q6: What will be the improvements if the movements are small in different burst images ?
>
> **A6:**  It should be noted that the degree of motion naturally varies across different burst sequences, and our method is designed to handle both small and large motions effectively. In cases of smaller movements, the learning of precise alignment becomes easier, often leading to improved reconstruction quality.
>
> ## Reference
> [1] Goutam Bhat, Martin Danelljan, and Radu Timofte. Ntire 2021 challenge on burst super-resolution: Methods and results. In Proceedings of the IEEE/CVF Conference on Computer Vision and Pattern Recognition (CVPR) Workshops, pages 613–626, 2021.
>
> [2] Eunggu Kang, Byeonghun Lee, Sunghoon Im, and Kyong Hwan Jin. Burstm: Deep burst multi-scale sr using fourier space with optical flow. In European Conference on Computer Vision, pages 459–477.Springer, 2025.
>
> [3] Qi Xie, Qian Zhao, Zongben Xu, et al. Fourier series expansion based filter parametrization for equivariant convolutions[J]. IEEE Transactions on Pattern Analysis and Machine Intelligence, 2022, 45(4): 4537-4551.
>
> [4] Marius Tico. Multi-frame image denoising and stabilization[C]//2008 16th European Signal Processing Conference. IEEE, 2008: 1-4.
>
> [5] Jianfeng Cai, Hui Ji, Chaoqiang Liu, et al. Blind motion deblurring using multiple images[J]. Journal of computational physics, 2009, 228(14): 5057-5071.

---

### Comment · Area_Chair_pExP · 2025-08-05
**Please begin your discussion with authors.**

Dear all reviewers,

Thanks for your help in assisting with the review of NeurIPS. The authors have provided the rebuttal. If possible, please refer to the comments from other reviewers and read the authors' rebuttal carefully. Please provide your further comments if you have any new questions, and begin the discussion.

Best wishes!

Your AC

---

### Note · Authors · 2025-08-14

We express our gratitude to all the reviewers for their valuable insights!

First of all, we are encouraged that the reviewers have some positive impressions of our work, including:
- **Introduces a novel Eq-CNN-based alignment method:** The proposed alignment method accurately transfers spatial transformations from image to feature domain, which is ignored in the prior works.
- **Theoretically justified:** The approach is theoretically sound and reduces feature misalignment compared to existing methods.
- **The paper is clearly written and easy to follow:** The work clearly describes the task setting of burst image super-resolution and provides theoretical and experimental justification.
- **Superior experimental performance:** The work conducts thorough ablation and comparative experiments to demonstrate the effectiveness of the proposed method.

These comments have, to some extent, recognized our contributions to the Burst SR task, as well as the Eq-CNN theory.

In the rebuttal and discussion period, we have responded to all the reviewers to address their concerns, and most of our responses have been accepted by the reviewers, except that Reviewer AJEB did not participate in the discussion with us. The remaining concerns, as we can understand from the reviewers' feedback, are the **contribution and novelty** (Reviewer ZHcB) and the **marginal performance improvement** (Reviewers UMQ2, ZHcB). Here, we want to make further clarifications about them.

- Regarding the **contribution and novelty**, we want to kindly emphasize that our main contribution is the **theoretically sound feature alignment**, as recognized as one of the strengths by all the reviewers, but not the application of existing MDTA or INR as concerned by Reviewer ZHcB. Such a sound feature alignment has been thoroughly discussed and justified both theoretically and empirically, and not only facilitates the Burst SR task, but also contributes to the Eq-CNN theory.

- Regarding the **marginal performance improvement**, as our reply to Reviewer UMQ2, we believe the quantitative results are sufficient to show the superiority of the proposed method. For the visual or qualitative assessment, which is relatively subjective, though we suppose the current results shown in the main text and appendix are clear, we acknowledge that we could do better, and promise to conduct a user study in the final version.

We appreciate all the reviewers' and AC's great efforts in the whole reviewing process!

---

### Decision · Program_Chairs · 2025-09-17

**Decision:**

Reject

**Comment:**

This paper proposes a burst image super-resolution (BISR) framework leveraging equivariant convolutional neural networks (Eq-CNNs) to learn alignment transformations in the image domain and transfer them consistently to the feature domain. The method integrates MDTA blocks and Implicit Neural Representation (INR) for reconstruction. Experiments on SyntheticBurst and BurstSR datasets show modest improvements over prior methods.  The strengths of this work are (i) the first application of Eq-CNNs to burst ISR as an alternative to flow- or deformable-based alignment; (ii) clear motivation and presentation, with sound theoretical analysis of the alignment framework.  The main weaknesses are as follows: (i) improvements over strong baselines (e.g., BurstM, BSRT) are not large and obvious; (ii) alignment is limited to rotation/translation equivariance, which may not generalize to real-world conditions with more complex motion. The reviewers agree that the use of Eq-CNNs for BISR alignment is novel and theoretically motivated. However, there is broad concern that the methodological contribution is incremental, as Eq-CNNs and MDTA/INR are existing components, and their combination does not demonstrate sufficient originality. Two reviewers (AJEB, UMQ2) raised strong concerns about the marginal performance gains (≈0.1 dB on BurstSR, where PSNR is already high) and limited novelty. Reviewer ZHcB also maintained a borderline reject rating with high confidence. Reviewer yD4L acknowledged these issues but was more convinced by the rebuttal, upgrading the rating to borderline accept. Overall, the consensus leans toward rejection. In the discussion phase, the authors provided clarifications on novelty, real-world application, limited performance, and additional ablations. While these responses addressed some questions raised by reviewers, they did not fundamentally resolve the concerns. Reviewers (UMQ2, ZHcB, AJEB) remained unconvinced and considered the clarifications incremental rather than transformative. Only one reviewer (yD4L) adjusted the rating upward, which does not alter the overall consensus.
Based on the comments from authors and reviewers. I have read this work and find the motivation is significant. However, the subjective comparison and experimental results are absolutely limited. I appreciate the author's theoretical contribution. However, compared with other works, this work cannot meet the bar of NeurIPS now. Therefore, I recommend rejection of this work.